# Genetic and epigenetic features of bilateral Wilms tumor predisposition in patients from the Children's Oncology Group AREN18B5-Q

Andrew J. Murphy [1,2,19,20] ✉, Changde Cheng [3,19], Justin Williams[3], Timothy I. Shaw [3], Emilia M. Pinto [4], Karissa Dieseldorff-Jones [3], Jack Brzezinski[5], Lindsay A. Renfro[6], Brett Tornwall[7], Vicki Huff[8], Andrew L. Hong [9], Elizabeth A. Mullen [10], Brian Crompton [10,11], Jeffrey S. Dome [12], Conrad V. Fernandez [13], James I. Geller[14], Peter F. Ehrlich[15], Heather Mulder [3], Ninad Oak [16], Jamie Maciezsek[4], Carolyn M. Jablonowski[1], Andrew M. Fleming [1,2], Prahalathan Pichavaram[1], Christopher L. Morton[1], John Easton [3], Kim E. Nichols [16], Michael R. Clay[17], Teresa Santiago[4], Jinghui Zhang [3], Jun Yang [1], Gerard P. Zambetti [4], Zhaoming Wang [3,18], Andrew M. Davidoff [1,2] & Xiang Chen [3,20] ✉

Developing synchronous bilateral Wilms tumor suggests an underlying (epi) genetic predisposition. Here, we evaluate this predisposition in 68 patients using whole exome or genome sequencing ($n = 85$ tumors from 61 patients with matched germline blood DNA), RNA-seq ($n = 99$ tumors), and DNA methylation analysis ($n = 61$ peripheral blood, $n = 29$ non-diseased kidney, $n = 99$ tumors). We determine the predominant events for bilateral Wilms tumor predisposition: 1)pre-zygotic germline genetic variants readily detectable in blood DNA [*WT1* (14.8%), *NYNRIN* (6.6%), *TRIM28* (5%), and BRCA-related genes (5%)] or 2)post-zygotic epigenetic hypermethylation at 11p15.5 *H19/ICR1* that may require analysis of multiple tissue types for diagnosis. Of 99 total tumor specimens, 16 (16.1%) have 11p15.5 normal retention of imprinting, 25 (25.2%) have 11p15.5 copy neutral loss of heterozygosity, and 58 (58.6%) have 11p15.5 *H19/ICR1* epigenetic hypermethylation (loss of imprinting). Here, we ascertain the epigenetic and genetic modes of bilateral Wilms tumor predisposition.

Wilms tumor (WT) is the most common kidney cancer of childhood and 5–7% of WT patients present with synchronous bilateral Wilms tumor (BWT)[1]. BWT development is highly suggestive of an underlying genetic or epigenetic predisposition. In 1972, Knudson and Strong hypothesized that, like retinoblastoma, familial WT and BWT developed from two genetic events (two-hit hypothesis), the first being either prezygotic (i.e., germline) or postzygotic (i.e., somatic in the early embryo) and the second always postzygotic[2]. In support of this hypothesis, patients with BWT have a younger median age at diagnosis

than those with unilateral WT[3]. Furthermore, WT precursor lesions known as nephrogenic rests (postnatal persistent clusters of undifferentiated embryonic kidney cells) and multifocal WT are more commonly present in patients with BWT than unilateral WT, supporting the concept of predisposition followed by stepwise accumulation of additional postzygotic somatic events leading to tumor development[4].

When compared with unilateral WT, BWT is more frequent in patients with structural birth defects and known predisposition for

WT, including *WT1* disorder (congenital/infantile or childhood onset of steroid-resistant nephrotic syndrome, genitourinary anomalies, predisposition for WT) and WAGR (Wilms tumor, aniridia, genitourinary anomalies, range of developmental delays)[5–10]. In addition, BWT has an increased predisposition in patients with Beckwith Wiedemann spectrum disorder (BWSp), implicating dysregulation of imprinting at chromosome 11p15.5, a region which houses a cluster of imprinted genes including the growth factor *IGF2*[11–13]. The expression of genes and noncoding RNAs at 11p15.5 is controlled by differential methylation of two imprinting control regions (ICR): *H19/ICR1* and *KCNQ1OT1/ICR2* (Supplementary Fig. 1). Among patients with BWSp, those with epigenetic gain of methylation at *H19/ICR1* (loss of imprinting - LOI) or paternal uniparental disomy (loss of genetic material from the maternal 11p15.5 locus with duplication of the paternal allele in this region; a state known as copy neutral loss of heterozygosity−LOH), both of which result in biallelic expression of *IGF2*, have the highest risk for any WT development[13,14]. The BWSp disease phenotype can vary in severity according to the type of molecular alteration at 11p15.5 and/or somatic mosaic distribution of the alteration throughout the body[15]. Somatic mosaicism refers to two (epi)genetically distinct populations of cells that are found in the same individual, due to a (epi)genetic variant that occurs after fertilization (i.e. postzygotic). This is in contrast to a germline (epi)genetic event, that occurs prior to fertilization and is therefore found in every cell of the body in an individual[16]. In fact, some patients with mosaic distribution of 11p15.5 abnormalities do not have overt syndromic features and are first diagnosed by detection of subtle clinical abnormalities and germline evaluation at the time of embryonal tumor presentation[17–20]. For this reason, international consensus guidelines now refer to Beckwith Wiedemann as a spectrum (BWSp) that can be diagnosed using clinical criteria or through molecular testing[21].

Known genetic variants associated with unilateral WT, including germline pathogenic variants in *WT1* and somatic pathogenic variants in *CTNNB1*, are thought to occur with increased frequency in BWT[22]. Recent studies have suggested that BWT predisposition in some patients is due to post-zygotic somatic mosaic hypermethylation at *H19/ICR1*, which results in clonal expansion of histologically normal renal cells during kidney development (clonal nephrogenesis) and subsequent bilateral and multifocal WT development[23,24]. However, a comprehensive assessment of the genetic/epigenetic landscape of predisposition for BWT remains undescribed.

The purpose of this study is to determine the landscape of genetic and other molecular events predisposing to BWT using a large cohort of BWT specimens from St. Jude Children's Research Hospital (SJCRH) and the Children's Oncology Group (COG). We hypothesize that paired synchronous BWT specimens will exhibit shared genetic or epigenetic predisposing molecular events, while also harboring secondary somatic variants unique to each tumor. Shared genetic or epigenetic events detected in paired synchronous BWT specimens (i.e. in both right and left kidney tumor samples) can be spatially and temporally inferred to occur prior to the lateralization of the mesodermal layer during embryonic gastrulation[25]. This study provides a comprehensive assessment of the genetic and epigenetic features of predisposition for BWT, the most common of which is somatic mosaic 11p15.5 *H19/ICR1* hypermethylation (LOI), which is shared among synchronous BWT samples and often detectable in adjacent non-diseased kidney.

## Results
### Germline variant analysis from blood and associated tumor findings
Sixty-one patients (SJCRH *n* = 11, COG *n* = 50) with available leukocyte-derived peripheral blood DNA were first assessed focusing on cancer predisposing germline genetic variants (Fig. 1). Overall, blood germline variants in pediatric cancer or WT predisposition genes were found in 25/61 (41%) BWT patients (Fig. 2; Supplementary

Data 1). Of these 25 patients, 20 had one predisposing germline variant (Fig. 2) and five had two predisposing germline variants: SJWLM066773 (*BRCA1* and *TRIM28*), SJWLM066792 (*NYNRIN and ASXL1*), SJWLM066774 (*WT1* and *TRIP13*), SJWLM066788 (*NYNRIN* and *KDM3B*), and SJWLM069390 (*NYNRIN* and *CTR9*). Inactivating pathogenic germline variants in *WT1*, a transcription factor critical for normal renal development that has tumor suppressor function in WT[26], were the most common and found in 9/61 (14.8%) patients. Of 9 patients with pathogenic *WT1* germline variants in this study, 3 had features of genetic syndromes and 6 did not. SJWLM066776 had Denys Drash syndrome, SJWLM066772 had a disorder of sexual development (DSD), and SJWLM066774 had congenital nephrotic syndrome and idiopathic dilated cardiomyopathy. For 14 tumor samples from 9 patients with germline *WT1* variants, 11p15.5 LOH (paternal uniparental disomy) determined by methylation analysis and/or the CONSERTING algorithm was present in all tumors (Fig. 2). Out of 14 tumor samples from the 9 patients with germline *WT1* variants, 10 (71.4%) were found to have acquired tumor somatic activating variants in exon 3 of *CTNNB1*, which codes for the Wnt pathway effector transcription factor β-Catenin. *CTNNB1* variants were the most common tumor somatic genetic variants in this cohort and converged on Serine at codon 45, a critical residue that is phosphorylated to control nuclear translocation of β-Catenin[22]. Among 10 total samples harboring blood germline *WT1* variants, somatic tumor 11p15.5 LOH, and somatic tumor *CTNNB1* variants, there were three sets of paired synchronous BWT (SJWILM066776, 066780, 051028) in which each of the paired tumors had somatic exon 3 *CTNNB1* variants. In two of these three cases, the tumor *CTNNB1* variants were distinct (SJWLM066776 *CTNNB1* p.T41A vs. p.S45del, SJWLM051028 *CTNNB1* p.S45P vs. p.S45del) and in the remaining case (SJWLM066780) the *CTNNB1* p.S45F variant was shared in both paired tumors (Fig. 2).

Blood germline variants in *NYNRIN*, a gene for which biallelic truncating variants were previously associated with hereditary WT, were found in 4/61 patients (6.6%)[27]. Germline variants in *TRIM28*, which encodes a nuclear transcriptional co-repressor that coordinates the deposition of repressive histone marks, were found in 3/61 patients (5%), two of whom exhibited epithelial predominant histology in at least one of their tumors, as has been previously shown for germline *TRIM28*-associated WT[28,29]. All blood germline *TRIM28*-associated WT were found to have normal tumor chromosome 11p15.5 copy number with ROI. One patient was found to have a *BRCA1* blood germline missense variant of uncertain significance (p.Q687P; SJWLM066773), one patient was found to have a pathogenic frameshift variant (p.T2766fs; SJWLM066783) in *BRCA2*, and one patient was found to have a variant of uncertain significance with predicted deleterious effect by PROVEAN in the BRCA2-related gene *PALB2* (p.S1155C; SJWLM069379). Loss of function variants in genes coding for the BRCA1-PALB2-BRCA2 protein complex, required for DNA homologous recombination repair, are associated with genomic instability and development of breast and ovarian cancer[30]. Blood germline *DICER1* splice site variants of uncertain significance were found in 3 patients (Fig. 2). DICER1 is an endoribonuclease critical for the generation of microRNAs and hereditary pathogenic variants in *DICER1* cause the *DICER1* hereditary cancer predisposition syndrome[31]. Pathogenic tumor somatic variants in microRNA processing genes are commonly found in WT and germline *DICER1* variants have been associated with WT in rare cases[32,33]. No patient in this cohort was found to have germline numeric or structural alterations on chromosome 11p15.5 at established thresholds for germline 11p15.5 LOI or LOH in leukocyte-derived DNA from peripheral blood; however, mosaic 11p15.5 LOH was identified in the peripheral blood of patient SJWLM069390, who also had 11p15.5 LOH detected in an adjacent normal kidney sample and who was reported to have clinical features of BWSp (Supplementary Fig. 2).

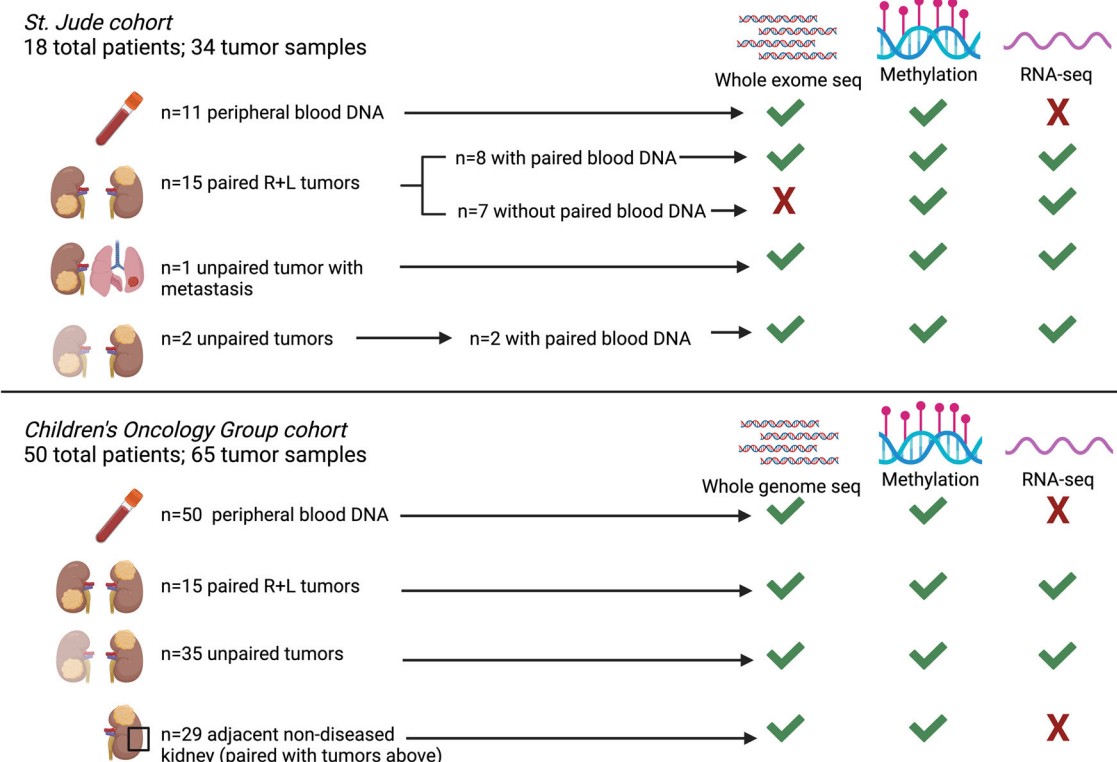

**Fig. 1 | Specimens and molecular assays used in the current study.** Whole exome or whole genome sequencing germline variant calls were made using DNA obtained from 61 patients with available DNA from peripheral blood (*n* = 11 SJCRH and *n* = 50 COG). Paired tumor sets are samples from both right and left tumors in a patient with synchronous BWT (*n* = 30). Unpaired tumors are samples from either the right or left tumor in a patient with synchronous BWT, but for whom one side was not available for analysis (*n* = 37). Adjacent non-diseased kidney was confirmed by a pathologist and came from patients with tumors in the study. Abbreviations: R – right L – left; Seq -sequencing. Graphic made with biorender.com.

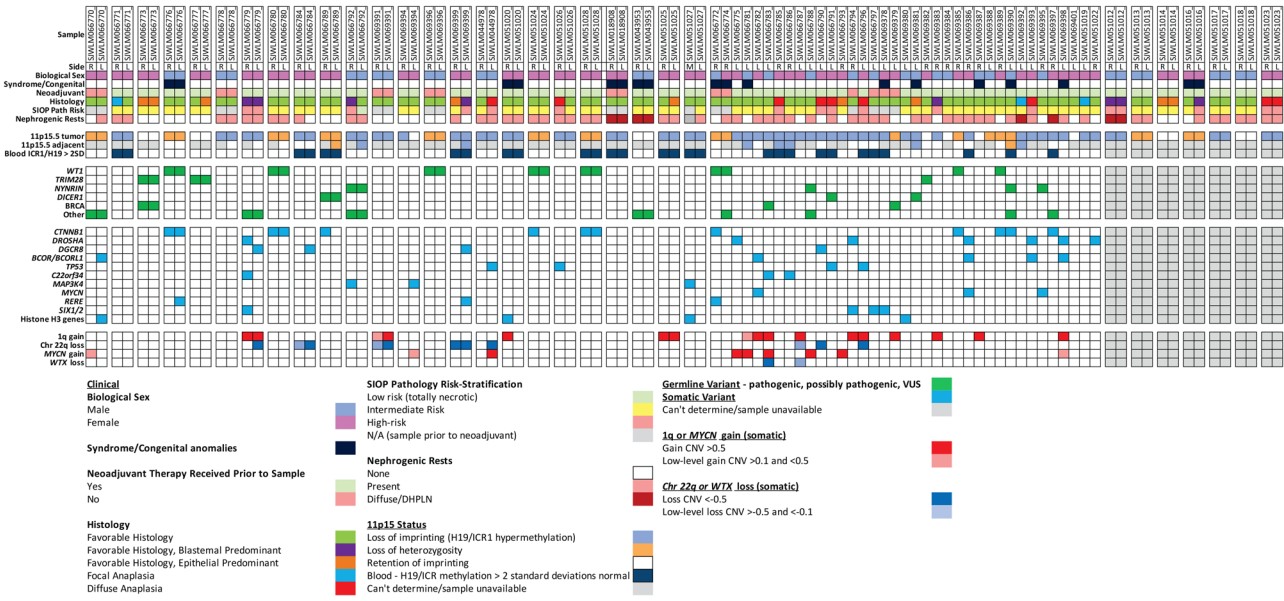

**Fig. 2 | Spectrum of clinical, genetic, and epigenetic features associated with bilateral Wilms tumor.** Paired synchronous BWT (*n* = 23; *n* = 1 primary tumor + corresponding metastasis) are outlined in paired boxes on the left side of the graphic. Additional paired synchronous BWT (*n* = 7) that did not have a germline peripheral blood sample available are shown in paired boxes on the right side of the graphic. These 7 paired synchronous BWT specimens were analyzed for methylation and RNA-seq only. Gray boxes indicate when a given finding could not be assessed due to sample availability. Unpaired specimens from BWT patients are outlined in the adjacent boxes without spacing in the center of the graphic. R right, L left, Adjacent adjacent non-diseased kidney, SD standard deviation, Chr chromosome, DHPLN diffuse hyperplastic perilobar nephroblastomatosis, CNV chromosomal copy number variant. Source data are available in Supplementary Data Files 1, 2 and 5.

The presence of blood germline variants was strongly associated with the tumor chromosome 11p15.5 status (Chi-square $p < 0.0001$). Among 37 tumor samples (from 25 patients) with germline genetic variants, 20 had 11p15.5 LOH (54%), 10 had 11p15.5 LOI (27%), and 7 had 11p15.5 ROI (18.9%). Among 53 tumor samples (from 39 patients with available whole genome or whole exome sequencing data) with 11p15.5 LOI, 43 did not have blood germline variants (81.1%) and 10 did have germline variants (18.9%; Fig. 2). As another way of looking at this association, among 48 tumor samples from patients without predisposing germline variants detected, 43 had 11p15.5 LOI and 5 did not. These data suggest two groups of predisposing events in BWT: 11p15.5 LOI or a germline genetic variant (often followed by 11p15.5 LOH).

## Tumor somatic variants are not shared in synchronous BWT

The heterogeneous histology and treatment-response (Supplementary Fig. 3) seen in BWT suggest the possibility of differing genetic variants among synchronous tumors in the same patient. Therefore, we conducted a paired analysis of 23 synchronous BWT sets with available matched germline DNA. Tumor somatic variants were ascertained by comparing tumor-derived DNA to that derived from peripheral blood leukocytes (Supplementary Data 2). Among these BWT (SJCRH $n = 8$; COG $n = 15$), 21 (91.3%) sample pairs had no shared somatic variants (Table 1). For the remaining two synchronous BWT pairs, one pair shared the *CTNNB1* p.S45F (SJWLM066780) and the other the *ROS1* p.Q1889fs variant (SJWLM069391; Table 1).

Considering all somatic variants determined by whole genome or whole exome sequencing in 85 available tumor samples, activating *CTNNB1* variants (13/85, 15.3%) were the most common, followed by *DROSHA* (7/85, 8.2%), *BCORL1/BCOR* (5/85, 5.9%), *DGCR8* (4/85, 4.7%), *TP53* (4/85, 4.7%), *SIX1/2* (4/85, 4.7%), *C22orf34* (3/85, 3.5%), *MAP3K4* (3/85, 3.5%), *MYCN* (3/85, 3.5%), and *RERE* (3/85, 3.5%). Variants in genes coding for histone H3 (4/85, 4.7%) were also recurrent in this study, including *HIST1H3I* p.K28M in SJWLM069380, *HIST1H3I* p.R117C in SJWLM51020, and *H3F3A* p.R50C in SJWLM66770 and the metastatic sample SJWLM51027 (Fig. 2). DAVID pathway analysis revealed that somatic variants in BWT were associated with the RNA/miRNA biogenesis, p53, and generic transcription pathways (Supplementary Data 3). Of note, no genes with recurrent somatic variants in tumor tissue were found to have variants in adjacent non-diseased kidney tissue (Supplementary Data 4).

Comparison of tumor copy number variants showed markedly different genome wide copy number profiles in paired synchronous BWT except for SJWLM069391 (Supplementary Fig. 4, 5). In addition to SJWLM069391, similar regions of chromosome 22q loss were noted in paired synchronous BWT SJWLM069399 and SJWLM066784 (Supplementary Fig. 5). Overall, chromosome 22q copy number loss was detected in 11/85 (12.9%) of BWT samples. Regarding tumor copy number variants used for risk stratification in completed or upcoming COG WT protocols, copy number gain at 1q was detected in 17/85 (20%) samples and combined LOH of 1p and 16q was found in 2/85 (2.4%) of samples. 1q gain was shared in 3 of 4 synchronous BWT pairs evaluated. Among these three synchronous BWT pairs with shared 1q gain, two pairs exhibited gain of the entire chromosome 1q arm (SJWLM069391, SJWLM066779) and one pair exhibited a different extent of 1q gain in the two tumor samples (SJWLM051025; Supplementary Fig. 4). Taken together, these data suggest that 1q gain was likely an independent genetic event in each tumor rather than from a shared clonal origin (Fig. 2; Supplementary Fig. 4).

Among the COG paired synchronous BWT samples ($n = 15$ patients, 30 tumors) that underwent whole genome sequencing, we compared shared noncoding somatic variants (Table 2). In this analysis, all paired synchronous BWT without shared noncoding somatic variants ($n = 5$) had an identifiable germline variant in a WT or pediatric cancer predisposition gene. For synchronous BWT pairs without an identifiable germline predisposing variant ($n = 6$), 11p15.5 LOI was

detected in five pairs and mixed 11p15.5 LOI/ROI in one pair. In addition, shared tumor somatic noncoding variants were detected in all six pairs without identifiable blood germline predisposing variants (Table 2). Four of these six pairs exhibited only one shared tumor noncoding somatic variant, implying spatial divergence extremely early during embryogenesis. These data suggest that the primary initiators for BWT predisposition are either germline genetic variants (pre-zygotic), or post-zygotic 11p15.5 LOI that can be inferred to occur early in embryogenesis before the right and left kidney primordia lateralize. This inference is supported by the relatively limited number of shared noncoding somatic variants in comparison to the overall number of somatic noncoding variants in each sample (Table 2). Of note, the paired BWT specimens from patient SJWLM069391 were found to have a shared somatic *ROS1* p.Q1889fs variant, 63 shared noncoding variants, and a nearly identical genome-wide copy number profile, which is atypical among these tumor sets and more consistent with multifocal WT from the same kidney rather than paired BWT.

## 11p15.5 status is shared in synchronous BWT

Thirty paired synchronous BWT (SJCRH = 15, COG = 15) were available for methylation analysis of chromosome 11p15.5 using MethylationEPIC beadchip array data. When available, whole genome sequencing data were also used to detect 11p15.5 copy neutral LOH using the CONSERTING algorithm[34]. Tumor purity calculated using a deconvolution-based approach from methylation array data correlated strongly with tumor purity calculated using whole genome sequencing data in the COG specimens (Pearson r = 0.78, $p = 3.2e-13$)[34,35]. Tumor purity estimates and corresponding 11p15.5 *H19/ICR1* and *KCNQ1OT1/ICR2* methylation β values are shown in Supplementary Data 5. Tumor purity positively correlated with *H19/ICR1* methylation β values (Pearson r = 0.43, $p = 8.3e-08$) and negatively correlated with *KCNQ1OT1/ICR2* methylation β values (Pearson r = -0.46, $p = 1.2e-08$), which suggests that tumor purity was a confounding factor when determining the LOH/LOI status of 11p15.5 from the *H19/ICR1* and *KCNQ1OT1/ICR2* methylation β values.

Tumor chromosomal 11p15.5 copy number or methylation status (ROI, LOH, LOI; Supplementary Fig. 1) was shared in 29/30 (96.7%) cases (Fig. 2; Supplementary Fig. 6; Supplementary Data 5). Overall, of 99 total tumor specimens from BWT patients, 15 (15.2%) had 11p15.5 ROI, 25 (25.2%) had 11p15.5 LOH, and 59 (59.6%) had 11p15.5 LOI for *H19/ICR1*. No patients were found to meet established thresholds for germline 11p15.5 LOH (*H19/ICR1* β > 0.7 and *KCNQ1OT1/ICR2* β < 0.3) or LOI (*H19/ICR1* β > 0.7 and *KCNQ1OT1/ICR2* β > 0.3) in their leukocyte-derived peripheral blood germline DNA sample. However, 9/29 (31%) adjacent non-diseased kidney samples met these thresholds for 11p15.5 LOH or LOI (LOH = 2, LOI = 7). When 11p15.5 LOH or LOI was found in adjacent non-diseased kidney tissue, it correlated with the tumor 11p15.5 status in all 9 cases (Fig. 2). Most patients with 11p15.5 LOH or 11p15.5 LOI detected in their tumor samples had no clinical features suggestive of a syndrome (Fig. 2). However, of the 9 patients with definitive 11p15.5 alterations in multiple tissues, SJWLM066781 had 11p15.5 LOI in their tumor and adjacent non-diseased kidney and had a ureteral duplication. SJWLM069381 had 11p15.5 LOI in their tumor and adjacent non-diseased kidney tissue and had hemihypertrophy. SJWLM069390 had 11p15 LOH detected in their tumor and adjacent non-diseased kidney tissue. In addition, this patient was found to have mosaic 11p15.5 LOH in their blood as determined by whole genome sequencing (Supplementary Fig. 2). This patient was noted to have a BWSp clinical phenotype.

Among 19 tumors with 11p15.5 LOH determined by whole genome sequencing, the breakpoints of 11p cnLOH overlapped both the 11p15.5 and *WT1*/11p13 loci in 18/19 cases (94.7%; Supplementary Fig. 7). Among paired synchronous BWT from patients with pathogenic germline *WT1* variants and 11p15.5 LOH detected in each of their tumors, the breakpoints of 11p LOH were not identical when the two

**Table 1 | Somatic variants are almost never shared in paired synchronous bilateral Wilms tumor**

| SJWLM | Blood germline | Somatic variants right kidney tumor | Somatic variants left kidney tumor | Shared both kidney tumors | Adjacent normal kidney | Blood H19/ICR1 methylation > 2 SD from normal |
|---|---|---|---|---|---|---|
| 066770 | REST p.H379P | DHX36, EIF1AX, HCG22, RPS10P7 | ALPK2, BCORL1, GUSBP4, H3F3A, IZKF4, PLCH2, PPFIA4, RBBP4, STAT4, REST, IL2RB, GRLF1, LRRC16B | 11p15.5 LOH | N/A | No |
| 066771 | None | VCX3A | LOC26102, PPCDC, R3HDM1 | 11p15.5 LOI | N/A | Yes |
| 066773 | TRIM28 p.Q318*, BRCA1 p.Q687P | CDK13, TAS2R60 | ITM2C, LOC100133161, LOC339822 | 11p15.5 ROI | N/A | No |
| 066776 | WT1 p.Q253* | AIMIL, CAMK2B, CTNNB1 p.S45F, DHX30, EBF4, NLRC5, OR4C11, PPL | CTNNB1 p.S45del, DBH, DDX60, NANOS1, NELL1, TUBB1, ZFHX4, ATE1, RERE | 11p15.5 LOH | 11p15.5 ROI R + L | No |
| 066777 | TRIM28 p.R795C | NCRNA00245, TCHH | C14orf73, REG3A, SLC12A1 | 11p15.5 ROI | N/A | No |
| 066778 | None | GPR97, IL31RA, RAD17 | CYC1, EMX2OS, KIAA0664, KIF6, MAML3, NES, NR0B1, PKHD1L1, SRRM5, TBX4, TRMT61B, ZNF833P | 11p15.5 LOI | 11p15.5 ROI L | No |
| 066779 | BLM p.K323R | ATP6AP2, C22orf34, DACH1, DROSHA, MAP7D3, PITPNM2, SIX1 | ABCC12, DGCR8, LOC401177, MARCH7, SLC17A7, SIDT2 | 11p15.5 LOI | 11p15.5 LOI L | No |
| 066780 | WT1 p.E401fs | ARHGAP9, C1orf106, CHST3, CTNNB1, OSBPL8, PSME3, RYR1 | C5orf46, CTNNB1, LOC283392, OR52K2, SHROOM2 | 11p15.5 LOH CTNNB1 p. S45F | 11p15.5 ROI L | No |
| 066784 | None | NRP1, ZNF469 | AZU1, C1orf71, DGCR8, DSCAML1, RNASEK | 11p15.5 LOI | 11p15.5 ROI R + L | Yes |
| 066789 | DICER1 M1402_E23 splice region | CTNNB1, FAM120C, GPR27, LOC730755, PPIR13L, RPAP1, SRRSF4, TYRO3, ZNF775 | ACTR10, BICD2, C10orf108, C1orf86, C21orf122, DSC2, IDH3G, KCND3, KIAA0125, NBEAL2 | 11p15.5 LOH | 11p15.5 LOH L | No |
| 066792 | NYNRIN p.R1592* ASXL1 p.V1297I | DNHD1, MAP3K4, QSOX2, SPERT, TROAP | None | 11p15.5 LOI | 11p15.5 LOI L | No |
| 069391 | None | CRAT, ROS1 | NPRL3, ROS1 | 11p15.5 LOI ROS1 p. Q1889fs | N/A | No |
| 069394 | None | SMC1A | EIF1AX, GSPT2, MAP3K4, TRIO, UNC80, VILL | Not shared 11p15.5 LOI – R; 11p15.5 ROI – L | N/A | No |
| 069396 | WT1 p.R441* | None | APOB, NBEAL2, PACS2, KCNQ1 | 11p15.5 LOH | N/A | No |
| 069399 | None | SPTBN5 | CHST6, CLIP1, DGCR8, MAGI3, RERE, SMC1B | 11p15.5 LOI | 11p15.5 LOI L | Yes |
| 044978 | None | MCPH1, STK10 | ATP6V1B2, C9orf131, DOCK9, LCN10, MDN1, PRRC2B, SLC6A20, TP53, ZNF805 | 11p15.5 LOI | N/A | No |
| 051020 | None | HIST1H3I, ZNF664 | HEATR3 | 11p15.5 LOI | N/A | Yes |
| 051024 | WT1 p.T305fs | NLGN3, UBE4A, ZFHX2, CTNNB1 | FAM199X, DSPP, PPP1R11 | 11p15.5 LOH | N/A | No |
| 051026 | None | ACTB, AMAC1, AP2B1, LDHAL6B, LZTR1, NKX2-1, PRRC2B, SEC14L3, TP53, TTC17 | CHD4, EVC, RBMX, CASQ2 | 11p15.5 LOI | N/A | No |
| 051028 | WT1 p.Y337* | GLDN, GPR149, CTNNB1 | CCDC61, CTNNB1, FAM159B, SNTG1, ZNF324, AEBP1 | 11p15.5 LOH | N/A | Yes |
| 018908 | None | CRIPAK, TMEM151B | None | 11p15.5 LOI | N/A | Yes |
| 043953 | CDC73 p.M1T | ANKRD34B, DNAH5, FRG1 | None | 11p15.5 ROI | N/A | No |
| 051025 | None | RYR1, IRS4 | None | 11p15.5 LOI | N/A | Yes |

N/A—sample not available for assessment; 11p15.5 LOI—loss of imprinting (aka ICR1/H19 hypermethylation). 11p15.5 LOH—copy neutral loss of heterozygosity (aka paternal uniparental disomy). 11p15.5 ROI—retention of imprinting (normal physiologic imprinting).

**Table 2 | Shared noncoding variants in paired synchronous bilateral Wilms tumor samples that underwent whole genome sequencing (n = 15)**

| Case ID | Tumor 1 noncoding somatic variant counts | Tumor 2 noncoding somatic variant counts | Shared noncoding somatic variant counts | 11p15.5 Status | Predisposing germline variants |
|---|---|---|---|---|---|
| SJWLM066770 | 136 | 119 | 0 | LOH | *REST* |
| SJWLM066771 | 54 | 74 | 6 | LOI | No |
| SJWLM066773 | 38 | 41 | 0 | ROI | *TRIM28, BRCA1* |
| SJWLM066776 | 122 | 174 | 5 | LOH | *WT1* |
| SJWLM066777 | 69 | 59 | 0 | ROI | *TRIM28* |
| SJWLM066778 | 379 | 68 | 1 | LOI | No |
| SJWLM066779 | 83 | 230 | 0 | LOI | *BLM* |
| SJWLM066780 | 97 | 104 | 0 | LOH | *WT1* |
| SJWLM066784 | 73 | 83 | 1 | LOI | No |
| SJWLM066789 | 125 | 58 | 2 | LOH | *DICER1* |
| SJWLM066792 | 79 | 16 | 1 | LOI | *ASXL1, NYNRIN* |
| SJWLM069391 | 102 | 121 | 63 | LOI | No |
| SJWLM069394 | 43 | 104 | 1 | LOI/ROI | No |
| SJWLM069396 | 38 | 90 | 2 | LOH | *WT1* |
| SJWLM069399 | 136 | 55 | 1 | LOI | No |

*LOH* loss of heterozygosity, *LOI* loss of imprinting, *ROI* retention of imprinting.

tumor samples were compared. The differential breakpoints of 11p LOH between paired synchronous BWT suggests that 11p LOH occurs as an independent genetic event in each tumor in most cases rather than having a shared clonal origin (Supplementary Fig. 7).

**Exploratory analysis of 11p15.5 H19/ICR1 methylation in peripheral blood**

The above analyses demonstrated that 11p15.5 LOI was found in paired synchronous BWT that also shared tumor noncoding somatic variants. In addition, 11p15.5 LOI was found in adjacent non-diseased kidney tissue, but not found in DNA obtained from peripheral blood leukocytes. This pattern in which some, but not all, cells throughout the body are affected by a genetic or epigenetic change that can be inferred to occur in the early embryo based on shared noncoding somatic variants is consistent with post-zygotic somatic mosaicism. However, we reasoned that, if present, post-zygotic somatic mosaicism could manifest as increased methylation at 11p15.5 *H19/ICR1* in peripheral blood cells because hematopoietic progenitor cells have a mesodermal embryonic origin[36]. To explore whether increased 11p15.5 *H19/ICR1* methylation could be detected on the cohort level in peripheral blood samples from patients with tumors bearing 11p15.5 LOI, we compared the *H19/ICR1* β values from leukocyte-derived DNA from peripheral blood among COG cohort patients according to the 11p15.5 status of their tumors (ROI, LOH, LOI). We found a statistically significant increase in *H19/ICR1* methylation detectable in peripheral blood in patients who had tumors with 11p15.5 LOI compared to those with retention of imprinting (Fig. 3a). Of note, this statistically significant gain of methylation was within the "normal" range below the *H19/ICR1* β value of 0.7 associated with established definitions for germline 11p15.5 LOI. In contrast, 7 of the adjacent non-diseased kidney samples from these COG cohort patients exhibited 11p15.5 LOI with a *H19/ICR1* β value > 0.7 (Fig. 3b).

To further explore the finding of increased 11p15.5 *H19/ICR1* methylation found in peripheral blood of BWT patients and to determine if this was different from healthy community controls and unilateral WT patients, we combined identically processed and normalized *H19/ICR1* methylation β values from leukocyte-derived DNA between our current BWT cohort (n = 61) and a cohort of healthy community controls (n = 282) and WT cancer long-term survivors (including survivors of both unilateral [n = 154] and BWT [n = 17]) from

the St. Jude Life Cohort Study[37]. For this exploratory analysis, we defined low-level gain of methylation as an *H19/ICR1* β value > two standard deviations of the mean *H19/ICR1* β value detected in the healthy community control cohort. We hypothesized that *H19/ICR1* values in peripheral blood DNA would be higher in BWT patients than healthy community controls and potentially unilateral WT patients. The peripheral blood 11p15.5 *H19/ICR1* and *KCNQ1OT1/ICR2* β values were determined to be normally distributed in all groups using the Kolmogorov-Smirnov test. The mean *H19/ICR1* peripheral blood methylation β value in the healthy community control cohort was 0.499 ± 0.0248 (mean ± standard deviation). The mean *H19/ICR1* peripheral blood methylation β value from the BWT cohort (0.534 ± 0.0298) was higher than both the unilateral WT (0.521 ± 0.0258; *p* < 0.0001) and healthy community control cohorts (*p* < 0.0001; Fig. 3D.) We determined that 26 BWT patients (n = 20 from current study; n = 6 from St. Jude Life cohort) had a *H19/ICR1* methylation β value greater than two standard deviations from the mean community control cohort β value (i.e., *H19/ICR1* β > 0.54864; Fig. 2). Of these 20 patients from the current study with low level gain of methylation at *H19/ICR1*, 17/20 (85%) were female, 18/20 (90%) had 11p15.5 LOI in their tumor, 2/20 (10%) had 11p15.5 LOH in their tumor, and 0 had 11p15.5 ROI in their tumor (Fig. 2).

**11p15.5 status and genome-wide methylation/molecular identity in BWT**

We performed unsupervised hierarchical clustering using methylation M values from the top 10,000 most variable probes in the 850 K methylationEPIC dataset from leukocyte-derived DNA, adjacent non-diseased kidney DNA, BWT specimens, and unilateral WT specimens. To include unilateral WT specimens in this analysis for comparison, we combined the data set from the current study with methylation data from our previous analysis of WT primary tumors and corresponding xenografts[38]. We noted that BWT predominantly clustered distinctly from unilateral WT and closer to non-diseased adjacent kidney tissue, consistent with previously published results (Fig. 4)[39]. Within the BWT cluster, subgroups of tumors found to have 11p15.5 LOH, LOI, and ROI clustered together. To validate this result using a different computational method, we performed TSNE clustering of 850 K methylation EPIC data, which also showed BWT clustered distinctly from unilateral WT and closer to adjacent non-diseased kidney tissue (Fig. 5).

Among BWT, clustered subgroups of 11p15.5 LOH, LOI, and ROI emerged in TSNE analysis. Paired synchronous BWT with 11p15.5 LOI had the greatest inter-tumor variability in clustering pattern, and this result was confirmed using a Spearman correlation matrix (Fig. 5a, b, Supplementary Fig. 8). In contrast, paired synchronous BWT samples clustered less robustly by unsupervised hierarchical clustering and TNSE clustering of total-strand RNA-seq data (Supplementary Fig. 9–10). These data suggest that 11p15.5 status serves as a biomarker for global methylation/molecular subgroups in BWT and that BWT have distinct methylation patterns from unilateral WT, with BWT being more similar in global methylation to non-diseased kidney tissue. RNA-seq data were used to compare differentially expressed genes between unilateral WT and BWT in this study (Supplementary Table 1).

To further explore the suggestion that tumor 11p15.5 status correlates with a broader tumor molecular identity rather than only changes at the 11p15.5 locus itself, we determined genome-wide alterations in allele-specific expression stratified by tumor 11p5.5 status (LOI, LOH, ROI) in the COG cohort using the Cis-X method, which integrates whole genome sequencing and total-strand RNA-seq data to determine the allele of origin for the RNA being expressed[40]. Due to sample size limitations, we filtered results according to an unadjusted $p$ value of <0.05 and a false-discovery rate (FDR; p-value corrected for multiple testing) of <0.25 for this analysis. As predicted, *KCNQ1OT1* (FDR $p = 0.0046$) and *KCNQ1* (FDR $p = 0.0046$) were found to have perturbation of allele-specific expression in tumor samples with 11p15.5 LOH, but not ROI or LOI, while *INS-IGF2* and *IGF2* showed perturbation of allele-specific expression in both samples with 11p15.5 LOH and LOI. Five genes and/or noncoding RNAs outside the 11p15.5 region were found to have disruption in the normal patterning of allele-specific expression that correlated with tumor 11p15.5 status: *RNF185*, *SNORD116-4*, *C1QL3*, *CLEC12A*, and *NPAS2*. *RNF185*, a gene located at chromosome 22q12.2 that codes for a component of the E3 ubiquitin ligase

pathway, was found to have reduction to monoallelic expression in 43.3% of tumor samples with 11p15.5 LOI (FDR $p = 0.240$) and 35.7% of tumor samples with 11p15.5 LOH (FDR $p = 0.041$), but 0% of samples with 11p15.5 ROI (Supplementary Table 2).

We then queried for possible mechanisms of *RNF185* reduction to monoallelic expression and noted recurrent chromosome 22q loss correlated with this phenomenon. Combined with the copy number data detailed above that showed recurrent chromosome 22q loss in our data set and shared chromosome 22q loss in 3 synchronous BWT pairs, we performed a focused analysis to determine the clinical and other molecular features of BWT samples with chromosome 22q loss. We found an enrichment for female biological sex, 11p15.5 LOI, and *DGCR8* RNA microprocessor pathogenic variants in this cohort of 11 tumor samples with chromosome 22q loss from 8 patients (Supplementary Table 3).

## Discussion

This study determined the landscape of predisposition for bilateral Wilms tumor. These data demonstrate two predominant modes of BWT susceptibility: (1) Pre-zygotic germline genetic variants readily detectable in DNA derived from peripheral blood (*WT1*, *NYNRIN*, *TRIM28*, BRCA complex genes) often followed by 11p15.5 LOH or (2) Post-zygotic epigenetic hypermethylation at the 11p15.5 *H19/ICR1* locus (LOI; Fig. 6) that may require analysis of multiple tissue types for definitive diagnosis. Multiple lines of evidence in this study discussed below support 11p15.5 *H19/ICR1* hypermethylation being a post-zygotic, somatic mosaic mesodermal epigenetic event that is common among patients who develop BWT. Therefore, we found that BWT predisposition in the absence of a predisposing germline variant is a manifestation of the Beckwith Wiedemann spectrum often without overt additional clinical features. Furthermore, this study demonstrates that 11p15.5 tumor status can be used as a surrogate biomarker for more global methylation patterns or molecular subgroups of BWT.

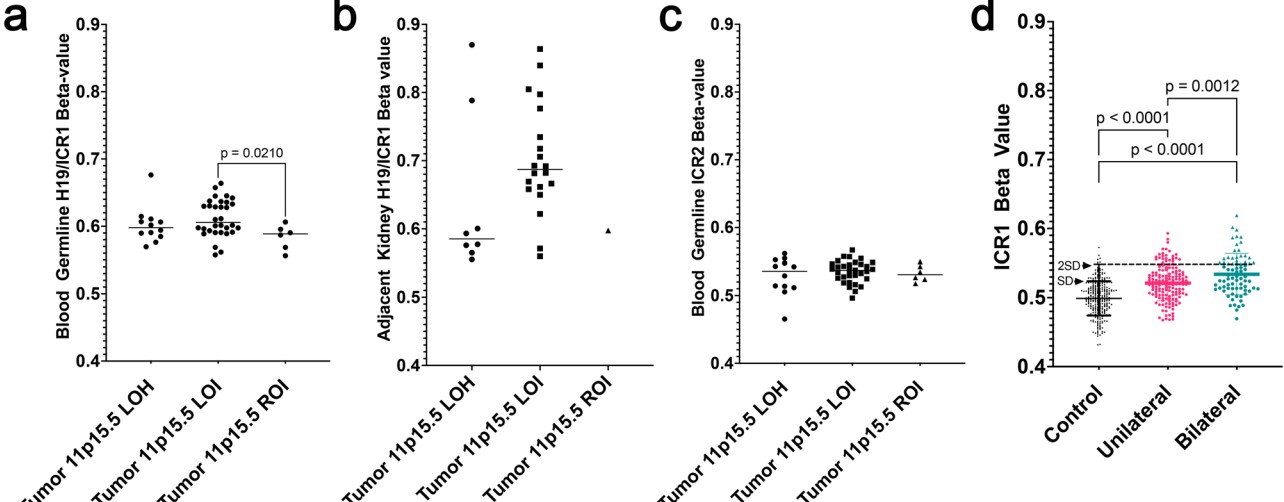

**Fig. 3 | Exploratory analysis of low-level *H19/ICR1* hypermethylation detectable in peripheral blood. a** Patients from the current study COG cohort with BWT containing 11p15.5 LOI were found to have a statistically significant increase in 11p15.5 *H19/ICR1* methylation detected in the peripheral blood when compared to patients with tumors having 11p15.5 retention of imprinting (unpaired, two-tailed $t$ test $p = 0.0210$; $n = 50$ biologically independent samples). **b** 11p15.5 LOI was often detectable above the threshold value (β > 0.7) for loss of imprinting in adjacent non-diseased kidney tissues ($n = 29$ biologically independent samples). **c** No differences were detected among patient groups at 11p15.5 *KCNQ1OT1/ICR2* in peripheral blood ($n = 50$ biologically independent samples). Of note, the single sample outlier with tumor 11p15.5 LOH, hypermethylation at *H19/ICR1*, and hypomethylation at *KCNQ1OT1/ICR2* detected in the peripheral blood was confirmed to have mosaicism

for 11p15.5 LOH detected in peripheral blood by whole genome sequencing. For (**a**–**c**), lines represent median values. **d** A significant increase in *H19/ICR1* methylation detected in peripheral blood was noted when BWT patients ($n = 78$) were compared to unilateral WT ($n = 154$ biologically independent samples) and healthy community control subjects ($n = 282$ biologically independent samples; $p$ values are ordinary one-way ANOVA with pairwise values corrected for multiple comparisons). 26 BWT patient blood samples (green triangles; $n = 20$ biologically independent from current study, $n = 6$ from survivor cohort) had low-level *H19/ICR1* gain of methylation defined as a β value greater than two standard deviations (2 SD) above the mean from the healthy community control cohort. For boxes in (**d**), measure of center lines are mean and whiskers are SD. LOH loss of heterozygosity, LOI loss of imprinting. Source data are available in the Source Data File.

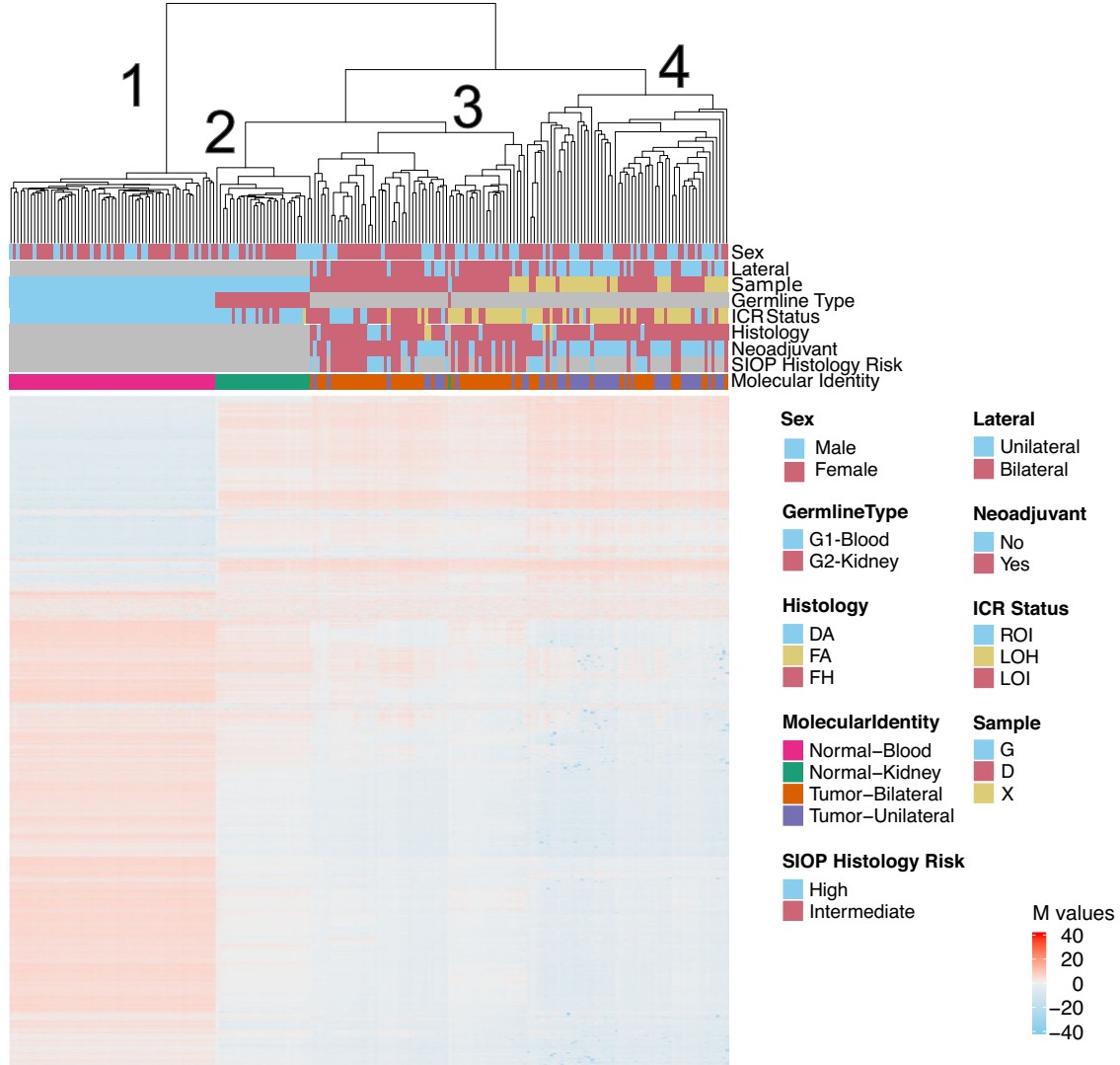

**Fig. 4 | Unsupervised hierarchical clustering of methylation M-values from the top 10,000 most variable probes in the 850 K EPIC Methylation Beadchip array.** Distinct clustering of DNA samples derived from peripheral blood (cluster 1), adjacent non-diseased kidney (cluster 2), a cluster of predominantly BWT (cluster 3), and a cluster of predominantly unilateral WT (cluster 4). Within the BWT cluster (cluster 3), samples with 11p15.5 LOH, 11p15.5 LOI, and 11p15.5 ROI cluster together. Notably, the BWT cluster (cluster 3) joins with adjacent non-diseased kidney (cluster 2) more closely than the unilateral WT cluster. $N = 212$ biologically independent samples. Histology – DA – diffuse anaplasia, FA focal anaplasia, FH favorable histology. ICR status: ROI retention of imprinting, LOH loss of heterozygosity, LOI loss of imprinting. Tumor type: G – germline sample, D – primary tumor sample, X – patient-derived xenograft. Source data are available in the Source Data File.

WT1 variants were the most common germline genetic variants associated with BWT predisposition in this cohort. In contrast to WT1 variants of purely somatic origin, which are often seen in unilateral WT, all patients with tumor WT1 alterations in the current study were found to have a blood germline variant in WT1[22]. Scott et al. and Huff et al. similarly found germline WT1 variants in BWT specimens containing WT1 alterations[41,42]. Although all tumor samples from patients with pathogenic WT1 germline variants in the current study exhibited 11p15.5 LOH, prior work from our group and others in unilateral Wilms tumor samples demonstrated somatic WT1 pathogenic variants without accompanied 11p15.5 LOH; therefore WT1 pathogenic variants and 11p15.5 LOH often, but not always, accompany one another[38,43]. These data demonstrate a strong association between WT1 germline variants and somatic 11p15.5 copy neutral LOH events (paternal uniparental disomy) that encompass both the 11p15.5 and WT1/11p13 loci, resulting in biallelic expression of IGF2 and biallelic inactivation of WT1 (Fig. 6). Therefore, it can be inferred that the germline genetic WT1 variants that lead to BWT predisposition occur on the paternal allele and

become homozygous in the tumor due to copy neutral LOH events at 11p13-11p15.5 loci that establish paternal uniparental disomy. The exact breakpoints of 11p15.5 LOH are different in paired synchronous BWT, demonstrating convergent tumor evolution via independent genetic events in each tumor specimen, consistent with the study by Valind et al.[44].

Two synchronous BWT sample pairs in this study (SJWLM066776 and SJWLM069396) from patients with inactivating pathogenic blood germline WT1 variants demonstrated a small number of shared non-coding tumor somatic variants in addition to tumor 11p15.5 LOH (Table 2). This constellation of findings implies 11p15.5 LOH can occur in the early embryo and may be required ahead of malignant transformation on the background of WT1 pathogenic variants. The sequence of pathogenic germline WT1 variants and subsequent somatic 11p15.5 LOH is often followed by development of activating tumor CTNNB1 variants, which were the most common somatic variants found in the current study. Our results are consistent with the temporal sequence first reported by Fukuzawa et al. who

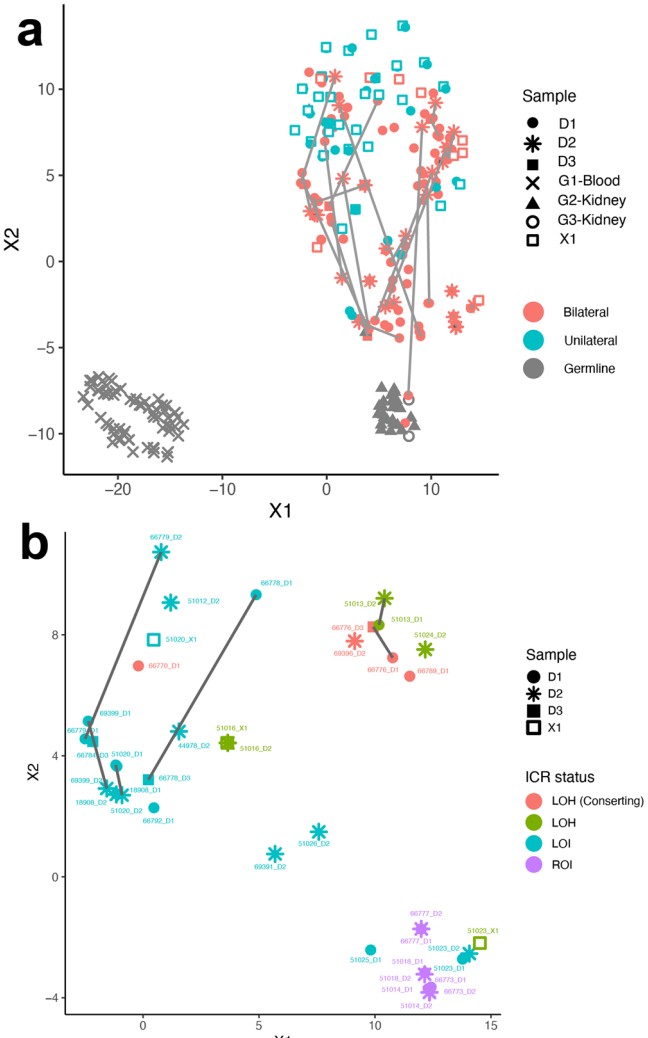

**Fig. 5 | TSNE clustering analysis of 850 K EPIC Methylation Beadchip array.**
**a** Predominant clusters of unilateral (teal) and bilateral (salmon) WT. BWT cluster closer to adjacent non-diseased kidney (gray triangles) than unilateral WT. Gray lines connect synchronous BWT. However, large differences in synchronous BWT correlated with differential tumor purity in each DNA sample. N = 267 biologically independent samples. **b** Clustering of BWT with a tumor purity filter applied (excluding specimens with tumor purity <80%) shows near adjacent clustering of paired synchronous BWT specimens. Within BWT, samples cluster according to 11p15.5 status: 11p15.5 LOI (teal), 11p15.5 LOH (red and green), and 11p15.5 ROI (purple). The greatest difference between paired synchronous BWT samples is seen in samples with 11p15.5 LOI (samples connected by gray lines). N = 40 biologically independent samples. G1 blood-derived germline DNA, G2 and G3 adjacent non-diseased kidney derived DNA, D1-D3 tumor derived DNA, X1 patient-derived xenograft DNA, ICR status – LOH – loss of heterozygosity, LOI loss of imprinting, ROI retention of imprinting. Source Data are available in the Source Data File.

demonstrated that *WT1* variants were detected in both nephrogenic rests and WT, but *CTNNB1* variants were only found in the adjacent WT[45]. *WT1* and *CTNNB1* variants often co-occur in WT and the spatial distribution of *CTNNB1* variants has been demonstrated to exhibit intra-tumor genetic heterogeneity[22,46]. WT driven by *WT1* variants are known to exhibit stromal/rhabdomyoblastic differentiation and poor volumetric regression in response to neoadjuvant chemotherapy[47,48]. Therefore, future knowledge of germline *WT1* status at diagnosis in patients with BWT could guide expectations regarding volumetric tumor regression and timing of surgical resection. Taken together, these data and a recent study by Hol. et al that demonstrated a much higher than predicted incidence of germline *WT1* variants in

unselected WT patients (either unilateral or bilateral, often without syndromic features), support expanded germline genetic testing at diagnosis for all patients with BWT[20]. It should also be noted that tumors from patients with pre-zygotic variants in Wilms tumor or cancer predisposition genes can also exhibit 11p15.5 LOI, but this is more rare than 11p15.5 LOH or ROI in this group.

In contrast, patients without blood germline genetic variants in WT or cancer predisposition genes were frequently found to have 11p15.5 *H19/ICR1* hypermethylation (LOI) in their tumors and adjacent non-diseased kidney tissue. We hypothesized that 11p15.5 LOI (*H19/ICR1* gain of methylation) was a somatic mosaic epigenetic event that occurred in the early embryo in a mesodermal progenitor cell during the time when genomic imprints are being established and led to BWT predisposition. Genomic DNA methylation, which is the primary responsible mechanism for imprinting, reaches its final level near the time of embryonic gastrulation (i.e., formation of the three germ layers – ectoderm, endoderm, mesoderm)[49]. This developmental timepoint is precisely when the mesodermal cells that give rise to the right and left intermediate mesoderm/kidney primordia are spatially isolated as they invaginate to establish the mesoderm and migrate to the right or left lateral sides of the embryo[50]. Multiple lines of evidence in this study support this mesodermal somatic mosaic hypothesis. First, 11p15.5 LOI is shared in many synchronous BWT, while other downstream common somatic genetic alterations (exact variants in *CTNNB1*, microRNA processing genes, *SIX1/2*, *TP53*, etc.) are not. Next, synchronous BWT with shared 11p15.5 LOI also exhibit a small number of shared noncoding variants relative to the total number of noncoding variants demonstrated in each paired tumor, consistent with shared clonal origin in the early embryo with subsequent divergence and independent evolution. Finally, the frequent detection of 11p15.5 *H19/ICR1* LOI in adjacent non-diseased kidney tissue and associated tumors but not blood is consistent with mosaicism throughout the body.

Okamoto et al. originally described mosaicism for 11p15.5 LOI in mesodermal tissues of patients with WT, including adjacent non-diseased kidney tissue of 8/8 patients with tumor 11p15.5 LOI in their study[51]. In addition, one of these eight patients (who had a clinically apparent BWSp phenotype) was found to have increased *H19* methylation detectable in peripheral blood[51]. Our current results build on these findings and the recent detailed analyses performed on a focused set of BWT specimens by Coorens et al by establishing the relative frequency of 11p15.5 LOI as a mode of predisposition for BWT[23]. Coorens et al. showed that 11p15.5 LOI likely occurs as a post-zygotic event that results in somatic mosaicism for *H19/ICR1* hypermethylation. Like our results, their analysis showed shared noncoding variants and associated hypermethylation of *H19/ICR1* in specimens from BWT patients. Also in their study, the *H19/ICR1* hypermethylation was detected in adjacent non-diseased kidney tissue and associated with clonal expansion of histologically normal renal cells deemed "clonal nephrogenesis." This clonal expansion of nephrogenic cells is thought to occur during kidney development and provides the precursor cell population for multifocal and BWT development. Our detection of 11p15.5 LOI or LOH in 9/29 (31%) samples of adjacent histologically non-diseased kidney tissue is also consistent with the concept of clonal nephrogenesis.

Coorens et al. concluded that somatic mosaic *H19/ICR1* hypermethylation is detectable in adjacent non-diseased kidney, but not in peripheral blood. Overall, our results are consistent with these findings because no peripheral blood sample in our cohort of BWT patients was found to have 11p15.5 LOI defined as an *H19/ICR1* β value > 0.7. Patients with 11p15.5 LOI detected in tumor and adjacent non-diseased kidney, but not in peripheral blood can be presumed to be mosaic for 11p15.5 LOI. However, we believe it is likely there are additional patients with low-level mosaicism for 11p15.5 LOI in the kidney and blood that does not achieve the thresholds used in this study.

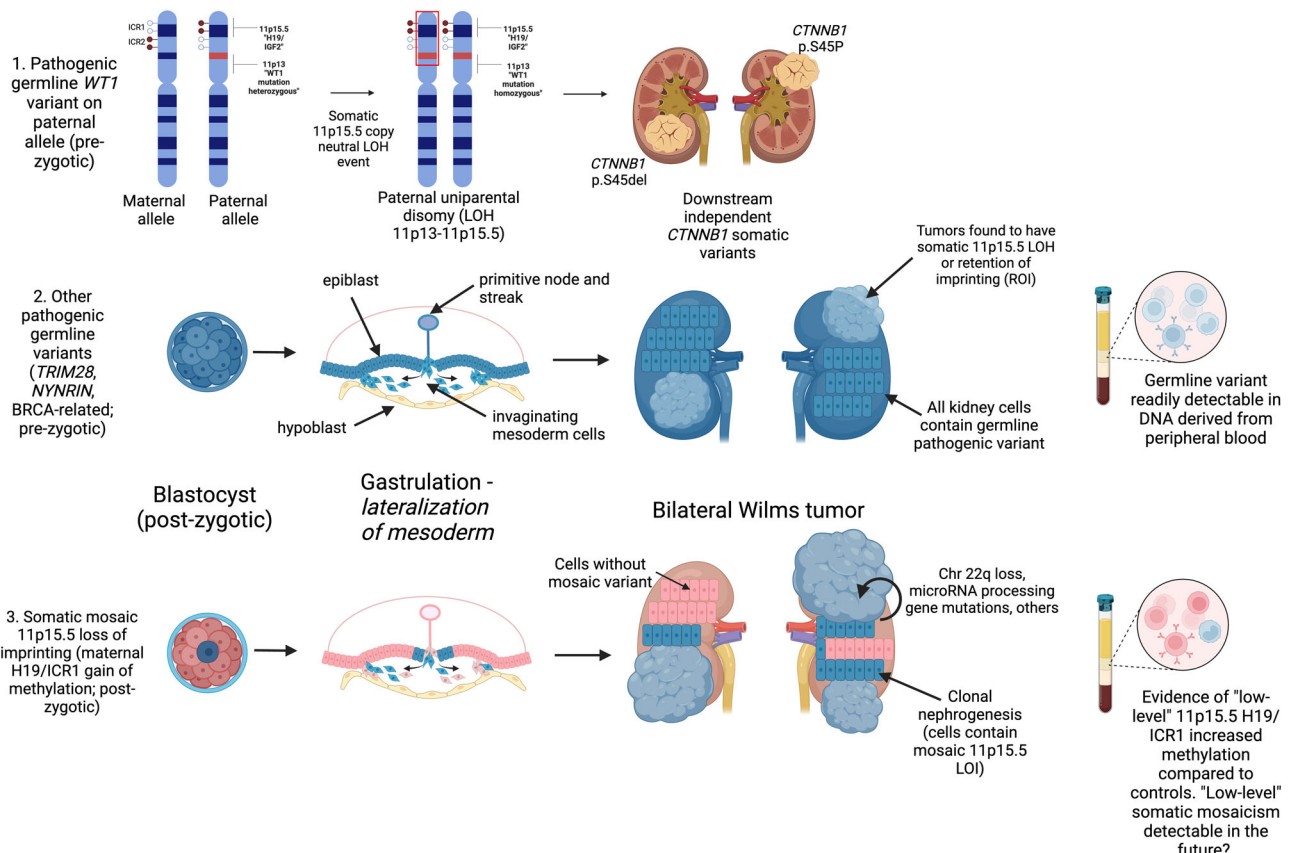

**Fig. 6 | Summary of Proposed Mechanisms for Bilateral Wilms tumor development.** The top panel depicts the molecular sequence leading to BWT in patients with pathogenic heterozygous *WT1* germline variants. Here, somatic 11p15.5 copy neutral loss of heterozygosity causes biallelic inactivation of *WT1* and biallelic expression of *IGF2*, a sequence which is often followed by downstream distinct *CTNNB1* somatic variants unique to each tumor. The middle panel depicts the general sequence of BWT development due to a pre-zygotic pathogenic germline variant in which the germline variant is present in all kidney cells. The bottom panel depicts somatic mosaic 11p15.5 loss of imprinting, in which 11p15.5 H19/ICR1 gain of methylation occurs on the maternal allele in a post-zygotic embryonic cell. This event must occur prior to lateralization of cells fated to become mesoderm during embryonic gastrulation. At this time in embryonic development, the cells that give rise to the intermediate mesoderm and therefore the kidneys are anatomically sequestered from one another (right and left). This lateralization results in a somatic mosaic distribution of 11p15.5 LOI throughout the body/mesoderm. Expansion of cellular clones containing the somatic mosaic alteration is termed clonal nephrogenesis and explains the detection of 11p15.5 LOI in adjacent non-diseased kidney tissue. BWT and/or multifocal WT arise from these clonal populations of kidney cells following additional somatic events including LOH of chromosome 22q, microRNA processing gene pathogenic variants, and others. LOH loss of heterozygosity, LOI loss of imprinting. Graphic made with biorender.com.

We reasoned that, because lymphocytes are derived from the embryonic mesodermal cell population, there could be evidence of low-level 11p15.5 *H19/ICR1* increased methylation detectable in peripheral blood. To further explore this concept, we defined low-level gain of methylation as an *H19/ICR1* methylation β value greater than two standard deviations above the mean *H19/ICR1* β value of a healthy control cohort. Our exploratory analysis of 11p15.5 *H19/ICR1* methylation in peripheral blood demonstrated that BWT patients with 11p15.5 LOI in their tumors had significantly higher peripheral blood *H19/ICR1* methylation values than patients with tumors having normal 11p15.5 retention of imprinting. We then hypothesized and subsequently determined that patients and survivors of BWT had higher levels of *H19/ICR1* methylation detectable in peripheral blood than a cohort of healthy community control subjects and unilateral WT patients. Twenty patients (90% of whom were female) from the current study were found to exhibit low-level gain of methylation at 11p15.5 *H19/ICR1*. While these statistically significant increases in *H19/ICR1* methylation in BWT patients still fall within the "normal" range, they provide evidence that a mosaic, minor population of lymphocytes with 11p15.5 *H19/ICR1* LOI may exist. These findings are unlikely to be related to circulating tumor DNA because the result was confirmed in BWT long-term survivors, whose blood DNA samples were obtained at least 5 years after cancer diagnosis.

Fiala et al. also previously demonstrated low-level gain of methylation detectable in peripheral blood samples from 7 female patients with BWT, which they defined as two standard deviations above the mean *H19/ICR1* methylation level in a healthy control cohort using methylation-sensitive PCR[24]. These findings demonstrate that a normal *H19/ICR1* methylation value from a blood sample is insufficient to exclude mosaic 11p15.5 LOI in a patient with BWT, and non-diseased kidney and tumor samples may be required for definitive evaluation. Because 11p15.5 *H19/ICR1* LOI is a purely epigenetic phenomenon, refinements in single-cell methylation or alternative approaches will be required to definitively evaluate an individual blood sample for evidence of low-level 11p15.5 LOI in the future. For 11p15.5 LOH, which is a genetic change caused by copy neutral loss of heterozygosity, we were able to confirm mosaicism detectable in a peripheral blood sample from a single patient using whole genome sequencing data (SJWLM069390).

Furthermore, our data show that 11p15.5 status can be used as a surrogate biomarker for more global methylation patterns/molecular subgroups of BWT because when samples were clustered according to broader, genome-wide patterns of methylation, they tended to cluster according to 11p15.5 status. Whether 11p15.5 status (ROI, LOI, LOH) correlates with volumetric or histologic response to neoadjuvant

chemotherapy, event-free, or overall survival will be the subject of future clinical translational investigation. However, as a preliminary window into this question, the current study showed that 15/17 (88.2%) specimens with SIOP high-risk post-treatment histology and 15/17 (88.2%) with the adverse prognostic biomarker 1q gain had 11p15.5 LOI. The current study provides strong rationale for prospectively following outcomes in BWT patients according to tumor 11p15.5 status and this will be included as an observational biologic aim in the Children's Oncology Group BWT protocol currently under development.

Specifically, our study showed that BWT with 11p15.5 LOI and LOH commonly exhibited reduction to monoallelic expression of the ubiquitin-ligase associated gene *RNF185* located at chromosome 22q12.2. The most common mechanism for *RNF185* reduction to monoallelic expression was chromosome 22q copy loss. All tumors that exhibited copy loss at 22q had also had 11p15.5 LOI and 6 of these 8 patients were female. This group also included all 4 patients with *DGCR8* microprocessor hotspot p.E518K pathogenic variants in the current study and included 5 patients with high-risk SIOP post-treatment histology. Allelic loss at chromosome 22q has been previously associated with high-risk WT[52,53]. Furthermore, in depth analysis of two cases of paired WT and precursor nephrogenic rests has implicated loss of chromosome 22 in the progression from perilobar nephrogenic rests to WT[54]. *DGCR8* (located at chromosome 22q11.21) pathogenic variants were shown to have an extreme female predominance in previous WT studies: Wegert et al. found that 23/26 (88.5%) and Walz et al. found that 15/17 (88.2%) patients with somatic *DGCR8* pathogenic variants were female[33,55]. The possible functional significance of *RNF185* in WT and the constellation of somatic mosaic 11p15.5 LOI, female biological sex, chromosome 22q loss, and *DGCR8* microprocessor pathogenic variants in the development and progression of WT will be the subjects of future investigation.

This study has limitations. The study was designed to maximize the number of eligible paired synchronous BWT specimens. Thus, not all tumors were treated according to the same protocol, some tumors samples were obtained as pre-treatment biopsies, and some samples were obtained after neoadjuvant therapy was administered at the time of surgical resection. Still, the number of paired synchronous BWT specimens was limited by inconsistent sample collection, insufficient tumor purity, availability, and heterogeneous treatment response which could have caused some tumors to be ineligible for inclusion due to necrosis, stroma, or tumor content thresholds. Our results show that tumor purity can affect measurement of methylation at 11p15.5 *H19/ICR1* and ICR2; however, variations in tumor purity in pre-treated BWT will always be present due to the heterogeneity discussed above. Because tumor purity correlated positively with *H19/ICR1* methylation and negatively with ICR2 methylation, samples of perfect purity would be expected to exhibit increased likelihood of 11p15.5 LOI or LOH; therefore, our results could underestimate the number of tumor samples with these findings. Furthermore, the study does not account for intra-tumor genetic heterogeneity known to be present in WT since only a single sample was included from each tumor. Therefore, prospective collection of BWT samples treated under a uniform protocol will be needed for clinical translational aims such as the association between biologic subgroups and treatment response to neoadjuvant therapy or long-term oncologic outcomes. Family history information was not available from the COG cohort and thus we could not determine if the patients who had germline variants in *BRCA1*, *BRCA2*, or *PALB2* had a family history of breast or ovarian cancer. The clinical evaluation and reported data about syndromic features may be incomplete, especially for patients who may have exhibited a subtle clinical phenotype. Finally, all data in this manuscript reporting evidence of mosaic 11p15.5 *H19/ICR1* gain of methylation detectable in peripheral blood are based on statistical comparison or observed changes in comparison to a normal cohort, while no patients in this manuscript exhibited hypermethylation above the established

threshold of a β value of 0.7 at 11p15.5 *H19/ICR1* detectable in DNA obtained from peripheral blood.

In conclusion, this study shows that the predisposition for BWT occurs primarily due to pre-zygotic germline genetic variants or post-zygotic 11p15.5 LOI (*H19/ICR1* hypermethylation). These findings underscore the rationale for an (epi)genotype-based approach in which molecular diagnostics can be used to subgroup BWT patients according to germline variants and/or 11p15.5 copy number and/or epigenetic alterations to determine how these modes of predisposition correlate with volumetric or histologic tumor response and long-term oncologic outcomes. These findings also suggest that in-depth (epi)genetic testing including genetic sequencing and methylation analysis of peripheral blood, adjacent kidney (when available), and tumor may be required to diagnose the means of predisposition to BWT.

## Methods

### Sample acquisition
This study, including 99 tumor samples from 68 patients with a diagnosis of synchronous BWT ($n = 18$ SJCRH, $n = 50$ COG), was approved by the SJCRH institutional review board (IRB# Pro00007515) and approved by the COG as study AREN18B5-Q. Written informed consent for future research was obtained from all patient parents or legal guardians prior to biospecimen collection. Participants were not compensated for participation in this study. Prior to nucleic acid isolation, sections corresponding to all frozen biospecimens were reviewed by a pathologist and determined to have greater than 50% viable tumor. Adjacent non-diseased kidney tissue was confirmed to contain histologically normal kidney and to be tumor-free. We first performed analysis on the cohort of SJCRH BWT specimens from 18 patients. Using preliminary data from this SJCRH cohort analysis, we applied for additional specimens from the COG to establish an additional cohort. Of note, after determining that somatic variants were almost never shared among synchronous BWT in the SJCRCH cohort using whole exome sequencing, we switched to whole genome sequencing of the COG samples ($n = 50$ patients) to determine if variants outside coding regions were shared among synchronous BWT.

Genomic DNA and total-RNA were isolated by the SJCRH Biorepository or the COG Biopathology Center. Qiagen DNA extraction kits were used for DNA isolation and Trizol for RNA isolation. DNA was quantified using PicoGreen and visualized in agarose gel for quality control. RNA was quantified using Qubit fluorometry assay and quality and integrity were evaluated using RNA integrity number (RIN) measurements performed using an Agilent Bioanalyzer System.

Tumor RNA was used for total strand RNA-seq and tumor DNA for methylation analysis using the 850 K methylationEPIC beadchip array (Illumina). Tumor DNA with available matched germline DNA from peripheral blood lymphocytes was used for whole exome (SJCRH) or whole genome sequencing (COG). Germline DNA derived from peripheral blood lymphocytes and DNA derived from adjacent histologically non-diseased kidney was used for whole exome (SJCRH) or whole genome sequencing (COG), and methylation analysis. A detailed account of specimens and sequencing or array modalities utilized in this study is shown in Fig. 1.

### Clinical data
Clinically relevant details were obtained from each BWT case including patient age at diagnosis, biological sex, tumor laterality, associated congenital anomalies or syndromes, neoadjuvant chemotherapy received, tumor histology, SIOP (Societe Internationale D'oncologie Pediatrique) post-treatment pathology risk stratification, and presence of nephrogenic rests[56]. For the St. Jude cohort, syndromes were ascertained by thorough review of the medical record and all molecular testing documentation. For the COG cohort, the presence of genetic syndromes was a collected/reported data point from the

originating studies and was delivered to the investigators after molecular analysis was complete.

## Whole exome and whole genome sequencing

Whole exome sequencing (SJCRH) or whole genome sequencing (COG) were performed on BWT DNA with available paired germline DNA from peripheral blood ($n = 61$ patients; $n = 87$ total tumor samples; Fig. 1). For variant discovery, a paired analysis was performed comparing tumor-derived DNA to germline DNA obtained from peripheral blood leukocytes. Single nucleotide variants, insertion/deletion/frameshift, and noncoding variants calls were made as previously described[38]. To analyze the pathways affected by tumor somatic variants in BWT we used the functional annotation tool DAVID to generate a set of enriched pathways combining KEGG, Reactome, and Wikipathways[57]. Using whole exome sequencing or whole genome sequencing data, a tumor copy number variant (CNV) analysis was performed using a threshold of CNV $\geq 0.5$ or $\leq -0.5$ for full copy number gain or loss at a given chromosomal locus, respectively. Low-level tumor copy number gain or loss was defined as CNV $\geq 0.1$ and $\leq 0.5$ or CNV $\leq -0.1$ and $\geq -0.5$ respectively. Areas of tumor copy neutral loss of heterozygosity (cnLOH), loss of heterozygosity (LOH) due to copy loss, or copy number gain were ascertained using the CONSERTING algorithm[34].

## Total-strand RNA-seq

Total-strand RNA-seq was performed on all BWT samples in the study ($n = 99$); RNA-seq was not performed on blood or adjacent non-diseased kidney. Total RNA-seq library preparation, sequencing, read mapping, and generation of gene level read counts and Fragments per kilobase million (FPKM) values were generated as previously described[38]. Integrated analysis of whole genome sequencing and total-strand RNA-seq data was performed using the previously described Cis-X method to determine genome wide allele-specific expression patterns in BWT[40].

## Methylation analysis

Genomic tumor, non-diseased kidney, and blood germline DNA were bisulfite converted using the EZ DNA Methylation kit (Zymo Research Corp). Converted samples were processed and hybridized to the Infinium MethylationEPIC Beadchip (850 K) array (Illumina) according to the manufacturer's instructions. Raw IDAT files containing summarized information from the beadchip array were pre-processed using subset-quantile within array normalization (SWAN) function as previously described[58]. The methylation score of each CpG site in the array is represented as a beta ($\beta$) value (methylated signal/methylated +unmethylated signals) and was computed using the R package minfi[59]. Methylation M values (log2 ratio of the intensity of methylated signal/unmethylated signal) were also computed using the R package minfi and used for EPIC-based differential methylation analyses including unsupervised hierarchical clustering, TSNE (t-distributed stochastic neighbor embedding), and Spearman correlation matrix analyses[59,60]. Tumor purity was estimated from the methylation array data using a deconvolution-based approach as previously described[35]. These purity estimates were validated by comparison to estimates derived from whole genome sequencing data in the COG specimens.

The imprinting status at the chromosome 11p15.5 locus was determined using methylation data as previously described[43]. Briefly, the average $\beta$ value for *H19/ICR1* was calculated using CpG probes located within the chr11:2,019,974-2,024,738 (GRCh38/hg38) range and the average $\beta$ value for 11p15.5 *KCNQ1OT1/ICR2* was calculated using probes located within the chr11:2,721,228-2,722,228 (GRCh38/hg38) range. Samples with average $\beta$ value *H19/ICR1* < 0.7 and *KCNQ1OT1/ICR2* > 0.3 were determined to have normal retention of imprinting (ROI), samples with *H19/ICR1* > 0.7 (hypermethylation) and *KCNQ1OT1/ICR2* > 0.3 were determined to have loss of imprinting (LOI),

at *H19/ICR1*, and samples with *H19/ICR1* > 0.7 (hypermethylation) and *KCNQ1OT1/ICR2* < 0.3 (hypomethylation) were determined to have loss of heterozygosity (LOH) at 11p15.5. LOH at 11p15.5 was designated if samples had LOH or partial LOH detectable using the CONSERTING algorithm from whole genome sequencing data[34]. For the exploratory analysis of *H19/ICR1* methylation in peripheral blood, low-level gain of methylation at 11p15.5 *H19/ICR1* was defined as a $\beta$ value greater than two standard deviations above the mean *H19/ICR1* $\beta$ value of a healthy control cohort.

## Unsupervised hierarchical clustering, TSNE clustering, Spearman Correlation Matrix

Unsupervised hierarchical clustering was performed using methylation M values from the 850 K MethylationEPIC beadchip array data set using all samples from the current BWT data set and BWT and unilateral samples from our prior WT xenograft analysis[38,60]. The top 10,000 most variable probes in the data set were used for clustering analysis. Probes located on the X and Y chromosomes were excluded to reduce biological sex bias. Total-strand RNA-seq LogTPM values were similarly used to perform unsupervised hierarchical clustering and TSNE clustering analysis using tumor-derived RNA.

## Germline genomic analysis from peripheral blood

Germline genetic variants were queried for all patients with an available leukocyte-derived DNA sample from peripheral blood (total $n = 61$; SJCRH $n = 11$, COG $n = 50$). Analysis was performed to query for single nucleotide substitution, nonsense, and insertion/deletion variants in 565 previously described cancer-related genes which specifically include the WT predisposition genes *DICER1, IGF2, TP53, WT1, ASXL1, BRCA2, CDC73, FBXW7, PIK3CA, BLM, BUB1B, CTR9, DIS3L2, GPC3, KDM3B, NYNRIN, PALB2, REST, TRIP13, TRIM28,* and *TRIM37*[27,61]. All insertion/deletion and nonsense variants were included in germline predisposition variant counts. The Clinvar database (https://www.ncbi.nlm.nih.gov/clinvar/) was queried to determine pathogenicity of single nucleotide variants[62]. Variants reported as benign in Clinvar were excluded and those reported as pathogenic or probably pathogenic were included in germline predisposition variant counts. Variants of uncertain significance (VUS), reported with no assertion, or unreported variants in Clinvar were further analyzed using the PROVEAN and PolyPhen2 algorithms[63–65]. Variants classified as deleterious by PROVEAN, possibly damaging or damaging by PolyPhen2 prediction score, were included in germline predisposition variant counts. Furthermore, unreported variants or VUS were included in germline predisposition variant counts if a significant increase in variant allele frequency in the tumor was identified compared to the germline tissue consistent with retention of mutated allele in the tumor (loss of heterozygosity).

## Long-term survivorship cohort analysis from peripheral blood DNA

Blood-derived Infinium MethylationEPIC Beadchip (850 K) array germline DNA methylation data from the 61 patients in the current study were combined with blood-derived germline DNA methylation data from 282 healthy community controls and 171 long-term Wilms tumor survivors (>5 years from cancer diagnosis; $n = 154$ unilateral, $n = 17$ bilateral) from the St. Jude Lifetime Cohort Study (SJLIFE)[37]. These data were normalized and processed with the subset-quantile within array normalization (SWAN) method using the R package minfi. The average $\beta$ values from the 11p15.5 *H19/ICR1* and *KCNQ1OT1/ICR2* regions defined above were computed as detailed above.

## Statistics and reproducibility

Samples initially received from two patients were excluded from the study due to mismatch detected in short tandem repeat profiles indicating kidney/tumor/blood samples were not from the same

patient (tumor/germline mismatch). No statistical method was used to predetermine sample size. The experiments were not randomized. Investigators were blinded from clinical details until molecular analysis was complete. Statistical tests are indicated in figure legends and were performed using R Software version v3.5.1 or 4.1.0 or Graphpad Prism v10 software. All p-values in the manuscript are two-sided and p < 0.05 was considered statistically significant.

## Reporting summary

Further information on research design is available in the Nature Portfolio Reporting Summary linked to this article.

## Data availability

The 850 K MethylationEPIC and RNA-seq data from the St. Jude and Children's Oncology Group bilateral Wilms tumor patients generated in this study are uploaded to the Gene Expression Omnibus (GEO) database (https://www.ncbi.nlm.nih.gov/geo/) under SuperSeries accession number GSE226480. The Whole exome sequencing and whole genome sequencing data generated in this study are available at the NCBI Sequence Read Archive (SRA) database under accession number PRJNA943166. 850 K MethylationEPIC data used from our prior publication[38] which are used for comparative analysis in the current study are available via the GEO database under accession number GSE110697. 850 K MethylationEPIC data from the Wilms tumor survivorship cohort and healthy community controls (Song, et al. Genome Med. 2021 Apr 6;13(1):53. https://doi.org/10.1186/s13073-021-00875-1) are available via the GEO database under accession numbers GSE197676, GSE197675, and GSE197674. Source Data are provided as a Source Data File. The remaining data are available within the Article, Supplementary Information, or Source Data File. Source data are provided with this paper.

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

## Acknowledgements

This manuscript was funded by the American Pediatric Surgical Association Foundation Jay Grosfeld Scholarship (A.J.M.), National Institutes of Health (NIH)/National Cancer Institute (NCI) Grants 1K08CA255569-01 (A.J.M.), 1L40CA242492-01 (A.J.M), and 1R01CA229739-01 (J.Y.), by NIH/NCI Comprehensive Cancer Center Support Grant 5P30CA021765-41 (St. Jude Children's Research Hospital), by American Cancer Society-Research Scholar Grant (130421-RSG-17-071-01-TBG, J.Y.), and by the American Lebanese and Syrian Associated Charities/St. Jude Children's Research Hospital. This work was additionally supported by the NCTN Operations Center Grant U10CA180886, the NCTN Statistics & Data Center Grant U10CA180899, and grants U10CA098543, U10CA098413, and U24CA114766 from the National Cancer Institute, National Institutes of Health, to support the Children's Oncology Group, and St. Baldrick's Foundation. The content is solely the responsibility of the authors and does not necessarily represent the official views of the National Institutes of Health. The authors would like to thank the Children's Oncology Group (COG) protocol coordinators, research coordinators, Clinical Research Assistants and other health professionals who contributed to acquiring samples used in this study. The authors in particular wish to thank the extended ARENO534 and AREN03B2 teams who contributed to central review and clinical annotation of these samples. The authors would also like to thank the patients and families who granted permission to participate in these studies.

## Author contributions

A.J.M., E.M.P, J.Z., G.P.Z., A.M.D., and X.C. conceived the project and generated hypotheses. A.J.M., E.M.P., V.H., J.S.D., A.L.H., E.A.M, P.F.E., C.V.F., L.A.R., and B.T. inventoried, organized, and acquired samples. A.J.M., E.M.P., H.M., J.E., A.M.F., M.R.C., and T.S. performed experiments. A.J.M., L.A.R., B.T., V.H., A.L.H., and Z.W. provided data. A.J.M., C.C., J.W., T.I.S., K.D-J., and X.C. performed data analysis. The initial manuscript draft was written by A.J.M, C.C., and X.C. All authors A.J.M, C.C., J.W., T.I.S.,

E.M.P., K. D-J., J.B., L.A.R., B.T., V.H., A.L.H, E.A.M, B.C., J.S.D, C.V.F., J.I.G, P.F.E., H.M., N.O., J.M., C.M.J., A.M.F., P.P., C.L.M., J.E., K.E.N., M.R.C., T.S., J.Z., J.Y., G.P.Z., Z.W., A.M.D, and X.C. performed data interpretation, discussion of results, and review and editing of the manuscript. A.J.M, G.P.Z, A.M.D, and X.C. performed project supervision and oversight.

## Competing interests

The authors declare no competing interests.

## Additional information

[1]Department of Surgery, St. Jude Children's Research Hospital, Memphis, TN 38105, USA. [2]Division of Pediatric Surgery, Department of Surgery, University of Tennessee Health Science Center, Memphis, TN 38105, USA. [3]Department of Computational Biology, St. Jude Children's Research Hospital, Memphis, TN 38105, USA. [4]Department of Pathology, St. Jude Children's Research Hospital, Memphis, TN 38105, USA. [5]Department of Oncology, The Hospital for Sick Children, Toronto, ON, Canada. [6]Children's Oncology Group and Department of Population and Public Health Sciences, Keck School of Medicine of University of Southern California, Los Angeles, CA, USA. [7]Children's Oncology Group Statistics and Data Center, Monrovia, CA, USA. [8]Department of Genetics, The University of Texas MD Anderson Cancer Center, Houston, TX, USA. [9]Department of Pediatrics, Emory University School of Medicine, Atlanta, GA, USA. [10]Department of Pediatric Oncology, Dana-Farber/Boston Children's Cancer and Blood Disorders Center and Harvard Medical School, Boston, MA 02215, USA. [11]Broad Institute of Harvard and MIT, Cambridge, MA, USA. [12]Center for Cancer and Blood Disorders, Children's National Hospital, Department of Pediatrics, George Washington University School of Medicine and Health Sciences, Washington, DC, USA. [13]IWK Health Center and Dalhousie University, Halifax, NS, Canada. [14]Division of Oncology, Cincinnati Children's Hospital Medical Center, University of Cincinnati, Cincinnati, OH, USA. [15]Section of Pediatric Surgery, C.S. Mott Children's Hospital, University of Michigan, Ann Arbor, MI, USA. [16]Department of Oncology, St. Jude Children's Research Hospital, Memphis, TN, USA. [17]Department of Pathology, University of Colorado Anschutz, Aurora, CO, USA. [18]Department of Epidemiology and Cancer Control, St. Jude Children's Research Hospital, Memphis, TN, USA. [19]These authors contributed equally: Andrew J. Murphy, Changde Cheng. [20]These authors jointly supervised this work: Andrew J. Murphy, Xiang Chen. ✉e-mail: andrew.murphy@stjude.org; Xiang.chen@stjude.org

