## [Peer Review File · Nature Communications]

Genetic and Epigenetic Features of Bilateral Wilms Tumor
Predisposition in Patients from the Children's Oncology Group
AREN18B5-QReviewers' Comments:

Reviewer #1:

Remarks to the Author:

This is a very carefully conducted in-depth genomic analysis of two cohorts of children with synchronous bilateral Wilms tumours. The first cohort came from a single institutional series sampled at St Jude Children's Research Hospital, using whole exome sequencing. The second 'expansion' cohort was sourced from the Children's Oncology Group biobank and was analysed using whole genome sequencing to allow inclusion of non-coding variants in the analyses of shared or unique changes between paired tumours from the same patient. The total cohort successfully analysed was 68 patients. The work described in this manuscript represents a significant contribution to the global knowledge on genomic and epi-genomic changes underlying the formation of bilateral WTs. In addition, the authors provide expanded evidence for early post-zygotic mosaicism and, for the first time, fascinating data on the quantitative increase in methylation levels of the H19/ICR1 in peripheral blood lymphocytes from longer term survivors of bilateral Wilms tumour, which is not seen in children with unilateral tumours or in normal controls.

The manuscript could be improved by consideration of the following points:

1. Abstract – suggest that the authors include a quantitative % statement on the frequency of the various epigenetic changes affecting 11p15 ICR1, to aid the reader who will only read the abstract.
2. Methods – includes a description of RNAseq being applied to the specimens, but no results of this assay are described or discussed. Is this data being held for a future paper or did it contribute to validating the outputs of the DAVID pathway analyses described on line 378 or identifying genes whose expression is affected by the methylation changes? Suggest that a rationale is provided for why RNAseq is mentioned in the paper.
3. Results - There are some discrepancies between numbers of patients with the various mutations written in the text and the numbers shown in Fig 1. This needs a thorough check and correction by the authors so that the text and figures match. For example:

3a: Lines 306-307, it is stated "Of these 25 patients, 19 had one predisposing germline variant, five had two predisposing germline variants, and one had three predisposing germline variants". Fig 1 shows 20 patients with one mutation and none with 3 predisposing germline mutations.

3b: Lines 313 onward: –"Out of 14 tumor samples from the 9 patients with germline WT1 variants, 11 (78.6%) were found to have acquired somatic activating variants in exon 3 of CTNNB1". However, fig 1 only shows 10 tumours with somatic CTNNB1 mutation.
- 4 Results - Line 373 – 377 – why is there no mention that 4 tumours had Histone H3 mutation. This should be added for completeness since all other somatic gene mutations are mentioned in order of frequency.
5. Results - Line 400 onwards. The paragraph describing the results of whole genome sequencing in 15 synchronous bilateral WT needs some attention.

The numbers of the different categories of patient samples in the text does not match that shown in table 1. For example, there are 5 paired synchronous BWT without shared noncoding somatic variants, not the n= 4 written in the paragraph. Similarly, there are 6 synchronous BWT pairs without an identifiable germline predisposing variant shown in table 1, rather than the n=5 written in the text, one of whom had mixed LOI/ROI at 11p15.5. Furthermore, 4/6 only had a single shared non-coding somatic mutation between the two tumours. This point should be emphasized as it may have implications for the timing of the post-zygotic LOI in early embryogenesis.

Why is no comment included on the remaining 4 cases, two of whom have germline WT1 mutation associated with a small number of shared somatic non-coding variants (2 & 5) and LOH at 11p15. Could the authors comment on this finding, albeit on only two tumours, e.g. does it imply a possible requirement for duplication of 11p15 ahead of malignant transformation on the background of WT1 mutation (as has been seen in animal models and previous analyses of Wilms tumours?).

Reviewer #2:

Remarks to the Author:

Murphy et al. present data comparing and contrasting blood, kidney, and tumor samples from patients diagnosed with bilateral Wilms tumors, using a variety of sequencing and methylation assays. Overall, there was a significant amount of work performed however the integration of the data and the conclusions drawn do not provide a clearly defined sequence. The authors propose possible steps the evolution of Wilms tumors from the early genetic changes to the somatic changes, which the authors present as signature in disease progression. However, it is unclear based on the division of samples how this occurs.

While there is a great deal of analysis that was conducted on human samples, the presentation of data, the interpretation of data, and the conclusions drawn need to be improved. The manuscript is very descriptive and driven by theories and not hypotheses. The figures are not well presented, some could be combined rather than separated, and several need labels to be added or corrected, and each part of each figure need to be referenced and explained in the text or not included. Some figures also depict samples that do not appear in the original descriptive Figure 2 or are extremely challenging to read as presented. The rationale for splitting of the discovery and expansion cohorts is unclear unless the results are revised to use the second cohort as a validation cohort. The current form of the paper does not clearly answer the overall question of what do the genetic and epigenetic findings mean in relation to the progression of bilateral Wilms tumors. The manuscript requires significant changes to the presentation and analysis of the data and conclusions.

Specific comments

1. The numbers of samples studied need to be clarified as do the paired-normal samples. The abstract does not provide complete representation of the samples tested and the conclusions drawn based on the samples tested. "Germline" vs "somatic" changes need to be clarified, with germline being in blood and somatic being in tissue.
2. Are the germline WT1 mutations in patients with identified genetic syndromes? If yes, what syndromes, if no, how was it ascertained that these patients are not syndromic?
3. WT1 plus LOH is indicated in one place as mutually inclusive, is that actually the case?
4. How is the LOH distinguished from GOM or LOI? Do these patients have underlying clinical features suggestive of a syndrome? How was this determined?
5. The somatic mosaicism for LOI falls within the normal range per the parameters defined in the study, so how is drift or somatic mosaicism defined for these BWT patients and long-term survivors? Please clarify the technique and data analysis approach used and revise this section in the abstract and the manuscript.
6. Reframe the mesoderm hypothesis to consider blood and the timing of imprinting being established during embryonal development. Clarify how the normal kidney fits into this and indicate which normal kidney samples are analyzed in parallel to tumor samples to support this hypothesis.

7. How do the authors define the patients with 11p15 alterations in blood and/or normal kidney in addition to tumor?
8. Is paternal uniparental isodisomy and LOH defined as the same thing/process? Please comment on the timing of these events and how that correlates with the data presented.
9. Figure 1 should be in the supplement.
10. Sample acquisition section: how do the levels of LOH or LOI compare to the % tumor in a specimen? How was this accounted for/determined?
11. Figure 2: Add the paired normal samples to the top part of the figure. Where are the normal samples in the discovery cohort? What were the tumor analyses compared to in this case? Are the signatures from discovery cohort comparable to expansion cohort? Figure 2 needs more detailing with respect to sample processing and comparison.
12. Why is the discovery cohort named as such? If there is no validation cohort because the cohorts are designed differently and are showing different things, please rename and state this in the manuscript.
13. Clinical data: how was it determined if the patients had congenital anomalies or syndromes? Please define this or state it is not known.
14. Total-strand RNA-seq – were normal sample run here? Where are the data?
15. Methylation analysis – define what was used as “germline.” Blood? Kidney? Define the criteria for normal/abnormal in the results and discussion of somatic mosaicism as it differs from what is defined in this section. Were normal kidney/tumor pairs tested? Be clearer if it is tumor/tumor pairs vs tumor/normal pairs. How were methylation results from the whole genome sequencing confirmed? In the results, methylation analysis and WGS results should both be provided. Please add these data. What was done with methylation data beyond the 11p15 region?
16. Long terms survivorship analysis: please define how this analysis was done more clearly as the discussion focuses on variations that are within the normal range per the parameters set.
17. How do the different or similar CTNNB1 mutations fit into the timing of the 11p15 alterations discussion?
18. Clarify how many WT1 patients also had 11p15 alterations.
19. Clarify the parameters of patients with 11p15 alterations in multiple tissues and their clinical phenotypes, specifically if there were subtle clinical phenotypes.
20. Figure 3 is too small to read and blurry.
21. Line 381-388 - For section, “Somatic variants are not shared in synchronous BWT”, the authors need to specify how many genes were used for pathway analysis. The Supplementary Table 1 shows 4 clusters. How were these clusters derived? Which genes were included in each set? Please discuss the method and results in greater details. How does the RNA data correlate with the methylation data for the same regions?
22. For section, “Somatic variants are not shared in synchronous BWT”, based on Supplementary figure 3, chromosome 22 also showed variations. Please comment on this.

23. Figure 4 is confusing and does not add to the explanation of the data. Please make a table instead and add blood status of 11p15 as well.

24. The authors mention in the method section that samples with average beta value H19/ICR1 <0.7 have normal ROI. Figure 5A,D show that all the samples from all three cohorts have normal ROI. Why did the authors make an age dependent methylation comparison of ICR1? Are the results within the normal range or outside or just demonstrating assay or sample variability? Specifically define LOI, LOH and ROI for this figure and how it is similar or different to the ranges for the other data presented. If 11p15.5 status is shared in BWT: What is the 11p15.5 status in the normal tissue? The "established thresholds" are not well defined in the context of the somatic analysis, please clarify. Please clarify why the extent of LOH did not overlap in one case. Was that between tumors, between tumor and kidney or blood? Was this LOH or paternal uniparental isodisomy?

25. The authors propose that post-zygotic 11.p15.5 alterations lead to development of BWT. However, Figure 6 shows that 11.p15.5 alterations are responsible for both unilateral and BWT. Lines 487-505 state molecular signatures that identify BWT. The authors need to state these molecular identifiers clearly. In addition, authors state, "We noted that BWT predominantly clustered distinctly from unilateral WT and closer to non-diseased adjacent kidney tissue, consistent with previously published results (Figure 6)." However, the published results state that there were no consistent differences in DNA methylation between unilateral and bilateral tumors. Please provide a consistent explanation between these results and the published results, given shared authorship with this manuscript. What is the threshold for defining a genetic/epigenetic syndrome in these patients?

26. Figure 6 displays 4 tissue types in study clustered separately. Authors need to show the molecular identities of each tissue. In addition, show these distinct clusters in Supplementary Figure 7. For Figure 6: "sex" not "gender" should be reported. Why are metastases included in this Figure and the analysis?

27. Figure 7: If tumor purity was initially defined as greater than 50% and then $>80\%$ it is unclear what samples throughout the paper have what levels of tumors and that influences the results. There needs to be clear assessment of that in each data set and each sample. Perhaps the sequencing data can be used as a proxy for this. Given that somatic levels of 11p15 changes are being discussed this needs to be thoroughly clarified throughout the manuscript.

28. The authors introduced pathogenic or likely pathogenic variants in the abstract. Figure 8 shows role of pathogenic variants in prezygotic stage. The authors need to add the predisposing variants to section of "3. somatic mosaic 11p15.5 loss of imprinting" based on findings in Table 1. Also please add this to the discussion.

29. In the discussion is the mechanism germline mutation plus LOH? If so, state that in the first paragraph. Does LOH occur first or after the germline mutations? How does LOH affect non-chromosome 11 germline changes?

30. How do the blood 11p15 findings factor into the developmental model?

31. Why does 11p15 alteration level increase with age?

32. How do different pathologic findings in these tumors fit into the two BWT models proposed? Addition of this pathology data would be informative.

33. How does "clonal nephrogenesis" result in BWT? Developmentally when does this clonality occur?

34. How do the changes in long term survivors of the 11p15 region validate somatic mosaic findings in

the BWT patients? It is unclear how the data presented support this statement.

35. The data on amplified clonal mosaicism increasing over time is limited, where the data on mosaic 11p15 changes in patients with Beckwith-Wiedemann Spectrum is more plentiful. Please comment on this in the context of the discussion.

36. How is 11p15 status a biomarker for global methylation patterns? Please revise this and provide the evidence to support that revision. Was it high or low stage risk? Both are stated?

Minor comments –

1. Reference which section of a figure is being discussed each time in the text, not just the overall figure number.

2. Line 105-109 - In the introduction, the authors need to include Beckwith-Wiedemann syndrome in the predisposition syndromes for Wilms tumor.

3. In Supplementary Figure 1 legend, the authors need to add tumor information with respect to complete LOH.

4. The authors need to improve the resolution of Supplementary Figure 2 and add scale bar to it.

5. Line 362-388 - The authors need to rearrange the section, "Somatic variants are not shared in synchronous BWT" such that somatic variants results are stated together and copy number variants results are stated together.

6. Supplementary Figure 3 displays a cross and asterisks. Please add the details for these in the figure legend.

7. Line 531-538 - Please add germline type abbreviation details for figure 7 legend.

Reviewer #3:

Remarks to the Author:

After spending over years in the Wilms tumor field I'm still surprised by how difficult the answer to the apparently simple question 'what is a Wilms tumor, where is it coming from and what caused it' turns out to be. In this manuscript, Murphy and colleagues focus on bilateral Wilms tumors which by their nature can provide unique insights in the causes and origins of these tumors. They perform a combined genomic and epigenomic analysis of BWTs from 68 patients. The nature of such a dataset itself makes this work a valuable addition to our understanding of this disease and to the literature.

Two important points before I give my opinion on this manuscript. First, I'm myself not an expert in the techniques and analyses used in this work, and how the primary data lead to the results as presented. I will simply assume this is all correct hoping other reviewers have more to say on this. Instead I will focus on the question if the presented data indeed leads to the conclusions as given, how this fits with the existing body of literature and what new insights we can gain from this. Second, these sort of manuscripts and papers are, due to the amount of data generated and the limitations placed on a manuscript size by the journals, never an easy read. This manuscript I find extremely hard work, which is maybe to be expected given the number of germline and somatic mutations and possible 11p15 changes. If (some of) my comments are caused by my misunderstanding the intended meaning of the text I apologize to the authors. I certainly would not be able to write this sort of manuscript better.

The manuscript roughly divides the group of BWTs in two: those with germline mutations (often combined with 11p15 LOH) and those without germline mutations (and instead showing 11p15 LOI). A variety of germline-mutated genes is identified, all of which have been linked to Wilms tumors though I think not necessarily as germline mutations (I might be wrong here). Mutations in WT1 were found to be most common in these cases. In these cases I miss (in my opinion important) details on the loss of the wild type allele, the second hit from the Knudson two-hit model. First, what sort of

mutation is responsible for this? Figure 8 presents a model where the germline mutation is on the paternal allele and LOH / paternal uniparental disomy on 11p15 is responsible for this loss of the wild type WT1 allele. If the germline mutation is indeed in the paternal allele this makes perfect sense (and has been suggested before), but I cannot find any data on whether this paternal or maternal mutant WT1 mutation has been tested. Is in the cases described here the mutant allele indeed the paternal allele? Without this, or if there are also cases with a maternal mutant copy, the model should not be presented as it is at the moment.

For me the biggest surprise in this part of the data was the observation of BRCA1-PALB2-BRCA2 germline mutations. As the authors mention, these germline mutations are usually found in families with (mainly) hereditary breast and ovarian cancer. Could the authors comment on whether these malignancies were also found in these families, and what might make the differences that in these few families BWTs are found whereas in most of families with these mutations do not have these?

Next, the co-occurrence of WT1 mutations with CTNNB1 mutations (especially affecting Ser45) is discussed. As the authors write, this is a confirmation of data that has been described several times before, and this is therefore not new but a useful confirmation nonetheless.

The discussion of the other-than WT1 germline mutations is a lot more scarce and fragmentary. Are these mutations predicted to be activating or inactivating, and coupled to this, are they heterozygous or homozygous? Are there (unique or recurring) additional somatic mutations in these tumors? Are there themes to be recognized with respect to some germline mutations and 11p15 status? Many of these things could be found in figure 3, but in my opinion these sort of figures (and many papers have them) provide too much data in one figure to be informative, an extra table summarizing what can be found specifically in germline mutation cases would be very useful.

I think the information in line 354-360 suggests that all the cases without germline mutations have 11p15 LOI. However, in this paragraph there are 37 tumors with germline mutations and 52 with 11p15 LOI, adding up to 89. However the total number of tumors is 99. Do I misunderstand this paragraph or is there a mistake in the numbers given?

An important part of the manuscript discusses the implications of the data for deducing the developmental stage that mutations / aberrations occur and the tumors start developing. This is especially something where the analysis of BWTs can be very informative. I fully agree with the conclusion that the small number of identical somatic mutations (like CTNNB1) in the two tumors of a patient means that this is a late event after the individual kidneys have started to form. However I struggle more with the interpretation that the many shared 11p15 aberrations mean these tumors arise post-zygotic but before the kidneys start to form (quickly after gastrulation). I agree this aberration is likely a very early event, but I'm not convinced that these are the cause of these tumors. The reason for my doubts on this is the mosaic 11p15 aberrations in normal kidney tissue presented here and in other publications. If the 11p15 event would be the causative event, and if there are no other mutations found (as I understand it), what makes the difference between a cell in the developing kidney that does become a Wilms tumor and a neighboring cell with the same 11p15 status that does not? It could be something epigenetic, but wouldn't at least a suggestion of this have been found in the methylation data?

I would happily admit that the early 11p15 event in these cases could provide the selective space in which additional events occur (whatever they are and for whatever reason they haven't been found here), which are then the cause of the tumors starting to develop. One could still call the 11p15 event the predisposing factor, but it would not be the causative event. This is not just a semantic argument, as it would make clear why this predisposing event in these cases cannot be used to determine the site and stage the tumors originate from, which would in my opinion the most important message from this work. I would be happy to be convinced otherwise by the authors.

My final two remarks are about the final two paragraphs in the discussion. The authors are very honest about the limitations of this study. Issues like genetic heterogeneity between cases in the causative mutations potentially make the dataset a collection of many different groups with only very few cases per group. Gadd et al showed in 2012 that different initiating mutations subdivide the

tumors in multiple groups with different histology and different stages of origin. If this is the case (and I fully believe this), can we even call these groups cases of the same disease, or are they different diseases with common symptoms? If their genetic and developmental origin is different, I tend to go for the latter. Differences in treatment regime between samples make this problem even bigger, as pretreatment might very well have an effect on the additional, later mutations that are found in the tumor as they can directly affect the selective pressure for these additional mutations. I certainly don't underestimate these sort of problems, especially in a disease where sample numbers are (luckily from the patient perspective) small to begin with. But at some point we run into so many known unknowns and unknown unknowns that the risk of overinterpretation of data becomes too big.

Finally the authors state their work has enabled them to provide strong support for the Knudson-Strong two-hit hypothesis (line 686-687). I fully agree that parts of the data, especially on the germline mutation cases, are completely consistent with this. I do not agree that these data make the already existing confirmation of the model in the 51 year since it was proposed substantially stronger than it already was without these data. If anything, if my issues are correct (and I might be completely wrong) these data show the limitations of the model in describing the origins of tumours at the level of detail we currently can. This would not invalidate the model, just nuance it, but it would in my opinion certainly not provide particular strong extra data to support it.

We would like to thank each of the reviewers for their detailed, rigorous review of our manuscript. This process has undoubtedly resulted in a better product. Itemized responses to each reviewer comment are in blue font below.

REVIEWER COMMENTS

Reviewer #1, expertise in Wilms Tumour and development (Remarks to the Author):

This is a very carefully conducted in-depth genomic analysis of two cohorts of children with synchronous bilateral Wilms tumours. The first cohort came from a single institutional series sampled at St Jude Children's Research Hospital, using whole exome sequencing. The second 'expansion' cohort was sourced from the Children's Oncology Group biobank and was analysed using whole genome sequencing to allow inclusion of non-coding variants in the analyses of shared or unique changes between paired tumours from the same patient. The total cohort successfully analysed was 68 patients. The work described in this manuscript represents a significant contribution to the global knowledge on genomic and epi-genomic changes underlying the formation of bilateral WTs. In addition, the authors provide expanded evidence for early post-zygotic mosaicism and, for the first time, fascinating data on the quantitative increase in methylation levels of the H19/ICR1 in peripheral blood lymphocytes from longer term survivors of bilateral Wilms tumour, which is not seen in children with unilateral tumours or in normal controls.

Response: Thank you for your detailed review of our manuscript.

The manuscript could be improved by consideration of the following points:

1. Abstract – suggest that the authors include a quantitative % statement on the frequency of the various epigenetic changes affecting 11p15 ICR1, to aid the reader who will only read the abstract.

Response: The abstract (previously over 300 words) has been modified according to this reviewer comment and the journal requirements, which limit the abstract to an unstructured 150-word format with no abbreviations or acronyms:

Abstract: *“Developing synchronous bilateral Wilms tumor suggests an underlying (epi)genetic predisposition. We evaluated this predisposition in 68 patients using whole exome or genome sequencing (n=85 tumors from 61 patients with matched germline blood DNA), RNA-seq (n=99 tumors), and DNA methylation analysis (n=61 peripheral blood, n=29 non-diseased kidney, n=99 tumors). We determined the predominant events for bilateral Wilms tumor predisposition: 1)pre-zygotic germline genetic variants readily detectable in blood DNA [WT1 (14.8%), NYNRIN (6.6%), TRIM28 (5%), and BRCA-related genes (5%)] or 2)post-zygotic epigenetic hypermethylation at 11p15.5 H19/ICR1 that is less readily detectable in peripheral blood in an individual and may require analysis of multiple tissue types for diagnosis. Of 99 total tumor specimens, 16 (16.1%) had 11p15.5 normal retention of imprinting, 25 (25.2%) had 11p15.5 copy neutral loss of heterozygosity, and 58 (58.6%) had 11p15.5 H19/ICR1 epigenetic hypermethylation (loss of imprinting). Here, we ascertained the (epi)genetic landscape of bilateral Wilms tumor predisposition.”*

2. Methods – includes a description of RNAseq being applied to the specimens, but no results of this assay are described or discussed. Is this data being held for a future paper or did it contribute to validating the outputs of the DAVID pathway analyses described on line 378 or identifying genes whose expression is affected by the methylation changes? Suggest that a rationale is provided for why RNAseq is mentioned in the paper.

Response: Thank you for calling our attention to this oversight. RNA-seq data have now been utilized/included in the manuscript in the following manner: unsupervised hierarchical clustering plot of all tumors (Supplementary Figure 9), TSNE plot of paired bilateral samples (Supplementary Figure 10), and the Cis-X allele-specific expression analysis that identified recurrent reduction to allele specific expression in *RNF185* located at chromosome 22q (Supplementary Table 2-3).

Overall, these results demonstrated that paired synchronous bilateral Wilms tumor samples clustered less robustly with one another by gene expression than methylation: **Results section (1st paragraph under subheading 11p15.5 status and genome-wide methylation/molecular identity in BWT):** *“In contrast, paired synchronous BWT samples clustered less robustly by unsupervised hierarchical clustering and TNSSE clustering of total-strand RNA-seq data (Supplementary Fig. 9-10).”*

Integrated analysis of RNA-seq and whole genome sequencing data using the Cis-X method identified a perturbation of allele specific expression due to chromosome 22 loss in samples with 11p15.5 LOI or LOH, which prompted a more in-depth investigation into chromosome 22 (detailed below): **Results section (2nd and third paragraphs under subheading 11p15.5 status and genome-wide methylation/molecular identity in BWT):** *“To further explore the suggestion that tumor 11p15.5 status correlates with a broader tumor molecular identity rather than only changes at the 11p15.5 locus itself, we determined genome-wide alterations in allele specific expression stratified by tumor 11p15.5 status (LOI, LOH, ROI) in the COG cohort using the Cis-X method, which integrates whole genome sequencing and total-strand RNA-seq data to determine the allele of origin for the RNA being expressed. Due to sample size limitations, we filtered results according to an unadjusted p-value of < 0.05 and a false-discovery rate p-value corrected for multiple testing of <0.25 for this analysis. As predicted, *KCNQ10T1* (FDR p=0.0046) and *KCNQ1* (FDR p=0.0046) were found to have perturbation of allele-specific expression in tumor samples with 11p15.5 LOH, but not ROI or LOI, while *INS-IGF2* and *IGF2* showed perturbation of allele-specific expression in both samples with 11p15.5 LOH and LOI. Five genes and/or noncoding RNAs outside the 11p15.5 region were found to have disruption in the normal patterning of allele-specific expression that correlated with tumor 11p15.5 status: *RNF185*, *SNORD116-4*, *C1QL3*, *CLEC12A*, and *NPAS2*. *RNF185*, a gene located at chromosome 22q12.2 that codes for a component of the E3 ubiquitin ligase pathway, was found to have reduction to monoallelic expression in 43.3% of tumor samples with 11p15.5 LOI (FDR p=0.240) and 35.7% of tumor samples with 11p15.5 LOH (FDR p=0.041), but 0% of samples with 11p15.5 ROI (Supplementary Table 2).*

*We then queried for possible mechanisms of *RNF185* reduction to monoallelic expression and noted recurrent chromosome 22q loss correlated with this phenomenon. Combined with the copy number data detailed above that showed recurrent chromosome 22q loss in our data set and shared chromosome 22q loss in 3 synchronous BWT pairs, we performed a focused analysis to determine the clinical and other molecular features of BWT samples with chromosome 22q loss. We found an enrichment for female*

biological sex, 11p15.5 LOI, and DGCR8 RNA microprocessor mutations in this cohort of 11 tumor samples with chromosome 22q loss from 8 patients (Supplementary Table 3)."

The DAVID pathway analysis was performed using inputs from whole exome/whole genome sequencing variants to elucidate pathways associated with genetic variants detected in bilateral Wilms tumor samples. The RNA-seq data were not incorporated into the DAVID pathway analysis.

3. Results - There are some discrepancies between numbers of patients with the various mutations written in the text and the numbers shown in Fig 1. This needs a thorough check and correction by the authors so that the text and figures match. For example:

Response: We greatly appreciate your detailed review of the numbers of patients and mutations in the manuscript text and figures. Your comments have helped to ensure consistency between the figures and text for this complex dataset. We have made the following corrections and clarifications in response to your comments noted below:

3a: Lines 306-307, it is stated "Of these 25 patients, 19 had one predisposing germline variant, five had two predisposing germline variants, and one had three predisposing germline variants". Fig 1 shows 20 patients with one mutation and none with 3 predisposing germline mutations.

Response: **Figure 2** continues to depict 25 total patients indicated by green boxes with any predisposing germline variants: SJWLM 066770, 066773, 066776, 066777, 066779, 066780, 066789, 066792, 069396, 051024, 051028, 043953, 066772, 066774, 066783, 066788, 066791, 069379, 069381, 069382, 069385, 069389, 069390, 069395, and 069397.

Figure 2 now depicts 5 total patients with two predisposing germline variants: SJWLM066773 (*BRCA1* and *TRIM28*), SJWLM066792 (*NYNRIN* and *ASXL1*), SJWLM066774 (*WT1* and *TRIP13*), SJWLM066788 (*NYNRIN* and *KDM3B*), and SJWLM069390 (*NYNRIN* and *CTR9*). There are no patients in this study that exhibited three predisposing germline variants.

The reason the original error was made is because there were two patients with pathogenic germline variants in non-cancer related genes and genes that are not biologically related to Wilms tumor or Wilms tumor predisposition that were initially included in counts. These genes were supposed to be removed from the manuscript text/figures at the advice of our collaborating cancer predisposition experts; however, the text edits were incomplete. The following revisions to the manuscript text now reflect the correct information:

Results (1st paragraph under subheading *Germline variant analysis from blood and associated tumor findings* subheading): "Of these 25 patients, 20 had one predisposing germline variant and five had two predisposing germline variants: SJWLM066773 (*BRCA1* and *TRIM28*), SJWLM066792 (*NYNRIN* and *ASXL1*), SJWLM066774 (*WT1* and *TRIP13*), SJWLM066788 (*NYNRIN* and *KDM3B*), and SJWLM069390 (*NYNRIN* and *CTR9*)."

3b: Lines 313 onward: –"Out of 14 tumor samples from the 9 patients with germline *WT1* variants, 11 (78.6%) were found to have acquired somatic activating variants in exon 3 of *CTNNB1*". However, fig 1 only shows 10 tumours with somatic *CTNNB1* mutation.

Response: The text has been corrected to reflect the correct number of 10 somatic activating variants in exon 3 of *CTNNB1* among 14 tumor samples from 9 patients with germline WT1 variants:

Results (1st paragraph under subheading *Germline variant analysis from blood and associated tumor findings* subheading): *“Out of 14 tumor samples from the 9 patients with germline WT1 variants, 10 (71.4%) were found to have acquired somatic activating variants in exon 3 of CTNNB1, which codes for the Wnt pathway effector transcription factor β -Catenin.”*

4 Results - Line 373 – 377 – why is there no mention that 4 tumours had Histone H3 mutation. This should be added for completeness since all other somatic gene mutations are mentioned in order of frequency.

Response: The following text has been added to the results section for completeness in response to this comment:

Results (2nd paragraph under subheading *Tumor somatic variants are not shared in synchronous BWT*): *“Variants in genes coding for histone H3 (4/85, 4.7%) were also found in this study, including HIST1H3I p.K28M in SJWLM069380, HIST1H3I p.R117C in SJWLM51020, and H3F3A p.R50C in SJWLM66770 and the metastatic sample SJWLM51027 (Fig. 2).”*

5. Results - Line 400 onwards. The paragraph describing the results of whole genome sequencing in 15 synchronous bilateral WT needs some attention.

The numbers of the different categories of patient samples in the text does not match that shown in table 1. For example, there are 5 paired synchronous BWT without shared noncoding somatic variants, not the n= 4 written in the paragraph. Similarly, there are 6 synchronous BWT pairs without an identifiable germline predisposing variant shown in table 1, rather than the n=5 written in the text, one of whom had mixed LOI/ROI at 11p15.5. Furthermore, 4/6 only had a single shared non-coding somatic mutation between the two tumours. This point should be emphasized as it may have implications for the timing of the post-zygotic LOI in early embryogenesis.

Response: The text has been corrected and modified according to this comment:

Results: (4th paragraph under subheading *Tumor somatic variants are not shared in synchronous BWT*): *“In this analysis, all paired synchronous BWT without shared noncoding somatic variants (n=5) had an identifiable germline variant in a WT or pediatric cancer predisposition gene. For synchronous BWT pairs without an identifiable germline predisposing variant (n=6), 11p15.5 LOI was detected in five pairs and mixed 11p15.5 LOI/ROI in one pair. In addition, shared tumor somatic noncoding variants were detected in all six pairs without identifiable blood germline predisposing variants (Table 2). Four of these six pairs exhibited only one shared tumor noncoding somatic variant, implying spatial divergence extremely early during early embryogenesis.”*

Why is no comment included on the remaining 4 cases, two of whom have germline WT1 mutation associated with a small number of shared somatic non-coding variants (2 & 5) and LOH at 11p15. Could the authors comment on this finding, albeit on only two tumours, e.g. does it imply a possible requirement for duplication of 11p15 ahead of malignant transformation on the background of WT1

mutation (as has been seen in animal models and previous analyses of Wilms tumours?).

Response: Thank you for this suggestion. We have added the following language to the discussion section of the manuscript: **Discussion (3rd paragraph):** *“Two synchronous BWT sample pairs in this study (SJWLM066776 and SJWLM069396) from patients with inactivating pathogenic blood germline WT1 variants demonstrated a small number of shared noncoding tumor somatic variants in addition to tumor 11p15.5 LOH (Table 2). This constellation of findings implies 11p15.5 LOH can occur in the early embryo and may be required ahead of malignant transformation on the background of WT1 mutation.”*

Reviewer #2, expertise in genetics and epigenetics for pediatric cancers (Remarks to the Author):

Murphy et al. present data comparing and contrasting blood, kidney, and tumor samples from patients diagnosed with bilateral Wilms tumors, using a variety of sequencing and methylation assays. Overall, there was a significant amount of work performed however the integration of the data and the conclusions drawn do not provide a clearly defined sequence. The authors propose possible steps the evolution of Wilms tumors from the early genetic changes to the somatic changes, which the authors present as signature in disease progression. However, it is unclear based on the division of samples how this occurs.

While there is a great deal of analysis that was conducted on human samples, the presentation of data, the interpretation of data, and the conclusions drawn need to be improved. The manuscript is very descriptive and driven by theories and not hypotheses. The figures are not well presented, some could be combined rather than separated, and several need labels to be added or corrected, and each part of each figure need to be referenced and explained in the text or not included. Some figures also depict samples that do not appear in the original descriptive Figure 2 or are extremely challenging to read as presented. The rationale for splitting of the discovery and expansion cohorts is unclear unless the results are revised to use the second cohort as a validation cohort. The current form of the paper does not clearly answer the overall question of what do the genetic and epigenetic findings mean in relation to the progression of bilateral Wilms tumors. The manuscript requires significant changes to the presentation and analysis of the data and conclusions.

Response: Thank you for your detailed review of our manuscript and precise suggestions for improvement.

Specific comments

1. The numbers of samples studied need to be clarified as do the paired-normal samples. The abstract does not provide complete representation of the samples tested and the conclusions drawn based on the samples tested. “Germline” vs “somatic” changes need to be clarified, with germline being in blood and somatic being in tissue.

Response: The abstract (previously over 300 words) has been modified according to this reviewer comment and the journal requirements, which limit the abstract to an unstructured 150-word format with no abbreviations or acronyms:

“Developing synchronous bilateral Wilms tumor suggests an underlying (epi)genetic predisposition. We evaluated this predisposition in 68 patients using whole exome or genome sequencing (n=85 tumors from 61 patients with matched germline blood DNA), RNA-seq (n=99 tumors), and DNA methylation analysis (n=61 peripheral blood, n=29 non-diseased kidney, n=99 tumors). We determined the predominant events for bilateral Wilms tumor predisposition: 1)pre-zygotic germline genetic variants readily detectable in blood DNA [WT1 (14.8%), NYNRIN (6.6%), TRIM28 (5%), and BRCA-related genes (5%)] or 2)post-zygotic epigenetic hypermethylation at 11p15.5 H19/ICR1 that is less readily detectable in peripheral blood in an individual and may require analysis of multiple tissue types for diagnosis. Of 99 total tumor specimens, 16 (16.1%) had 11p15.5 normal retention of imprinting, 25 (25.2%) had 11p15.5 copy neutral loss of heterozygosity, and 58 (58.6%) had 11p15.5 H19/ICR1 epigenetic hypermethylation (loss of imprinting). Here, we ascertained the (epi)genetic landscape of bilateral Wilms tumor predisposition.”

2. Are the germline WT1 mutations in patients with identified genetic syndromes? If yes, what syndromes, if no, how was it ascertained that these patients are not syndromic?

Response: The methods section has been updated to address this question: **Methods (under Clinical Data subheading):** *“For the St. Jude cohort, syndromes were ascertained by thorough review of the medical record and all molecular testing documentation. For the COG cohort, the presence of genetic syndromes was a collected/reported data point from the originating studies and was delivered to the investigators after molecular analysis was complete.”*

Our data are consistent with recent unselected cohorts that demonstrated patients with pathogenic germline variants often have no overt clinical phenotype other than the development of Wilms tumor (Hol, J.A. *et al.* Prevalence of (Epi)genetic Predisposing Factors in a 5-Year Unselected National Wilms Tumor Cohort: A Comprehensive Clinical and Genomic Characterization. *J Clin Oncol*, JCO2102510 [2022]).

The results section has been updated to address this question: **Results (under Germline variant analysis from blood and associated tumor findings heading):** *“Of 9 patients with pathogenic WT1 germline variants in this study, 3 had features of genetic syndromes and 6 did not. SJWLM066776 had Denys Drash syndrome, SJWLM066772 had a disorder of sexual development (DSD), and SJWLM066774 had congenital nephrotic syndrome and idiopathic dilated cardiomyopathy.”*

3. WT1 plus LOH is indicated in one place as mutually inclusive, is that actually the case?

Response: Out of 14 tumor samples from the 9 patients with germline *WT1* variants in the current study, all 14 (100%) were found to have 11p15.5 LOH. However, these findings are not mutually inclusive because tumor samples without *WT1* germline variants were also found to have 11p15.5 LOH in this study as demonstrated in **Figure 2**.

The explanation for this strong association between pathogenic germline *WT1* variants and 11p15.5 LOH is indicated in **Figure 6** in which a heterozygous inactivating germline *WT1* pathogenic variant present on

the paternal allele becomes homozygous by copy neutral loss of heterozygosity at 11p with breakpoints that include both the *WT1* locus at 11p13 and the H19/IGF2 locus at 11p15.5. In fact, the breakpoints of 11p15.5 LOH encompass both loci in 18/19 cases (94.7%) with 11p15.5 LOH in this study.

However, in our prior study (Murphy, A.J. *et al.* Forty-five patient-derived xenografts capture the clinical and biological heterogeneity of Wilms tumor. *Nat Commun* **10**, 5806 [2019].), samples from 5/7 patients with *WT1* mutations had 11p15.5 LOH. 4 of these 7 patients had bilateral Wilms tumor and all these 4 had 11p15.5 LOH. The two patients that did not have 11p15.5 LOH were unilateral Wilms tumor samples. Also, there are multiple examples of unilateral Wilms tumors in the NCI-TARGET data set that have *WT1* mutations and do not have 11p15.5 LOH. We suspect that the 11p15.5 LOH/*WT1* mutual inclusivity is true in patients with germline *WT1* variants and bilateral Wilms tumor, but when *WT1* mutations are somatic in patients with unilateral Wilms tumor this association may not hold true.

The discussion section has been updated in response to this comment: **Discussion (2nd paragraph):** *“Although all tumor samples from patients with pathogenic *WT1* germline variants in the current study exhibited 11p15.5 LOH, prior work from our group and others in unilateral Wilms tumor samples demonstrated somatic *WT1* mutations without 11p15.5 LOH.”*

4. How is the LOH distinguished from GOM or LOI? Do these patients have underlying clinical features suggestive of a syndrome? How was this determined?

Response: 11p15.5 LOH was distinguished from 11p15.5 LOI (aka 11p15.5 H19/ICR1 GOM) using data from the MethylationEPIC beadchip array. The average β value for H19/ICR1 was calculated using CpG probes located within the chr11:2,019,974-2,024,738 (GRCh38/hg38) range and the average β value for 11p15.5 KCNQ1OT1/ICR2 was calculated using probes located within the chr11:2,721,228-2,722,228 (GRCh38/hg38) range. Samples with average β value H19/ICR1 < 0.7 and KCNQ1OT1/ICR2 > 0.3 were determined to have normal retention of imprinting (ROI), samples with H19/ICR1 > 0.7 (hypermethylation) and KCNQ1OT1/ICR2 > 0.3 were determined to have loss of imprinting (LOI) at H19/ICR1, and samples with H19/ICR1 > 0.7 (hypermethylation) and KCNQ1OT1/ICR2 < 0.3 (hypomethylation) were determined to have loss of heterozygosity (LOH) at 11p15.5. LOH at 11p15.5 was also designated if samples had LOH or partial LOH detectable using the CONSERGING algorithm from whole genome sequencing data. This is detailed in the **methods section of the manuscript (2nd paragraph under Methylation analysis subheading).**

Thank you for your attention to details of the clinical syndromes. The methods section has been updated to address this question: **Methods (under Clinical Data subheading):** *“For the St. Jude cohort, syndromes were ascertained by thorough review of the medical record and all molecular testing documentation. For the COG cohort, the presence of genetic syndromes was a collected/reported data point from the originating studies and was delivered to the investigators after molecular analysis was complete.”*

The results section has been updated to address this question: **Results (2nd paragraph under 11p15.5 status is shared in synchronous BWT subheading):** *“Most patients with 11p15.5 LOH or 11p15.5 LOI detected in their tumor samples had no clinical features suggestive of a syndrome (Fig. 2). However, of the 9 patients with definitive 11p15.5 alterations in multiple tissues, SJWLM066781 had 11p15.5 LOI in their tumor and adjacent non-diseased kidney and had a ureteral duplication. SJWLM069381 had 11p15.5 LOI in their tumor and adjacent non-diseased kidney tissue and had hemihypertrophy.*

SJWLM069390 had 11p15 LOH detected in their tumor and adjacent non-diseased kidney tissue. In addition, this patient was found to have mosaic 11p15.5 LOH in their blood as determined by whole genome sequencing (Supplementary Fig. 2). This patient was noted to have a BWSp clinical phenotype.”

5. The somatic mosaicism for LOI falls within the normal range per the parameters defined in the study, so how is drift or somatic mosaicism defined for these BWT patients and long-term survivors? Please clarify the technique and data analysis approach used and revise this section in the abstract and the manuscript.

Response: The text of the results section has been adjusted to address this comment: **Results (4th-6th paragraphs under 11p15.5 status is shared in synchronous BWT subheading):** *“11p15.5 LOI being found in paired synchronous BWT that also contained shared tumor noncoding somatic variants and in adjacent non-diseased kidney tissue (but not above established thresholds in DNA obtained from leukocytes) is suggestive of post-zygotic somatic mosaicism. We reasoned that, if present, evidence of post-zygotic somatic mosaicism for chromosome 11p15.5 LOI should be detectable in peripheral blood on the cohort level since hematopoietic progenitor cells have a mesodermal embryonic origin.³⁴ To explore whether evidence of somatic mosaicism could be detected on the cohort level in peripheral blood samples from patients with tumors bearing 11p15.5 LOI, we compared the H19/ICR1 β values from leukocyte-derived DNA from peripheral blood among patients according to the 11p15.5 status of their tumors (ROI, LOH, LOI). We found a statistically significant increase in H19/ICR1 methylation detectable in peripheral blood in patients who had tumors with 11p15.5 LOI compared to those with retention of imprinting (Fig. 3A). Of note, this statistically significant gain of methylation was low-level with none of the peripheral blood samples achieving the β value of 0.7 associated with frank germline 11p15.5 LOI. In contrast, 7 of the adjacent non-diseased kidney samples met the threshold for frank germline 11p15.5 LOI (Fig. 3B).”*

“To validate the finding of somatic mosaicism for 11p15.5 LOI detectable in peripheral blood on the cohort level, we combined identically processed and normalized H19/ICR1 methylation β values from leukocyte-derived DNA between our current BWT cohort (n=61) and a cohort of healthy community controls (n=282) and WT cancer long-term survivors (including survivors of both unilateral [n=154] and BWT [n=17]) from the St. Jude Life Cohort Study.³⁵ We hypothesized that H19/ICR1 values in peripheral blood DNA would be higher in BWT patients than healthy community controls and potentially unilateral WT patients. The peripheral blood 11p15.5 H19/ICR1 and KCNQ10T1/ICR2 β values were determined to be normally distributed in all groups using the Kologov-Smirnov test. The mean H19/ICR1 peripheral blood methylation β value in the healthy community control cohort was 0.499 ± 0.0248 (mean \pm standard deviation). The mean H19/ICR1 peripheral blood methylation β value from the BWT cohort (0.534 ± 0.0298) was higher than both the unilateral WT (0.521 ± 0.0258 ; $p < 0.0001$) and healthy community control cohorts ($p < 0.0001$; Fig 3D.) Using the method of Fiala, et. al.²³, we determined that 26 BWT patients (n=20 from current study; n=6 from St. Jude Life cohort) had a H19/ICR1 methylation β value greater than two standard deviations from the mean community control cohort β value (i.e., H19/ICR1 $\beta > 0.54864$; Fig. 2; Supplementary Table 1). Of these 20 patients, 17/20 (85%) were female, 18/20 (90%) had 11p15.5 LOI in their tumor, 2/20 (10%) had 11p15.5 LOH in their tumor, and 0 had 11p15.5 ROI in their tumor (Supplementary Table 1).”

“However, because leukocyte DNA methylation is known to change with age³⁶ and the healthy control population with peripheral blood DNA available for this study was derived from cancer survivors of a higher

median age than our current study population, we wanted to assess H19/ICR1 methylation as a function of age. We noted an inverse relationship between methylation at H19/ICR1 and increasing age in healthy community controls (Fig. 3E). In contrast, we detected positive correlation between increasing age and H19/ICR1 methylation in BWT patients and long-term survivors. This positive correlation was not seen in long-term survivors of unilateral WT (Fig. 3E)."

The limitations paragraph of the discussion section has also been updated to address this comment: **Discussion (2nd to last paragraph):** *"Finally, all data in this manuscript reporting evidence of mosaic 11p15.5 H19/ICR1 gain of methylation detectable in peripheral blood are based on statistical comparison or observed changes in comparison to a normal cohort, while no patients in this manuscript exhibited frank hypermethylation above the established threshold of a β value of 0.7 at 11p15.5 H19/ICR1 detectable in DNA obtained from peripheral blood."*

6. Reframe the mesoderm hypothesis to consider blood and the timing of imprinting being established during embryonal development. Clarify how the normal kidney fits into this and indicate which normal kidney samples are analyzed in parallel to tumor samples to support this hypothesis.

Response: Thank you for this suggestion to clarify, or more explicitly state, the underlying hypothesis. We have rewritten the discussion section of the manuscript with this specific comment in mind. Although the whole discussion section was reworked with this comment in mind, the most relevant section is noted here: **Discussion (4th paragraph):** *"We hypothesized that 11p15.5 LOI (H19/ICR1 gain of methylation) was a somatic mosaic epigenetic event that occurred in the early embryo in a mesodermal progenitor cell during the time when genomic imprints are being established and led to BWT predisposition. Genomic DNA methylation, which is the primary responsible mechanism for imprinting, reaches its final level near the time of embryonic gastrulation (i.e., formation of the three germ layers – ectoderm, endoderm, mesoderm).⁴⁸ This developmental timepoint is precisely when the mesodermal cells that give rise to the right and left intermediate mesoderm/kidney primordia are spatially isolated as they invaginate to establish the mesoderm and migrate to the right or left lateral sides of the embryo.⁴⁹ Multiple lines of evidence in this study support this mesodermal somatic mosaic hypothesis. First, 11p15.5 LOI is shared in many synchronous BWT, while other downstream common somatic genetic alterations (exact variants in CTNNB1, microRNA processing genes, SIX1/2, TP53, etc.) are not. Next, synchronous BWT with shared 11p15.5 LOI also exhibit a small number of shared noncoding variants relative to the total number of noncoding variants demonstrated in each paired tumor, consistent with shared clonal origin in the early embryo with subsequent divergence and independent evolution. Finally, the frequent detection of 11p15.5 H19/ICR1 hypermethylation at frank germline thresholds ($\beta > 0.7$) in adjacent non-diseased kidney tissue and associated tumors but not blood is consistent with mosaicism throughout the body."*

7. How do the authors define the patients with 11p15 alterations in blood and/or normal kidney in addition to tumor?

Response: We have updated the results section of the manuscript to feature patients more clearly with 11p15 alterations in multiple tissues: **Results (2nd paragraph under 11p15.5 status is shared in synchronous BWT subheading):** *"However, of the 9 patients with definitive 11p15.5 alterations in multiple tissues, SJWLM066781 had 11p15.5 LOI in their tumor and adjacent non-diseased kidney and*

had a ureteral duplication. SJWLM069381 had 11p15.5 LOI in their tumor and adjacent non-diseased kidney tissue and had hemihypertrophy. SJWLM069390 had 11p15 LOH detected in their tumor and adjacent non-diseased kidney tissue. In addition, this patient was found to have mosaic 11p15.5 LOH in their blood as determined by whole genome sequencing (Supplementary Fig. 2). This patient was noted to have a BWSp clinical phenotype.”

Except for patient SJWLM069390 (featured in Supplementary Figure 2), who had definitive determination of 11p15.5 LOH mosaicism detected in peripheral blood, we could not definitively assign mosaic status to a blood sample from any other patient in this study. However, detection of 11p15.5 alterations in kidney and tumor, but not blood is highly suggestive of somatic mosaicism. This has been highlighted more in the **Discussion section** and throughout the manuscript. In addition (**Discussion, 6th paragraph**): *“Because the data discussed above support 11p15.5 LOI somatic mosaicism as a major contributor to the predisposition for BWT development, we hypothesized and confirmed that patients and survivors of BWT had higher levels of H19/ICR1 methylation detectable in peripheral blood than a cohort of healthy community control subjects and unilateral WT. Using the method of Fiala et. al.²³, we determined that 20 patients from the current study had “low-level” gain of methylation detectable in peripheral blood defined as two standard deviations above the mean H19/ICR1 beta value in the community control cohort. Seventeen of these 20 patients were female and 18/20 had 11p15.5 LOI detected in their tumor sample. Fiala et. al. previously showed that 7/7 BWT patients with “low-level” H19/ICR1 hypermethylation detectable in peripheral blood were also female.²³”*

8. Is paternal uniparental isodisomy and LOH defined as the same thing/process? Please comment on the timing of these events and how that correlates with the data presented.

Response: Thank you for this clarifying question. We have attempted to include language with more clarity in the introduction section of the manuscript to address this question. 11p15.5 LOH and 11p15.5 uniparental isodisomy are defined as synonymous terms in this study: **Introduction (2nd paragraph)**: *“Among patients with BWSp, those with epigenetic gain of methylation at H19/ICR1 (loss of imprinting - LOI) or paternal uniparental disomy (loss of genetic material from the maternal 11p15.5 locus with duplication of the paternal allele in this region; a state known as copy neutral loss of heterozygosity - LOH), both of which result in biallelic expression of IGF2, have the highest risk for any WT development.”*

The discussion section has been modified in response to this comment in an effort to focus more explicitly on the timing of 11p15.5 LOH events: **Discussion (3rd paragraph)**: *“Two synchronous BWT sample pairs in this study (SJWLM066776 and SJWLM069396) from patients with inactivating pathogenic blood germline WT1 variants demonstrated a small number of shared noncoding tumor somatic variants in addition to tumor 11p15.5 LOH (Table 2). This constellation of findings implies 11p15.5 LOH can occur in the early embryo and may be required ahead of malignant transformation on the background of WT1 mutation. The sequence of pathogenic germline WT1 mutation and subsequent somatic 11p15.5 LOH is often followed by development of activating tumor CTNNB1 variants, which were the most common somatic variants found in the current study. Our results are consistent with the temporal sequence first reported by Fukuzawa et. al. who demonstrated that WT1 variants were detected in nephrogenic rests and WT, but CTNNB1 variants were only found in the adjacent WT.⁴² WT1 and CTNNB1 variants often co-occur in WT and the spatial distribution of CTNNB1 variants has been demonstrated to exhibit intra-tumor genetic heterogeneity.^{21,43} WT driven by WT1 variants are known to exhibit stromal/rhabdomyoblastic*

differentiation and poor volumetric regression in response to neoadjuvant chemotherapy.^{44,45} Therefore, future knowledge of germline WT1 status at diagnosis in patients with BWT could guide expectations regarding volumetric tumor regression and timing of surgical resection. Taken together, these data and a recent study by Hol. et. al that demonstrated a much higher than predicted incidence of germline WT1 variants in unselected WT patients (either unilateral or bilateral, often without syndromic features), support expanded germline genetic testing at diagnosis for all patients with BWT...¹⁹

9. Figure 1 should be in the supplement.

Response: Thank you for this suggestion. Figure 1 has been relocated to the Supplement (now **Supplementary Figure 1**) and the remaining figures and supplementary figures have been re-numbered accordingly.

10. Sample acquisition section: how do the levels of LOH or LOI compare to the % tumor in a specimen? How was this accounted for/determined?

Response: Because most bilateral Wilms tumor specimens are pretreated with chemotherapy (as indicated in Figure 2) and therefore can exhibit significant necrosis or fibrosis/stromal change, we did not have the luxury of only utilizing samples with extremely high tumor purity in this study. Furthermore, because of the genetic heterogeneity described among paired synchronous bilateral Wilms tumor in this study, even different tumors within the same patient can differentially respond to treatment and exhibit different purity when samples are collected. We set a percent tumor threshold of >50% for the study, which was determined by pediatric pathologists at the time of sample inclusion and nucleic acid isolation. After nucleic acids were received, tumor purity estimates were then computed with a deconvolution-based approach using DNA methylation data as previously described (Chakravarthy, A. *et al.* Pan-cancer deconvolution of tumour composition using DNA methylation. *Nat Commun* **9**, 3220 [2018]). For tumor samples with corresponding whole genome sequencing data, we found a high positive correlation between tumor purity estimates derived from DNA methylation and whole genome sequencing data ($r=0.78$, $p=3.2e-13$).

In response to this comment, we have now reported the tumor purity estimates for each sample along with the corresponding 11p15.5 H19/ICR1 and ICR2 methylation Beta values in **Supplementary Data 5**. As we predicted, tumor purity positively correlated with ICR1 methylation Beta values [Pearson's correlation, **0.43** ($P = 8.3e-08$); Spearman's correlation, **0.66** ($P < 2.2e-16$)]; and negatively correlated with ICR2 methylation Beta values [Pearson's correlation, **-0.46** ($P = 1.2e-08$); Spearman's correlation, **-0.52** ($P = 4.6e-11$)].

We have updated the methods section in response to this comment: **Methods (1st paragraph under Methylation analysis subheading):** "Tumor purity was estimated from the methylation array data using a deconvolution-based approach as previously described.⁵⁸ These purity estimates were validated by comparison to estimates derived from whole genome sequencing data in the COG specimens."

We have updated the results section in response to this comment: **Results (1st paragraph under 11p15.5 status is shared in synchronous BWT subheading):** "Tumor purity calculated using a deconvolution-based approach from methylation array data correlated strongly with tumor purity calculated using whole

genome sequencing data in the COG specimens (Pearson $r=0.78$, $p=3.2e-13$). Tumor purity estimates and corresponding 11p15.5 H19/ICR1 and KCNQ1OT1/ICR2 methylation β values are shown in Supplementary Data 5. Tumor purity positively correlated with ICR1 methylation β values (Pearson $r=0.43$, $p=8.3e-08$) and negatively correlated with KCNQ1OT1/ICR2 methylation β values (Pearson $r=-0.46$, $p=1.2e-08$).

We have updated the limitations paragraph of the discussion section in response to this comment:

Discussion (2nd to last paragraph): *“Still, the number of paired synchronous BWT specimens was limited by inconsistent sample collection, insufficient tumor purity, availability, and heterogeneous treatment response which could have caused some tumors to be ineligible for inclusion due to necrosis, stroma, or tumor content thresholds. Our results show that tumor purity can affect measurement of methylation at 11p15.5 H19/ICR1 and ICR2; however, variations in tumor purity in pre-treated BWT will always be present due to the heterogeneity discussed above.”*

11. Figure 2: Add the paired normal samples to the top part of the figure. Where are the normal samples in the discovery cohort? What were the tumor analyses compared to in this case? Are the signatures from discovery cohort comparable to expansion cohort? Figure 2 needs more detailing with respect to sample processing and comparison.

Response: In the cohort from St. Jude Children’s Research Hospital, paired adjacent non-diseased kidney tissue was not available for analysis. Because of the high rate of bilateral nephron-sparing surgery performed for bilateral Wilms tumor patients using an enucleation technique at this center, no normal kidney is typically resected at the time of surgery. In contrast, in the COG study AREN0534, only 39% of the cohort underwent bilateral nephron-sparing surgery, and the remainder underwent radical nephroureterectomy on at least one side. Therefore, adjacent non-diseased kidney tissue was available only from the COG cohort, usually just from one kidney. “Paired normal” samples are shown in Figure 1 ($n=29$) and are termed “adjacent non-diseased kidney.” For determination of tumor somatic variants, a comparison between sequencing data from tumors and blood germline was made in both the St. Jude and COG cohorts as outlined in the **Methods section**: *“For variant discovery, a paired analysis was performed comparing tumor-derived DNA to germline DNA obtained from peripheral blood leukocytes.”*

12. Why is the discovery cohort named as such? If there is no validation cohort because the cohorts are designed differently and are showing different things, please rename and state this in the manuscript.

Response: Thank you for pointing this out. We agree that no formal *a priori* designation of discovery and validation cohorts was made. Therefore, the terms discovery and validation cohort have been eliminated from the manuscript. The cohorts are now simply referred to as St. Jude and COG cohorts. The text and **Figure 1** have been updated to reflect these changes.

13. Clinical data: how was it determined if the patients had congenital anomalies or syndromes? Please define this or state it is not known.

Response: The methods section has been revised according to this comment: **Methods (under Clinical Data subheading):** *“For the St. Jude cohort, syndromes were ascertained by thorough review of the medical record and all molecular testing documentation. For the COG cohort, the presence of genetic syndromes was a collected/reported data point from the originating studies and was delivered to the*

investigators after molecular analysis was complete.”

14. Total-strand RNA-seq – were normal sample run here? Where are the data?

Response: Thank you for calling our attention to this oversight. RNA-seq data have now been utilized/included in the manuscript in the following manner: unsupervised hierarchical clustering plot of all tumors (Supplementary Figure 9), TSNE plot of paired bilateral samples (Supplementary Figure 10), and the Cis-X allele-specific expression analysis that identified recurrent reduction to allele specific expression in *RNF185* located at chromosome 22q (Supplementary Table 2-3).

Overall, these results demonstrated that paired synchronous bilateral Wilms tumor samples clustered less robustly with one another by gene expression than methylation: **Results section (1st paragraph under subheading 11p15.5 status and genome-wide methylation/molecular identity in BWT):** *“In contrast, paired synchronous BWT samples clustered less robustly by unsupervised hierarchical clustering and TNSNE clustering of total-strand RNA-seq data (Supplementary Fig. 9-10).”*

Integrated analysis of RNA-seq and whole genome sequencing data using the Cis-X method identified a perturbation of allele specific expression due to chromosome 22 loss in samples with 11p15.5 LOI or LOH, which prompted a more in-depth investigation into chromosome 22 (detailed below): **Results section (2nd and third paragraphs under subheading 11p15.5 status and genome-wide methylation/molecular identity in BWT):** *“To further explore the suggestion that tumor 11p15.5 status correlates with a broader tumor molecular identity rather than only changes at the 11p15.5 locus itself, we determined genome-wide alterations in allele specific expression stratified by tumor 11p15.5 status (LOI, LOH, ROI) in the COG cohort using the Cis-X method, which integrates whole genome sequencing and total-strand RNA-seq data to determine the allele of origin for the RNA being expressed. Due to sample size limitations, we filtered results according to an unadjusted p-value of < 0.05 and a false-discovery rate p-value corrected for multiple testing of <0.25 for this analysis. As predicted, *KCNQ10T1* (FDR p=0.0046) and *KCNQ1* (FDR p=0.0046) were found to have perturbation of allele-specific expression in tumor samples with 11p15.5 LOH, but not ROI or LOI, while *INS-IGF2* and *IGF2* showed perturbation of allele-specific expression in both samples with 11p15.5 LOH and LOI. Five genes and/or noncoding RNAs outside the 11p15.5 region were found to have disruption in the normal patterning of allele-specific expression that correlated with tumor 11p15.5 status: *RNF185*, *SNORD116-4*, *C1QL3*, *CLEC12A*, and *NPAS2*. *RNF185*, a gene located at chromosome 22q12.2 that codes for a component of the E3 ubiquitin ligase pathway, was found to have reduction to monoallelic expression in 43.3% of tumor samples with 11p15.5 LOI (FDR p=0.240) and 35.7% of tumor samples with 11p15.5 LOH (FDR p=0.041), but 0% of samples with 11p15.5 ROI (Supplementary Table 2).*

*We then queried for possible mechanisms of *RNF185* reduction to monoallelic expression and noted recurrent chromosome 22q loss correlated with this phenomenon. Combined with the copy number data detailed above that showed recurrent chromosome 22q loss in our data set and shared chromosome 22q loss in 3 synchronous BWT pairs, we performed a focused analysis to determine the clinical and other molecular features of BWT samples with chromosome 22q loss. We found an enrichment for female biological sex, 11p15.5 LOI, and *DGCR8* RNA microprocessor mutations in this cohort of 11 tumor samples with chromosome 22q loss from 8 patients (Supplementary Table 3).”*

15. Methylation analysis – define what was used as “germline.” Blood? Kidney? Define the criteria for normal/abnormal in the results and discussion of somatic mosaicism as it differs from what is defined in this section. Were normal kidney/tumor pairs tested? Be clearer if it is tumor/tumor pairs vs tumor/normal pairs. How were methylation results from the whole genome sequencing confirmed? In the results, methylation analysis and WGS results should both be provided. Please add these data. What was done with methylation data beyond the 11p15 region?

Response: 850K MethylationEPIC beadchip analysis was performed on all germline samples including DNA derived from peripheral blood and DNA derived from adjacent non-diseased kidney. In response to this comment, samples are now more clearly identified in the **Figure 5 and 6** legends: “G1 – blood, G2 – kidney.” In addition, we have now attempted to include the word “blood” when referring to germline results throughout the manuscript.

Because 11p15.5 loss of heterozygosity is a genetic event (copy neutral loss of heterozygosity aka paternal uniparental disomy) in which the maternal allele is deleted and the paternal allele is duplicated, this phenomenon could be validated by whole genome sequencing data using the CONSERING algorithm. For 11p15.5 loss of imprinting (LOI; H19/ICR1 gain of methylation), since this is a purely epigenetic event, the phenomenon could not be validated using whole genome sequencing data. Beyond the 11p15.5 region, methylation data were used for unsupervised hierarchical clustering and TSNE clustering of samples to demonstrate a relationship between 11p15.5 methylation status and a more global pattern of methylation in samples (**Figures 4-5**).

16. Long terms survivorship analysis: please define how this analysis was done more clearly as the discussion focuses on variations that are within the normal range per the parameters set.

Response: The methods section of the manuscript has been updated in response to this comment: **Methods (final paragraph under Long-term survivorship cohort analysis from peripheral blood DNA subheading):** “Blood-derived Infinium MethylationEPIC Beadchip (850K) array germline DNA methylation data from the 61 patients in the current study were combined with blood-derived germline DNA methylation data from 282 healthy community controls and 171 long-term Wilms tumor survivors (>5 years from cancer diagnosis; n=154 unilateral, n=17 bilateral) from the St. Jude Lifetime Cohort Study (SJLIFE).³⁵ These data were normalized and processed with the subset-quantile within array normalization (SWAN) method using the R package minifi. The average β values from the 11p15.5 H19/ICR1 and KCNQ1OT1/ICR2 regions defined above were computed as detailed above. Because of known changes in leukocyte global DNA methylation that occur with age, methylation at 11p15.5 H19/ICR1 was plotted according to age.⁶⁴ Using these data, the relationship between age and methylation at 11p15.5 H19/ICR1 was determined using a linear regression model. The relationship between age and methylation from unilateral WT and BWT samples were compared against the learned model from the healthy community control population.”

The limitations paragraph of the discussion section has been updated in response to this comment: **Discussion (2nd to last paragraph):** “Finally, all data in this manuscript reporting evidence of mosaic 11p15.5 H19/ICR1 gain of methylation detectable in peripheral blood are based on statistical comparison or observed changes in comparison to a normal cohort, while no patients in this manuscript exhibited frank

hypermethylation above the established threshold of a β value of 0.7 at 11p15.5 H19/ICR1 detectable in DNA obtained from peripheral blood.”

17. How do the different or similar CTNNB1 mutations fit into the timing of the 11p15 alterations discussion?

Response: Different single nucleotide variants in paired synchronous bilateral Wilms tumors from the same patient that result in activating exon 3 Beta catenin mutations is suggestive of convergent evolution toward Wnt pathway activation of independent cellular clones that become distinct tumors. This is outlined in the results section: **Results (1st paragraph under Germline variant analysis from blood and associated tumor findings subheading):** *“Among 10 total samples harboring blood germline WT1 variants, somatic tumor 11p15.5 LOH, and somatic tumor CTNNB1 variants, there were three sets of paired synchronous BWT (SJWLM066776, 066780, 051028) in which each of the paired tumors had somatic exon 3 CTNNB1 variants. In two cases, the tumor CTNNB1 variants were distinct (SJWLM066776 CTNNB1 p.T41A vs. p.S45del, SJWLM051028 CTNNB1 p.S45P vs. p.S45del) and in the remaining case (SJWLM066780) the CTNNB1 p.S45F variant was shared in both paired tumors (Fig. 2).”*

The different CTNNB1 mutations constitute independent genetic events but are convergent evolution toward Wnt pathway activation; this suspected process and the timing of events is shown in the top panel of **Figure 6**.

18. Clarify how many WT1 patients also had 11p15 alterations.

Response: Out of 14 tumor samples from the 9 patients with germline WT1 variants in the current study, all 14 (100%) were found to have 11p15.5 LOH. Thus, in the current study all tumor samples from patients with pathogenic germline variants in WT1 also had 11p15.5 LOH.

The results section has been updated in response to this comment: **Results (1st paragraph under Germline variant analysis from blood and associated tumor findings subheading):** *“For 14 tumor samples from 9 patients with germline WT1 variants, 11p15.5 LOH (paternal uniparental disomy) determined by methylation analysis and/or the CONSERGING algorithm was present in all tumors (Figure 2).”*

19. Clarify the parameters of patients with 11p15 alterations in multiple tissues and their clinical phenotypes, specifically if there were subtle clinical phenotypes.

Response: The results section has been updated to address this question: **Results (2nd paragraph under 11p15.5 status is shared in synchronous BWT subheading):** *“Most patients with 11p15.5 LOH or 11p15.5 LOI detected in their tumor samples had no clinical features suggestive of a syndrome (Fig. 2). However, of the 9 patients with definitive 11p15.5 alterations in multiple tissues, SJWLM066781 had 11p15.5 LOI in their tumor and adjacent non-diseased kidney and had a ureteral duplication. SJWLM069381 had 11p15.5 LOI in their tumor and adjacent non-diseased kidney tissue and had hemihypertrophy. SJWLM069390 had 11p15 LOH detected in their tumor and adjacent non-diseased*

kidney tissue. In addition, this patient was found to have mosaic 11p15.5 LOH in their blood as determined by whole genome sequencing (Supplementary Fig. 2). This patient was noted to have a BWSp clinical phenotype.”

20. Figure 3 is too small to read and blurry.

Response: A high-resolution version of this figure (now **Figure 2**) has now been provided to the journal with the resubmitted manuscript. I believe the figure becomes blurry when it is incorporated into the reviewer *.pdf.

21. Line 381-388 - For section, “Somatic variants are not shared in synchronous BWT”, the authors need to specify how many genes were used for pathway analysis. The Supplementary Table 1 shows 4 clusters. How were these clusters derived? Which genes were included in each set? Please discuss the method and results in greater details. How does the RNA data correlate with the methylation data for the same regions?

Response: The purpose of the pathway analysis was to determine the functional cellular pathways that were affected by somatic variants detected in bilateral Wilms tumor samples. The pathway analysis was performed using the DAVID Bioinformatics Database/Resource (<https://david.ncifcrf.gov/home.jsp>). The input for the DAVID pathway analysis consisted of the entire list of genes (with counts) determined to exhibit somatic mutations in the bilateral Wilms tumor samples as outlined in **Supplementary Data 2**. The functional annotation clustering is part of the DAVID algorithm and classifies highly related genes into functionally related groups. RNA and methylation data were not utilized for this analysis.

In response to this comment, former Supplementary Table 1 has now been provided as **Supplementary Data 4** so the reader can click on the gene lists in the excel spreadsheet to determine the number and names of each highly-related gene that is a member of the annotated lists. For example, the WIKIPATHWAYS WP2338~miRNA biogenesis gene set includes the genes ENSG00000124571 (*XPO5*), ENSG00000113360 (*DROSHA*), ENSG00000100697 (*DICER1*), and ENSG00000128191 (*DGCR8*).

22. For section, “Somatic variants are not shared in synchronous BWT”, based on Supplementary figure 3, chromosome 22 also showed variations. Please comment on this.

Response: Thank you for this critical observation, which was previously unappreciated but upon consideration of this comment has tied several pieces of data within the manuscript together.

The results section has been modified in response to this comment/observation: **Results (3rd paragraph under Tumor somatic variants are not shared in synchronous BWT subheading):** *“Comparison of tumor copy number variants showed markedly different genome wide copy number profiles in paired synchronous BWT except for SJWLM069391 (Supplementary Fig. 4-5). In addition to SJWLM069391, similar regions of chromosome 22q loss were noted in paired synchronous BWT SJWLM069399 and SJWLM066784 (Supplementary Fig. 5). Overall, chromosome 22q copy number loss was detected in 11/85 (12.9%) of BWT samples.”*

This observation about recurrent chromosome 22q copy loss tied into an analysis of allele specific expression which we have now included in the manuscript:

Methods section (Total-strand RNA-seq subheading): *“Integrated analysis of whole genome sequencing and total-strand RNA-seq data was performed using the previously described Cis-X method to determine genome wide allele-specific expression patterns in BWT.²⁹”*

Results section (2nd and 3rd paragraphs under 11p15.5 status and genome-wide methylation/molecular identity in BWT subheading): *“To further explore the suggestion that tumor 11p15.5 status correlates with a broader tumor molecular identity rather than only changes at the 11p15.5 locus itself, we determined genome-wide alterations in allele specific expression stratified by tumor 11p15.5 status (LOI, LOH, ROI) in the COG cohort using the Cis-X method, which integrates whole genome sequencing and total-strand RNA-seq data to determine the allele of origin for the RNA being expressed. Due to sample size limitations, we filtered results according to an unadjusted p-value of < 0.05 and a false-discovery rate p-value corrected for multiple testing of <0.25 for this analysis. As predicted, KCNQ1OT1 (FDR p=0.0046) and KCNQ1 (FDR p=0.0046) were found to have perturbation of allele-specific expression in tumor samples with 11p15.5 LOH, but not ROI or LOI, while INS-IGF2 and IGF2 showed perturbation of allele-specific expression in both samples with 11p15.5 LOH and LOI. Five genes and/or noncoding RNAs outside the 11p15.5 region were found to have disruption in the normal patterning of allele-specific expression that correlated with tumor 11p15.5 status: RNF185, SNORD116-4, C1QL3, CLEC12A, and NPAS2. RNF185, a gene located at chromosome 22q12.2 that codes for a component of the E3 ubiquitin ligase pathway, was found to have reduction to monoallelic expression in 43.3% of tumor samples with 11p15.5 LOI (FDR p=0.240) and 35.7% of tumor samples with 11p15.5 LOH (FDR p=0.041), but 0% of samples with 11p15.5 ROI (Supplementary Table 2).*

We then queried for possible mechanisms of RNF185 reduction to monoallelic expression and noted recurrent chromosome 22q loss correlated with this phenomenon. Combined with the copy number data detailed above that showed recurrent chromosome 22q loss in our data set and shared chromosome 22q loss in 3 synchronous BWT pairs, we performed a focused analysis to determine the clinical and other molecular features of BWT samples with chromosome 22q loss. We found an enrichment for female biological sex, 11p15.5 LOI, and DGCR8 RNA microprocessor mutations in this cohort of 11 tumor samples with chromosome 22q loss from 8 patients (Supplementary Table 3).”

The discussion section has been updated to include discussion of data related to this comment:

Discussion (9th paragraph): *“Specifically, our study showed that BWT with 11p15.5 LOI and LOH commonly exhibited reduction to monoallelic expression of the ubiquitin-ligase associated gene RNF185 located at chromosome 22q12.2. The most common mechanism for RNF185 reduction to monoallelic expression was chromosome 22q copy loss. All tumors that exhibited copy loss at 22q had also had 11p15.5 LOI and 6 of these 8 patients were female. This group also included all 4 patients with DGCR8 microprocessor hotspot p.E518K mutations in the current study and included 5 patients with high-risk SIOF post-treatment histology. Allelic loss at chromosome 22q loss has been previously associated with high-risk WT.^{48,49} Furthermore, in depth analysis of two cases of paired WT and precursor nephrogenic rests has implicated loss of chromosome 22 in the progression from perilobar nephrogenic rests to WT.⁵⁰ DGCR8 (located at chromosome 22q11.21) mutations were shown to have an extreme female predominance in previous WT studies: Wegert et. al. found that 23/26 (88.5%) and Walz et. al. found that 15/17 (88.2%) patients with somatic DGCR8 mutations were female.^{32,51} The possible functional significance of RNF185 in WT and the constellation of somatic mosaic 11p15.5 LOI, female biological sex, chromosome 22q loss, and DGCR8 microprocessor mutations in the development and progression of WT will be the subjects of future investigation.”*

Figure 6 has been modified to include chromosome 22q loss and microRNA processing gene mutations as key events in bilateral Wilms tumor development.

23. Figure 4 is confusing and does not add to the explanation of the data. Please make a table instead and add blood status of 11p15 as well.

Response: Figure 4 has been made into **Table 1**. Blood status of 11p5.5 (whether H19/ICR1 methylation in blood was greater than 2 standard deviations above the healthy control cohort mean value) has now been added to **Table 1**.

24. The authors mention in the method section that samples with average beta value H19/ICR1 <0.7 have normal ROI. Figure 5A,D show that all the samples from all three cohorts have normal ROI. Why did the authors make an age dependent methylation comparison of ICR1? Are the results within the normal range or outside or just demonstrating assay or sample variability? Specifically define LOI, LOH and ROI for this figure and how it is similar or different to the ranges for the other data presented. If 11p15.5 status is shared in BWT: What is the 11p15.5 status in the normal tissue? The “established thresholds” are not well defined in the context of the somatic analysis, please clarify. Please clarify why the extent of LOH did not overlap in one case. Was that between tumors, between tumor and kidney or blood? Was this LOH or paternal uniparental isodisomy?

Response:

Thank you for this important observation and the chance to clarify our hypothesis, results, and conclusions: **Results (5th paragraph under 11p15.5 status is shared in synchronous BWT subheading):** *“To validate the finding of somatic mosaicism for 11p15.5 LOI detectable in peripheral blood on the cohort level, we combined identically processed and normalized H19/ICR1 methylation β values from leukocyte-derived DNA between our current BWT cohort (n=61) and a cohort of healthy community controls (n=282) and WT cancer long-term survivors (including survivors of both unilateral [n=154] and BWT [n=17]) from the St. Jude Life Cohort Study.³⁵ We hypothesized that H19/ICR1 values in peripheral blood DNA would be higher in BWT patients than healthy community controls and potentially unilateral WT patients. The peripheral blood 11p15.5 H19/ICR1 and KCNQ10T1/ICR2 β values were determined to be normally distributed in all groups using the Kologov-Smirnov test. The mean H19/ICR1 peripheral blood methylation β value in the healthy community control cohort was 0.499 ± 0.0248 (mean \pm standard deviation). The mean H19/ICR1 peripheral blood methylation β value from the BWT cohort (0.534 ± 0.0298) was higher than both the unilateral WT (0.521 ± 0.0258 ; $p < 0.0001$) and healthy community control cohorts ($p < 0.0001$; Fig 3D.) Using the method of Fiala, et. al.²³, we determined that 26 BWT patients (n=20 from current study; n=6 from St. Jude Life cohort) had a H19/ICR1 methylation β value greater than two standard deviations from the mean community control cohort β value (i.e., H19/ICR1 $\beta > 0.54864$; Fig. 2; Supplementary Table 1). Of these 20 patients, 17/20 (85%) were female, 18/20 (90%) had 11p15.5 LOI in their tumor, 2/20 (10%) had 11p15.5 LOH in their tumor, and 0 had 11p15.5 ROI in their tumor (Supplementary Table 1).”*

Leukocyte DNA methylation is known to change with age (Terry, M.B., Delgado-Cruzata, L., Vin-Raviv, N., Wu, H.C. & Santella, R.M. DNA methylation in white blood cells: association with risk factors in epidemiologic studies. *Epigenetics* **6**, 828-37 (2011). Because of the wide range of ages included in the survivor analysis, we wanted to provide a control for DNA methylation changes according to age. In doing

so, we noted that bilateral survivors and patients had a strikingly different pattern in leukocyte DNA methylation at 11p15 ICR1 than unilateral patients and healthy community controls as depicted in Figure 4. The methods section of the manuscript has been updated to reflect this comment: **Methods (under Long-term survivorship cohort analysis from peripheral blood DNA subheading):** *“Because of known changes in leukocyte global DNA methylation that occur with age, methylation at 11p15.5 H19/ICR1 was plotted according to age.⁴¹ The relationship between age and methylation from unilateral WT and BWT samples were compared against the learned model from the healthy community control population.”*

Results (6th paragraph under 11p15.5 status is shared in synchronous BWT subheading): *“However, because leukocyte DNA methylation is known to change with age³⁶ and the healthy control population with peripheral blood DNA available for this study was derived from cancer survivors of a higher median age than our current study population, we wanted to assess H19/ICR1 methylation as a function of age. We noted an inverse relationship between methylation at H19/ICR1 and increasing age in healthy community controls (Fig. 3E). In contrast, we detected positive correlation between increasing age and H19/ICR1 methylation in BWT patients and long-term survivors. This positive correlation was not seen in long-term survivors of unilateral WT (Fig. 3E).”*

The limitations paragraph of the discussion section has been updated in response to this comment: **Discussion (2nd to last paragraph):** *“Finally, all data in this manuscript reporting evidence of mosaic 11p15.5 H19/ICR1 gain of methylation detectable in peripheral blood are based on statistical comparison or observed changes in comparison to a normal cohort, while no patients in this manuscript exhibited frank hypermethylation above the established threshold of a β value of 0.7 at 11p15.5 H19/ICR1 detectable in DNA obtained from peripheral blood.”*

The **Results** section has also been updated in response to this comment: *“Among 19 tumors with 11p15.5 LOH determined by whole genome sequencing, the extent of 11p cnLOH overlapped both the 11p15.5 and WT1/11p13 loci in 18/19 cases (94.7%; Supplementary Figure 7). Among paired synchronous BWT from patients with pathogenic germline WT1 mutations and 11p15.5 LOH detected in each of their tumors, the extent of 11p LOH was not identical when the two tumor samples were compared. The differential extent of 11p LOH between paired synchronous BWT suggests that 11p LOH occurs as an independent genetic event in each tumor in most cases rather than having a shared clonal origin (Supplementary Figure 7).”*

25. The authors propose that post-zygotic 11.p15.5 alterations lead to development of BWT. However, Figure 6 shows that 11.p15.5 alterations are responsible for both unilateral and BWT. Lines 487-505 state molecular signatures that identify BWT. The authors need to state these molecular identifiers clearly. In addition, authors state, “We noted that BWT predominantly clustered distinctly from unilateral WT and closer to non-diseased adjacent kidney tissue, consistent with previously published results (Figure 6).” However, the published results state that there were no consistent differences in DNA methylation between unilateral and bilateral tumors. Please provide a consistent explanation between these results and the published results, given shared authorship with this manuscript. What is the threshold for defining a genetic/epigenetic syndrome in these patients?

Response: Thank you for this observation. 11p15.5 alterations are certainly present in both unilateral and bilateral WT. However, the emphasis of the current manuscript is that 11p15.5 is often somatic mosaic in patients who develop bilateral Wilms tumor and therefore describes a mechanism by which both kidneys are predisposed to tumor development.

The publication the reviewer refers to (Brzezinski, J. *et al.* Clinically and biologically relevant subgroups of Wilms tumour defined by genomic and epigenomic analyses. *Br J Cancer* **124**, 437-446 [2021].) demonstrates two Wilms tumor subgroups (cluster A and B) according to global methylation patterns determined by the Infinium HumanMethylation450 BeadChip (Illumina) or the Infinium MethylationEPIC BeadChip (Illumina). They demonstrated that bilateral disease was significantly more prevalent in Subgroup A. Their combined results from the discovery and validation cohort analyses showed that subgroup A (n=22 total tumors) was 68% bilateral and subgroup B (n=62 total tumors) was 26% bilateral ($p=0.0007$; shown in Table 1 of the manuscript). Subgroup A (which was predominantly bilateral tumors) clustered closer to normal kidney than Subgroup B. As the reviewer notes, the study found that there were no differences between unilateral and bilateral samples according to methylation within each of these two clusters (e.g. there were no differences in methylation between unilateral and bilateral tumors within subgroup A). However, since the primary cluster differentiation among samples was enriched for bilateral Wilms tumors in one of the groups, essentially bilateral and unilateral Wilms tumors are clustering different from one another.

This is very consistent with results from our current study depicted in **Figure 4 (unsupervised hierarchical clustering of samples according to the top 10,000 most variable probes in the methylation dataset)**– which show two clusters among Wilms tumor samples (clusters 3 and 4 in the Figure). Cluster 3 is enriched for bilateral tumors and clusters closer to adjacent non-diseased kidney tissue (cluster 2), while cluster 4 is enriched for unilateral tumors. However, you can see within each cluster that there are both unilateral and bilateral tumors.

26. Figure 6 displays 4 tissue types in study clustered separately. Authors need to show the molecular identities of each tissue. In addition, show these distinct clusters in Supplementary Figure 7. For Figure 6: “sex” not “gender” should be reported. Why are metastases included in this Figure and the analysis?

Response: The term “gender” has now been replaced with “sex” in **Figure 4** (former Figure 6) and in all figures and supplementary Figures in the manuscript. The tissue type has been clarified in the **Figure 4** legend (Blood, Non-diseased kidney, Wilms tumor). The metastatic samples have now been eliminated from the analysis and this figure, and the analysis and figure for the TSNE clustering analysis.

Supplementary Figure 8 (former Supplementary Figure 7) contains only paired synchronous bilateral tumor tissues oriented in a Spearman correlation matrix from most to least similar according to methylation. Because it contains only paired tumors, no unilateral tumors, and no “normal tissues” it could not be clustered in similar manner to Figure 4.

27. Figure 7: If tumor purity was initially defined as greater than 50% and then >80% it is unclear what samples throughout the paper have what levels of tumors and that influences the results. There needs to be clear assessment of that in each data set and each sample. Perhaps the sequencing data can be used as a proxy for this. Given that somatic levels of 11p15 changes are being discussed this needs to be thoroughly clarified throughout the manuscript.

Response: Please see above response to comment 10 regarding tumor purity and accompanying changes to the manuscript.

28. The authors introduced pathogenic or likely pathogenic variants in the abstract. Figure 8 shows role of pathogenic variants in prezygotic stage. The authors need to add the predisposing variants to section of “3. somatic mosaic 11p15.5 loss of imprinting” based on findings in Table 1. Also please add this to the discussion.

Response: Panel 2 of now **Figure 6** addresses the pathogenic germline variants in the pre-zygotic stage. The possibility of 11p15.5 loss of imprinting can also occur in patients who have pathogenic germline variants. As the reviewer pointed out, Table 1 also illustrates this possibility, although it is less frequent than 11p15.5 LOH or 11p15.5 ROI in patients with germline variants. The discussion section has been updated to reflect this comment: **Discussion (3rd paragraph):** *“It should also be noted that tumors from patients with pre-zygotic variants in Wilms tumor or cancer predisposition genes can also exhibit 11p15.5 LOI, but this is more rare than 11p15.5 LOH or ROI in this group.”*

29. In the discussion is the mechanism germline mutation plus LOH? If so, state that in the first paragraph. Does LOH occur first or after the germline mutations? How does LOH affect non-chromosome 11 germline changes?

Response: Our proposed mechanism as outlined in Figure 7 (the best example being in the context of germline *WT1* pathogenic variants) is that germline mutations occur before 11p15.5 LOH. This is outlined in the **2nd and 3rd paragraphs in the Discussion section**. The first paragraph of the discussion section has been updated in response to this comment: **Discussion section (1st paragraph):** *“(1) Pre-zygotic germline genetic variants readily detectable in DNA derived from peripheral blood (*WT1*, *NYNRIN*, *TRIM28*, *BRCA* complex genes) often followed by 11p15.5 LOH...”*

30. How do the blood 11p15 findings factor into the developmental model?

Response: Prior work from Coorens et. al. suggested that mosaic 11p15.5 alterations occurred in the early embryo and were enriched in mesodermal tissues, specifically the bilateral kidneys. These carefully conducted studies were performed on a multitude of tissues, but from a limited number of patients. The authors concluded that failure to detect the 11p15.5 alterations in the blood was supportive of mosaicism. We wholeheartedly agree with the findings from these studies. However, because the blood cells are also derived from the mesodermal germ layer during embryonic development, we sought additional evidence on the cohort level to support the concept of mosaicism. We reasoned that since blood cells are originally derived from the mesodermal layer that we should also be able to detect such alterations using DNA isolated from peripheral blood leukocytes. On a population/study cohort level, we did identify such alterations that suggest somatic mosaicism, although it is exceedingly difficult to assign a label of “somatic mosaicism” to an individual case. Our results document that such alterations are likely present in peripheral blood of bilateral Wilms tumor patients and that 20 patients from the current study exhibited “low-level” H19/ICR1 hypermethylation defined by Fiala et. al as greater than two standard deviations above the mean methylation value at this locus from a normal control cohort. We attempted to perform a single-cell methylation analysis using banked peripheral blood to document this phenomenon more completely, but the existing technology did not permit such an approach and this will have to remain a future direction of our work.

31. Why does 11p15 alteration level increase with age?

Response: We hypothesize that progressive, clonal selection of blood cells/leukocytes with 11p15.5 alterations in patients with somatic mosaicism for 11p15.5 loss of imprinting leads to a higher measured H19/ICR1 methylation beta value in older subjects. This was not found in community controls. In the kidney this phenomenon has been termed clonal nephrogenesis but would be difficult to measure longitudinally. In the bloodstream, prior literature has noted that such a process may be responsible for late-onset beta thalassemia as noted in the discussion section: Discussion (7th paragraph): *“Progressive selection of mosaic hematopoietic cells containing uniparental paternal isodisomy for 11p15 has been suggested as a mechanism for late onset β -thalassemia major in patients who are heterozygous carriers of pathogenic HBB variants (which is also located at 11p15); however, evidence for this phenomenon is limited overall.”*⁴⁷

32. How do different pathologic findings in these tumors fit into the two BWT models proposed? Addition of this pathology data would be informative.

Response: The current study showed that 15/17 (88.2%) specimens with SIOP high-risk post-treatment histology (defined as blastemal predominance or diffuse anaplasia after neoadjuvant chemotherapy treatment) had 11p15.5 LOI. This observation leads to our future hypothesis that tumors with 11p15.5 LOI may have inferior radiographic response rates or oncologic outcomes in patients with BWT. This is noted as a future direction in the **Discussion** section of the manuscript.

33. How does “clonal nephrogenesis” result in BWT? Developmentally when does this clonality occur?

Response: Progressive selection and expansion of nephrogenic clones with normal histology and H19/ICR1 gain of methylation provides predisposition for BWT development. However, additional hits (such as loss of chromosome 22q, microRNA processing gene mutations) then occur before tumors develop. Our evidence provides support from this original clonality tracing back to the early embryo, but then subsequent expansion of these nephrogenic clones is thought to occur during kidney development.

34. How do the changes in long term survivors of the 11p15 region validate somatic mosaic findings in the BWT patients? It is unclear how the data presented support this statement.

Response: The Discussion section has been reworked in response to this comment: **Discussion (6-8th paragraphs):** *“Identification of mosaic 11p15.5 LOI in the peripheral blood is difficult to definitively determine in an individual patient sample. Therefore, we sought to evaluate for supporting evidence of 11p15.5 H19/ICR1 hypermethylation (LOI) on the cohort level. Indeed, our analysis showed a statistically significant increase in H19/ICR1 methylation in peripheral blood from patients with 11p15.5 LOI in their tumor specimens when compared to those with 11p15.5 ROI in their tumors, further supporting the concept of epigenetic somatic mosaicism with detection of low-level gain of methylation. Because the data discussed above support 11p15.5 LOI somatic mosaicism as a major contributor to the predisposition for BWT development, we hypothesized and confirmed that patients and survivors of BWT had higher levels of H19/ICR1 methylation detectable in peripheral blood than a cohort of healthy community control subjects and unilateral WT. Using the method of Fiala et. al.²³, we determined that 20 patients from the current study had “low-level” gain of methylation detectable in peripheral blood defined as two standard deviations above the mean H19/ICR1 beta value in the community control cohort. Seventeen of these 20 patients were female and 18/20 had 11p15.5 LOI detected in their tumor sample. Fiala et. al. previously*

showed that 7/7 BWT patients with “low-level” H19/ICR1 hypermethylation detectable in peripheral blood were also female.²³

Because we incorporated DNA samples from a long-term survivorship cohort and because global leukocyte DNA methylation changes with age, we started this analysis by plotting peripheral blood H19/ICR1 beta values versus age. We found a positive correlation between increased age and H19/ICR1 methylation that was specific to BWT patients and long-term survivors and not found in healthy community controls or unilateral WT patients or long-term survivors. We speculate that increased methylation with age at 11p15.5 H19/ICR1 in the BWT population could be due to gradual selection/expansion of cellular clones with H19/ICR1 hypermethylation in the peripheral blood over time in a manner similar to what was described as clonal nephrogenesis in the kidney.²² Progressive selection of mosaic hematopoietic cells containing uniparental paternal isodisomy for 11p15 has been suggested as a mechanism for late onset β -thalassemia major in patients who are heterozygous carriers of pathogenic HBB variants (which is also located at 11p15); however, evidence for this phenomenon is limited overall.⁵¹ These results are very unlikely to be related to circulating tumor DNA because the positive correlation between age and H19/ICR1 methylation was also seen in BWT long-term survivors, whose DNA samples were obtained at least 5 years after cancer diagnosis. These results provide further support for the 11p15.5 LOI mosaicism hypothesis because they provide indirect evidence of expanding 11p15.5 LOI mosaic clones in peripheral blood with time.

Furthermore, our data show that 11p15.5 status can be used as a surrogate biomarker for more global methylation patterns/molecular subgroups of BWT because when samples were clustered according to broader patterns of methylation, they tended to cluster according to 11p15.5 status. Whether 11p15.5 status (ROI, LOI, LOH) correlates with volumetric or histologic response to neoadjuvant chemotherapy, event-free, or overall survival will be the subject of future clinical translational investigation. However, as a preliminary window into this question, the current study showed that 15/17 (88.2%) specimens with SIOP high-risk post-treatment histology and 15/17 (88.2%) with the adverse prognostic biomarker 1q gain had 11p15.5 LOI. The current study provides strong rationale for prospectively following outcomes in BWT patients according to tumor 11p15.5 status and this will be included as an observational biologic aim in the Children’s Oncology Group BWT protocol currently under development.”

35. The data on amplified clonal mosaicism increasing over time is limited, where the data on mosaic 11p15 changes in patients with Beckwith-Wiedemann Spectrum is more plentiful. Please comment on this in the context of the discussion.

Response: We agree with this comment. We have added this observation to the the discussion section: **Discussion (7th paragraph):** “Progressive selection of mosaic hematopoietic cells containing uniparental paternal isodisomy for 11p15 has been suggested as a mechanism for late onset β -thalassemia major in patients who are heterozygous carriers of pathogenic HBB variants (which is also located at 11p15); however, evidence for this phenomenon is limited overall.⁴⁷”

36. How is 11p15 status a biomarker for global methylation patterns? Please revise this and provide the evidence to support that revision. Was it high or low stage risk? Both are stated?

Response: When tumors are clustered according to methylation M values derived from the top 10,000 most variable CpG probes in the methylation array data set (using both unsupervised hierarchical or TSNE approaches), the bilateral Wilms tumors illustrate clustering according to 11p15 status as depicted in **Figure 4 and Figure 5B**. We do not understand if this pertains to risk or stage of tumors; however, among tumors with SIOP high-risk histology (which is used clinically to define treatment and prognosis), the majority had 11p15 LOI. This association will need to be explored prospectively in future studies.

The comment about 11p15 status determining outcome in low-risk tumors may be confusing in this context. This observation came from a study looking specifically at unilateral Wilms tumor with several low-risk parameters (age <2 years, tumor weight < 500g, stage I, favorable histology). It has been eliminated from the **Discussion section** to improve clarity of the manuscript and avoid confusion.

Minor comments –

1. Reference which section of a figure is being discussed each time in the text, not just the overall figure number.

Response: The Figure references have now been updated to include sections/panels within each figure where applicable.

2. Line 105-109 - In the introduction, the authors need to include Beckwith-Wiedemann syndrome in the predisposition syndromes for Wilms tumor.

Response: Beckwith-Wiedemann syndrome is listed as a predisposition syndrome in the introduction section of the manuscript: **Introduction (2nd paragraph):** *“In addition, BWT has an increased predisposition in patients with Beckwith Wiedemann spectrum disorder (BWSp), implicating dysregulation of imprinting at chromosome 11p15.5, a region which houses a cluster of imprinted genes including the growth factor IGF2.^{11-13”}*

3. In Supplementary Figure 1 legend, the authors need to add tumor information with respect to complete LOH.

Response: Supplementary Figure 1 is now **Supplementary Figure 2**. Information regarding tumor LOH has been added to the Figure legend: *“Complete 11p15.5 loss of heterozygosity (B allele frequency of 1.0) is observed in DNA derived from adjacent non-diseased kidney and Wilms tumor in the same patient. The region of 11p loss of heterozygosity in both the adjacent non diseased kidney and Wilms tumor overlaps the IGF2 locus at 11p15.5 and the WT1 locus at 11p13.”*

4. The authors need to improve the resolution of Supplementary Figure 2 and add scale bar to it.

Response: Supplementary Figure 2 is now Supplementary Figure 3. It has now been replaced with a higher resolution version that contains labeled scale bars (1 mm for low power, 200 um for high power) on each image.

5. Line 362-388 - The authors need to rearrange the section, “Somatic variants are not shared in synchronous BWT” such that somatic variants results are stated together and copy number variants results are stated together.

Response: We have rearranged the section per the reviewer suggestion. Now somatic variants are outlined in the first two paragraphs of the section and copy number variants are outlined in the third paragraph.

6. Supplementary Figure 3 displays a cross and asterisks. Please add the details for these in the figure legend.

Response: A legend depicting the * and X has been added to now added to both panels of Supplementary Figure 4.

7. Line 531-538 - Please add germline type abbreviation details for figure 7 legend.

Response: The figure legends for now figures 4-5 have been updated to include germline type (blood or kidney).

Reviewer #3, expertise in Wilms tumour genetics (Remarks to the Author):

After spending over years in the Wilms tumor field I’m still surprised by how difficult the answer to the apparently simple question ‘what is a Wilms tumor, where is it coming from and what caused it’ turns out to be. In this manuscript, Murphy and colleagues focus on bilateral Wilms tumors which by their nature can provide unique insights in the causes and origins of these tumors. They perform a combined genomic and epigenomic analysis of BWTs from 68 patients. The nature of such a dataset itself makes this work a valuable addition to our understanding of this disease and to the literature.

Response: We appreciate your comments and review of our manuscript.

Two important points before I give my opinion on this manuscript. First, I’m myself not an expert in the techniques and analyses used in this work, and how the primary data lead to the results as presented. I will simply assume this is all correct hoping other reviewers have more to say on this. Instead I will focus on the question if the presented data indeed leads to the conclusions as given, how this fits with the existing body of literature and what new insights we can gain from this.

Second, these sort of manuscripts and papers are, due to the amount of data generated and the limitations placed on a manuscript size by the journals, never an easy read. This manuscript I find extremely hard work, which is maybe to be expected given the number of germline and somatic mutations and possible 11p15 changes. If (some of) my comments are caused by my misunderstanding the intended meaning of the text I apologize to the authors. I certainly would not be able to write this sort of manuscript better.

Response: We appreciate your comments, thoughts, and review of our manuscript.

The manuscript roughly divides the group of BWTs in two: those with germline mutations (often combined with 11p15 LOH) and those without germline mutations (and instead showing 11p15 LOI). A variety of germline-mutated genes is identified, all of which have been linked to Wilms tumors though I think not necessarily as germline mutations (I might be wrong here). Mutations in WT1 were found to be most common in these cases. In these cases I miss (in my opinion important) details on the loss of the wild type allele, the second hit from the Knudson two-hit model. First, what sort of mutation is responsible for this? Figure 8 presents a model where the germline mutation is on the paternal allele and LOH / paternal uniparental disomy on 11p15 is responsible for this loss of the wild type WT1 allele. If the germline mutation is indeed in the paternal allele this makes perfect sense (and has been suggested before), but I cannot find any data on whether this paternal or maternal mutant WT1 mutation has been tested. Is in the cases described here the mutant allele indeed the paternal allele? Without this, or if there are also cases with a maternal mutant copy, the model should not be presented as it is at the moment.

Response: In the case of *WT1* mutations, the 2nd hit is a copy neutral loss of heterozygosity event (deletion of the maternal allele and duplication of the paternal allele) with breakpoints that encompass the *WT1* locus at 11p13 and the 11p15.5 locus. This results in biallelic expression of *IGF2* and biallelic/homozygous knockout of *WT1*. This mechanism is detailed in the top panel of **Figure 6**. Furthermore, in response to comments from this reviewer, we have now included **Supplementary Data 1**, which details which blood germline variants exhibit LOH/increases in variant allele frequency in the accompanied tumor tissue.

Regarding the germline *WT1* mutations being on the paternal allele, we are inferring this based on the methylation array data. Parental DNA is not available for definitive confirmation in this study. This can only be inferred without parental DNA. However, the methylation characteristics of the paternal versus maternal allele at the 11p15.5 region are among the best characterized methylation differences that account for genetic imprinting at any chromosomal locus and thus we feel comfortable asserting this inference based on our data. Although it is not possible to definitively determine the parental allele of origin for the pathogenic germline *WT1* variants in this study due to sample availability, we will make this a priority in future work. The discussion section of the manuscript has been updated in response to this comment: **Discussion (2nd paragraph):** *“Therefore, it can be inferred that the germline genetic variants that lead to BWT predisposition occur on the paternal allele and become homozygous in the tumor due to copy neutral LOH events at 11p13-11p15.5 loci that establish paternal uniparental disomy.”*

For me the biggest surprise in this part of the data was the observation of *BRCA1*-*PALB2*-*BRCA2* germline mutations. As the authors mention, these germline mutations are usually found in families with (mainly) hereditary breast and ovarian cancer. Could the authors comment on whether these malignancies were also found in these families, and what might make the differences that in these few families BWTs are found whereas in most of families with these mutations do not have these?

Response: As noted in response to the comment above, parental DNA was not available for this study, and thus we cannot definitively conclude whether these germline mutations were inherited or acquired *de novo*. Family history information was not available from patient samples in the COG cohort. Because the three patients with *BRCA1*, *BRCA2*, or *PALB2* germline variants were from the COG cohort, we are

unable to ascertain any family history associated with these cases. This has been noted as an additional limitation of the manuscript: **Discussion (2nd to last paragraph):** *“Family history information was not available from the COG cohort and thus we could not determine if the patients who had germline variants in BRCA1, BRCA2, or PALB2 had a family history of breast or ovarian cancer.”*

Next, the co-occurrence of WT1 mutations with CTNNB1 mutations (especially affecting Ser45) is discussed. As the authors write, this is a confirmation of data that has been described several times before, and this is therefore not new but a useful confirmation nonetheless.

Response: Thank you for this comment. We agree that, although described in multiple prior publications focusing on Wilms tumor overall, noting that there is co-occurrence of *WT1* and *CTNNB1* pathogenic variants in patients with bilateral Wilms tumor (in whom the original *WT1* alterations are all germline rather than somatic) is an important observation to highlight in the manuscript. It establishes that the *WT1* mutations are happening first; although this was also strongly suspected or previously known. No changes to the manuscript text were made based on this comment.

The discussion of the other-than *WT1* germline mutations is a lot more scarce and fragmentary. Are these mutations predicted to be activating or inactivating, and coupled to this, are they heterozygous or homozygous? Are there (unique or recurring) additional somatic mutations in these tumors? Are there themes to be recognized with respect to some germline mutations and 11p15 status? Many of these things could be found in figure 3, but in my opinion these sort of figures (and many papers have them) provide too much data in one figure to be informative, an extra table summarizing what can be found specifically in germline mutation cases would be very useful.

Response: Thank you for this comment. **Supplementary Data 1** has been added summarizing the germline mutations in the study. The fraction of reads from the variant versus wild type alleles featured in **Supplementary Data 1** in the germline tissue versus the tumor tissue can allow the reader to interpret whether the germline mutations are heterozygous or homozygous. All described germline mutations in this manuscript are heterozygous, with many becoming homozygous in the accompanied tumor tissue as outlined by the term (VAF (variant allele frequency) increase in tumor) in **Supplementary Data 1**.

Patients with germline variants detected in peripheral blood in *WT1* and other cancer or Wilms tumor predisposition genes were found to more frequently exhibit 11p15.5 LOH or retention of normal imprinting status in their tumors compared to patients without detection of germline variants in peripheral blood. Comments in the results section have been modified in response to this comment: **Results (3rd paragraph under germline variant analysis from blood and associated tumor findings):** *“The presence of blood germline variants was strongly associated with the tumor chromosome 11p15.5 status (Chi-square $p < 0.0001$). Among 37 tumor samples (from 25 patients) with germline variants, 20 had 11p15.5 LOH (54%), 10 had 11p15.5 LOI (27%), and 7 had 11p15.5 ROI (18.9%). Among 53 tumor samples (from 39 patients with available whole genome or whole exome sequencing data) with 11p15.5 LOI, 43 did not have blood germline variants (81.1%) and 10 did have germline variants (18.9%; Fig. 2). As another way of looking at this association, among 48 tumor samples from patients without predisposing germline variants detected, 43 had 11p15.5 LOI and 5 did not. These data suggest two groups of predisposing events in BWT: 11p15.5 LOI or a germline genetic variant (often followed by 11p15.5 LOH).”*

The most consistent theme from these data is the WT1 germline mutation followed by 11p15.5 LOH followed by *CTNNB1* sequence that is outlined in **Figure 6**.

I think the information in line 354-360 suggests that all the cases without germline mutations have 11p15 LOI. However, in this paragraph there are 37 tumors with germline mutations and 52 with 11p15 LOI, adding up to 89. However the total number of tumors is 99. Do I misunderstand this paragraph or is there a mistake in the numbers given?

Response: We appreciate your careful review of the numbers in the manuscript.

There are 99 total tumor samples from 68 patients with synchronous bilateral Wilms tumor in this study. We have again re-verified these numbers (shown in **Figure 1 and 2**) based on this comment. Please note (as reflected in **Figures 1 and 2**) that there are 14 tumor samples from 7 patients with synchronous BWT who did not have available blood germline samples for genetic sequencing. These samples are represented by the paired boxes on the right side of the **Figure 2** graphic with grayed out sequencing data boxes. Therefore, we could only perform the methylation array (and RNA-seq) on these patient samples. Nevertheless, we were able to determine the 11p15.5 status in these samples from the methylation array data. We decided to include these samples to strengthen the finding that 11p15 status is shared between synchronous bilateral tumors (it adds 7 more instances of this phenomenon in a very rare cohort). We acknowledge this may add confusion to the manuscript, but during this review we have assured that all numbers are reported correctly and paired synchronous bilateral samples were exceedingly hard to come by for this area of study.

For further clarification:

There are 37 tumor samples (from 25 patients) with germline variants. Among those 37 tumor samples, 20 had 11p15.5 LOH, 10 had 11p15.5 LOI, and 7 had 11p15.5 ROI. We have re-verified these numbers based on this comment and changed the language in this paragraph for clarity: **Results (3rd paragraph under germline variant analysis from blood and associated tumor findings):** *“The presence of germline variants was strongly associated with the tumor chromosome 11p15.5 status (Chi-square $p < 0.0001$). Among 37 tumor samples (from 25 patients) with germline variants, 20 had 11p15.5 LOH (54%), 10 had 11p15.5 LOI (27%), and 7 had 11p15.5 ROI (18.9%). Among 52 tumor samples (from 39 patients with available whole genome or whole exome sequencing data) with 11p15.5 LOI, 42 did not have germline variants (80.7%) and 10 did have germline variants (19.2%; Figure 2). These data suggest two groups of predisposing events in BWT: 11p15.5 LOI or a germline genetic variant (often followed by 11p15.5 LOH).”*

The statement/interpretation that all tumor samples without germline mutations have 11p15.5 LOI is reflective of an overall theme but is not entirely accurate. As you can see in Figure 2, there are six tumor samples from patients without germline mutations that did not have 11p15.5 LOI in the tumors:

SJWLM069394 – R side: 11p15.5 ROI

SJWLM069387 – 11p15.5 ROI

SJWLM069388 – 11p15.5 LOH

SJWLM069384 – 11p15.5 ROI

SJWLM069398 – 11p15.5 ROI

SJWLM069401 11p15.5 ROI

In other words, this means that among 48 tumor samples from patients without germline predisposing variants, 42 tumor samples had 11p15.5 LOI and 6 did not. The following text has been added to this paragraph of the results section: **Results (3rd paragraph under germline variant analysis from blood and associated tumor findings):** *“As another way of looking at this association, among 48 tumor samples from patients without predisposing germline variants detected, 42 had 11p15.5 LOI and 6 did not.”*

An important part of the manuscript discusses the implications of the data for deducing the developmental stage that mutations / aberrations occur and the tumors start developing. This is especially something where the analysis of BWTs can be very informative. I fully agree with the conclusion that the small number of identical somatic mutations (like CTNNB1) in the two tumors of a patient means that this is a late event after the individual kidneys have started to form. However I struggle more with the interpretation that the many shared 11p15 aberrations mean these tumors arise post-zygotic but before the kidneys start to form (quickly after gastrulation). I agree this aberration is likely a very early event, but I'm not convinced that these are the cause of these tumors. The reason for my doubts on this is the mosaic 11p15 aberrations in normal kidney tissue presented here and in other publications. If the 11p15 event would be the causative event, and if there are no other mutations found (as I understand it), what makes the difference between a cell in the developing kidney that does become a Wilms tumor and a neighboring cell with the same 11p15 status that does not? It could be something epigenetic, but wouldn't at least a suggestion of this have been found in the methylation data?

I would happily admit that the early 11p15 event in these cases could provide the selective space in which additional events occur (whatever they are and for whatever reason they haven't been found here), which are then the cause of the tumors starting to develop. One could still call the 11p15 event the predisposing factor, but it would not be the causative event. This is not just a semantic argument, as it would make clear why this predisposing event in these cases cannot be used to determine the site and stage the tumors originate from, which would in my opinion be the most important message from this work. I would be happy to be convinced otherwise by the authors.

Response: Thank you for these insightful comments. We wholeheartedly agree that 11p15.5 “events” are the basis of bilateral Wilms tumor predisposition but do not themselves cause bilateral Wilms tumor. We agree that they provide the “selective space in which additional events occur.” We think the distinction between tumor predisposition and actual tumor development is critical to understand. As a common clinical example of this distinction, we believe all patients with Beckwith Wiedemann syndrome (especially the cohort with overt/complete 11p15.5 LOI in their germline) are predisposed to Wilms tumor, but not all will develop it. Therefore, in response to this comment, we have attempted to carefully go through the manuscript and indicate that 11p15.5 events are setting the stage for BWT predisposition rather than tumor development: **Discussion (1st paragraph):** *“This study determined the landscape of predisposition for bilateral Wilms tumor...Therefore, we found that BWT predisposition in the absence of a predisposing germline variant is a manifestation of the Beckwith Wiedemann spectrum often without overt additional clinical features.”*

We have attempted to state our hypothesis in terms of predisposition, rather than actual tumor development: **Discussion (4th paragraph):** *“We hypothesized that 11p15.5 LOI (H19/ICR1 gain of methylation) was a somatic mosaic epigenetic event that occurred in the early embryo in a mesodermal progenitor cell during the time when genomic imprints are being established and led to BWT predisposition.”*

My final two remarks are about the final two paragraphs in the discussion. The authors are very honest about the limitations of this study. Issues like genetic heterogeneity between cases in the causative mutations potentially make the dataset a collection of many different groups with only very few cases per group. Gadd et al showed in 2012 that different initiating mutations subdivide the tumors in multiple groups with different histology and different stages of origin. If this is the case (and I fully believe this), can we even call these groups cases of the same disease, or are they different diseases with common symptoms? If their genetic and developmental origin is different, I tend to go for the latter. Differences in treatment regime between samples make this problem even bigger, as pretreatment might very well have an effect on the additional, later mutations that are found in the tumor as they can directly affect the selective pressure for these additional mutations. I certainly don't underestimate these sort of problems, especially in a disease where sample numbers are (luckily from the patient perspective) small to begin with. But at some point we run into so many known unknowns and unknown unknowns that the risk of overinterpretation of data becomes too big.

Response: We can't definitively answer this question with the available data. However, given the profound genetic and epigenetic heterogeneity in Wilms tumor predisposition and development, we agree that what we regard as a single disease is likely more complicated. And, in fact, therefore we aim to study bilateral Wilms tumor radiographic and oncologic outcomes according to *WT1* germline status and tumor 11p15 status in the future – we hypothesize that the disease behaves quite differently clinically depending on the mode of predisposition that led to tumor development. We have reworded the discussion section to more clearly illustrate how the different modes of predisposition can lead to different clinical impact and how we must prospectively capture this to ascertain new biomarkers for BWT treatment.

Finally the authors state their work has enabled them to provide strong support for the Knudson-Strong two-hit hypothesis (line 686-687). I fully agree that parts of the data, especially on the germline mutation cases, are completely consistent with this. I do not agree that these data make the already existing confirmation of the model in the 51 year since it was proposed substantially stronger than it already was without these data. If anything, if my issues are correct (and I might be completely wrong) these data show the limitations of the model in describing the origins of tumours at the level of detail we currently can. This would not invalidate the model, just nuance it, but it would in my opinion certainly not provide particular strong extra data to support it.

Response: In their original model, Knudson and Strong also state that the “first” hit could be a post-zygotic somatic event. Remarkably, their inference was based only on epidemiologic and histologic data for bilateral Wilms tumor and perhaps similarities to retinoblastoma. Nothing about the genetics of Wilms tumor was known at the time. We believe the core finding of the current study is that 11p15.5 LOI post-zygotic somatic mosaicism is likely the most common mode of predisposition for eventual

bilateral Wilms tumor formation. Therefore, we have provided critical additional evidence supporting the Knudson-Strong two-hit hypothesis. Nevertheless, we have eliminated reference to the Knudson-Strong hypothesis in the discussion section of the manuscript in response to this reviewer comment. We did not find that it added critical information to the discussion, but it was a helpful tool to introduce several concepts in the introduction section.

Reviewers' Comments:

Reviewer #1:

Remarks to the Author:

The authors have revised their manuscript extensively, including provision of new data and correction of figures and numbers of samples. Their comprehensive revision has addressed all my comments to my satisfaction. They also appear to have answered both of the other reviewers' comments in a similar expansive fashion.

Reviewer #2:

Remarks to the Author:

The authors have revised the manuscript significantly. However, there are still several issues with the manuscript that need to be addressed. Specifically, there needs to be clarity of language and word usage for "somatic mosaicism", "post-zygotic", and "germline". These terms can have different meanings and usage by oncologists and geneticists, and they need to be defined and used appropriately. Additionally, the thresholds for LOI and LOH measurements in the different tissues need to be uniform, if the thresholds are defined one way and as previously described for tumor and kidney but then used differently looking at trends in blood, with all of the data being used to create a unified narrative, there is an issue with the narrative and the way the data are being analyzed to form the narrative. The blood findings and the changes overtime need to be reframed or removed from the results because the current presentation confounds the results. Additionally, in light of the definitions requested above, the authors need to consider and comment on the molecular findings in the blood in line with the current clinical and molecular definitions of BWS if these results are left in the manuscript. They also need to comment of why the blood methylation over time is included and why BWT and non-BWT patients who are not matched for age make sense for comparison. All of these issues were previously raised by the reviewer and not sufficiently addressed in the author response. Please address each of these comments as well as the remaining comments listed below as not completely addressed.

1. The numbers of samples studied need to be clarified as do the paired-normal samples. The abstract does not provide complete representation of the samples tested and the conclusions drawn based on the samples tested. "Germline" vs "somatic" changes need to be clarified, with germline being in blood and somatic being in tissue.

Response: The abstract (previously over 300 words) has been modified according to this reviewer comment and the journal requirements, which limit the abstract to an unstructured 150-word format with no abbreviations or acronyms:

"Developing synchronous bilateral Wilms tumor suggests an underlying (epi)genetic predisposition. We evaluated this predisposition in 68 patients using whole exome or genome sequencing (n=85 tumors from 61 patients with matched germline blood DNA), RNA-seq (n=99 tumors), and DNA methylation analysis (n=61 peripheral blood, n=29 non-diseased kidney, n=99 tumors). We determined the predominant events for bilateral Wilms tumor predisposition: 1)pre-zygotic germline genetic variants readily detectable in blood DNA [WT1 (14.8%), NYNRIN (6.6%), TRIM28 (5%), and BRCA-related genes (5%)] or 2)post-zygotic epigenetic hypermethylation at 11p15.5 H19/ICR1 that is less readily detectable in peripheral blood in an individual and may require analysis of multiple tissue types for diagnosis. Of 99 total tumor specimens, 16 (16.1%) had 11p15.5 normal retention of imprinting, 25 (25.2%) had 11p15.5 copy neutral loss of heterozygosity, and 58 (58.6%) had 11p15.5 H19/ICR1 epigenetic hypermethylation (loss of imprinting). Here, we ascertained the (epi)genetic landscape of bilateral Wilms tumor predisposition."

Reviewer Response: The authors have addressed comment 1.

2. Are the germline WT1 mutations in patients with identified genetic syndromes? If yes, what syndromes, if no, how was it ascertained that these patients are not syndromic?

Response: The methods section has been updated to address this question: Methods (under Clinical Data subheading): "For the St. Jude cohort, syndromes were ascertained by thorough review of the medical record and all molecular testing documentation. For the COG cohort, the presence of genetic syndromes was a collected/reported data point from the originating studies and was delivered to the investigators after molecular analysis was complete."

Our data are consistent with recent unselected cohorts that demonstrated patients with pathogenic germline variants often have no overt clinical phenotype other than the development of Wilms tumor (Hol, J.A. et al. Prevalence of (Epi)genetic Predisposing Factors in a 5-Year Unselected National Wilms Tumor Cohort: A Comprehensive Clinical and Genomic Characterization. *J Clin Oncol*, JCO2102510 [2022]).

The results section has been updated to address this question: Results (under Germline variant analysis from blood and associated tumor findings heading): "Of 9 patients with pathogenic WT1 germline variants in this study, 3 had features of genetic syndromes and 6 did not. SJWLM066776 had Denys Drash syndrome, SJWLM066772 had a disorder of sexual development (DSD), and SJWLM066774 had congenital nephrotic syndrome and idiopathic dilated cardiomyopathy."

Reviewer Response: The authors have addressed comment 2.

3. WT1 plus LOH is indicated in one place as mutually inclusive, is that actually the case?

Response: Out of 14 tumor samples from the 9 patients with germline WT1 variants in the current study, all 14 (100%) were found to have 11p15.5 LOH. However, these findings are not mutually inclusive because tumor samples without WT1 germline variants were also found to have 11p15.5 LOH in this study as demonstrated in Figure 2.

The explanation for this strong association between pathogenic germline WT1 variants and 11p15.5 LOH is indicated in Figure 6 in which a heterozygous inactivating germline WT1 pathogenic variant present on the paternal allele becomes homozygous by copy neutral loss of heterozygosity at 11p with breakpoints that include both the WT1 locus at 11p13 and the H19/IGF2 locus at 11p15.5. In fact, the breakpoints of 11p15.5 LOH encompass both loci in 18/19 cases (94.7%) with 11p15.5 LOH in this study.

However, in our prior study (Murphy, A.J. et al. Forty-five patient-derived xenografts capture the clinical and biological heterogeneity of Wilms tumor. *Nat Commun* 10, 5806 [2019].), samples from 5/7 patients with WT1 mutations had 11p15.5 LOH. 4 of these 7 patients had bilateral Wilms tumor and all these 4 had 11p15.5 LOH. The two patients that did not have 11p15.5 LOH were unilateral Wilms tumor samples. Also, there are multiple examples of unilateral Wilms tumors in the NCI-TARGET data set that have WT1 mutations and do not have 11p15.5 LOH. We suspect that the 11p15.5 LOH/WT1 mutual inclusivity is true in patients with germline WT1 variants and bilateral Wilms tumor, but when WT1 mutations are somatic in patients with unilateral Wilms tumor this association may not hold true.

The discussion section has been updated in response to this comment: Discussion (2nd paragraph): "Although all tumor samples from patients with pathogenic WT1 germline variants in the current study exhibited 11p15.5 LOH, prior work from our group and others in unilateral Wilms tumor samples demonstrated somatic WT1 mutations without 11p15.5 LOH."

Reviewer Response: The authors state, "Although all tumor samples from patients with pathogenic WT1 germline variants in the current study exhibited 11p15.5 LOH, prior work from our group and others in unilateral Wilms tumor samples demonstrated somatic WT1 mutations without 11p15.5 LOH." and have referred to -

a) Murphy, A.J. et al. Forty-five patient-derived xenografts capture the clinical and biological heterogeneity of Wilms tumor. *Nat Commun* 10, 5806 (2019).

b) Gadd, S. et al. A Children's Oncology Group and TARGET initiative exploring the genetic landscape of Wilms tumor. *Nat Genet* 49, 1487-1494 (2017).

The data in these papers clearly indicate that WT1 plus LOH are not mutually exclusive. Please amend comments.

4. How is the LOH distinguished from GOM or LOI? Do these patients have underlying clinical features suggestive of a syndrome? How was this determined?

Response: 11p15.5 LOH was distinguished from 11p15.5 LOI (aka 11p15.5 H19/ICR1 GOM) using data from the MethylationEPIC beadchip array. The average β value for H19/ICR1 was calculated using CpG probes located within the chr11:2,019,974-2,024,738 (GRCh38/hg38) range and the average β value for 11p15.5 KCNQ1OT1/ICR2 was calculated using probes located within the chr11:2,721,228-2,722,228 (GRCh38/hg38) range. Samples with average β value H19/ICR1 < 0.7 and KCNQ1OT1/ICR2 > 0.3 were determined to have normal retention of imprinting (ROI), samples with H19/ICR1 > 0.7 (hypermethylation) and KCNQ1OT1/ICR2 > 0.3 were determined to have loss of imprinting (LOI) at H19/ICR1, and samples with H19/ICR1 > 0.7 (hypermethylation) and KCNQ1OT1/ICR2 < 0.3 (hypomethylation) were determined to have loss of heterozygosity (LOH) at 11p15.5. LOH at 11p15.5 was also designated if samples had LOH or partial LOH detectable using the CONSERTING algorithm from whole genome sequencing data. This is detailed in the methods section of the manuscript (2nd paragraph under Methylation analysis subheading).

Thank you for your attention to details of the clinical syndromes. The methods section has been updated to address this question: Methods (under Clinical Data subheading): "For the St. Jude cohort, syndromes were ascertained by thorough review of the medical record and all molecular testing documentation. For the COG cohort, the presence of genetic syndromes was a collected/reported data point from the originating studies and was delivered to the investigators after molecular analysis was complete."

The results section has been updated to address this question: Results (2nd paragraph under 11p15.5 status is shared in synchronous BWT subheading): "Most patients with 11p15.5 LOH or 11p15.5 LOI detected in their tumor samples had no clinical features suggestive of a syndrome (Fig. 2). However, of the 9 patients with definitive 11p15.5 alterations in multiple tissues, SJWLM066781 had 11p15.5 LOI in their tumor and adjacent non-diseased kidney and had a ureteral duplication. SJWLM069381 had 11p15.5 LOI in their tumor and adjacent non-diseased kidney tissue and had hemihypertrophy. SJWLM069390 had 11p15 LOH detected in their tumor and adjacent non-diseased kidney tissue. In addition, this patient was found to have mosaic 11p15.5 LOH in their blood as determined by whole genome sequencing (Supplementary Fig. 2). This patient was noted to have a BWSp clinical phenotype."

Reviewer Response: The authors need to state that the clinical evaluation and reported data may not be complete. There are several publications indicating that clinical diagnosis of BWS may be missed until after a WT is detected and as such patients may not be labeled "syndromic" in these cancer databases. Additionally, while the ranges of methylation utilized in this work and in the work cited are quite broad and may miss lower level mosaicism for methylation changes and LOH consistent with BWSp. This is noted in the blood levels for these patients but discounted in the results and discussion. Together, this mosaicism question needs to be addressed further.

5. The somatic mosaicism for LOI falls within the normal range per the parameters defined in the study, so how is drift or somatic mosaicism defined for these BWT patients and long-term survivors? Please clarify the technique and data analysis approach used and revise this section in the abstract and the manuscript.

Response: The text of the results section has been adjusted to address this comment: Results (4th-6th paragraphs under 11p15.5 status is shared in synchronous BWT subheading): "11p15.5 LOI being found in paired synchronous BWT that also contained shared tumor noncoding somatic variants and in adjacent non-diseased kidney tissue (but not above established thresholds in DNA obtained from leukocytes) is suggestive of post-zygotic somatic mosaicism. We reasoned that, if present, evidence of post-zygotic somatic mosaicism for chromosome 11p15.5 LOI should be detectable in peripheral blood on the cohort level since hematopoietic progenitor cells have a mesodermal embryonic origin.³⁴ To explore whether evidence of somatic mosaicism could be detected on the cohort level in peripheral blood samples from patients with tumors bearing 11p15.5 LOI, we compared the H19/ICR1 β values from leukocyte-derived DNA from peripheral blood among patients according to the 11p15.5 status of

their tumors (ROI, LOH, LOI). We found a statistically significant increase in H19/ICR1 methylation detectable in peripheral blood in patients who had tumors with 11p15.5 LOI compared to those with retention of imprinting (Fig. 3A). Of note, this statistically significant gain of methylation was low-level with none of the peripheral blood samples achieving the β value of 0.7 associated with frank germline 11p15.5 LOI. In contrast, 7 of the adjacent non-diseased kidney samples met the threshold for frank germline 11p15.5 LOI (Fig. 3B)."

"To validate the finding of somatic mosaicism for 11p15.5 LOI detectable in peripheral blood on the cohort level, we combined identically processed and normalized H19/ICR1 methylation β values from leukocyte-derived DNA between our current BWT cohort (n=61) and a cohort of healthy community controls (n=282) and WT cancer long-term survivors (including survivors of both unilateral [n=154] and BWT [n=17]) from the St. Jude Life Cohort Study.³⁵ We hypothesized that H19/ICR1 values in peripheral blood DNA would be higher in BWT patients than healthy community controls and potentially unilateral WT patients. The peripheral blood 11p15.5 H19/ICR1 and KCNQ1OT1/ICR2 β values were determined to be normally distributed in all groups using the Kolmogorov-Smirnov test. The mean H19/ICR1 peripheral blood methylation β value in the healthy community control cohort was 0.499 ± 0.0248 (mean \pm standard deviation). The mean H19/ICR1 peripheral blood methylation β value from the BWT cohort (0.534 ± 0.0298) was higher than both the unilateral WT (0.521 ± 0.0258 ; $p < 0.0001$) and healthy community control cohorts ($p < 0.0001$; Fig 3D.) Using the method of Fiala, et al.²³, we determined that 26 BWT patients (n=20 from current study; n=6 from St. Jude Life cohort) had a H19/ICR1 methylation β value greater than two standard deviations from the mean community control cohort β value (i.e., H19/ICR1 $\beta > 0.54864$; Fig. 2; Supplementary Table 1). Of these 20 patients, 17/20 (85%) were female, 18/20 (90%) had 11p15.5 LOI in their tumor, 2/20 (10%) had 11p15.5 LOH in their tumor, and 0 had 11p15.5 ROI in their tumor (Supplementary Table 1)."

"However, because leukocyte DNA methylation is known to change with age³⁶ and the healthy control population with peripheral blood DNA available for this study was derived from cancer survivors of a higher median age than our current study population, we wanted to assess H19/ICR1 methylation as a function of age. We noted an inverse relationship between methylation at H19/ICR1 and increasing age in healthy community controls (Fig. 3E). In contrast, we detected positive correlation between increasing age and H19/ICR1 methylation in BWT patients and long-term survivors. This positive correlation was not seen in long-term survivors of unilateral WT (Fig. 3E)."

The limitations paragraph of the discussion section has also been updated to address this comment: Discussion (2nd to last paragraph): "Finally, all data in this manuscript reporting evidence of mosaic 11p15.5 H19/ICR1 gain of methylation detectable in peripheral blood are based on statistical comparison or observed changes in comparison to a normal cohort, while no patients in this manuscript exhibited frank hypermethylation above the established threshold of a β value of 0.7 at 11p15.5 H19/ICR1 detectable in DNA obtained from peripheral blood."

Reviewer response: The authors had stated, "Samples with average β value H19/ICR1 < 0.7 and KCNQ1OT1/ICR2 > 0.3 were determined to have normal retention of imprinting (ROI)." Based on this any change in methylation within this range is normal. By stating, "We found a statistically significant increase in H19/ICR1 methylation detectable in peripheral blood in patients who had tumors with 11p15.5 LOI compared to those with retention of imprinting", the authors are contradicting their own findings. Furthermore, several groups have demonstrated low-level mosaic LOI or LOH/UPD as molecularly diagnostic of BWSp, this text needs to be revised to reflect that these patients have molecular findings suggestive of BWSp that may fall below the threshold for detection for this molecular testing modality. The drift of the changes over time is not relevant and it is unclear why the authors are discussing it in such detail.

6. Reframe the mesoderm hypothesis to consider blood and the timing of imprinting being established during embryonal development. Clarify how the normal kidney fits into this and indicate which normal kidney samples are analyzed in parallel to tumor samples to support this hypothesis.

Response: Thank you for this suggestion to clarify, or more explicitly state, the underlying hypothesis. We have rewritten the discussion section of the manuscript with this specific comment in mind. Although the whole discussion section was reworked with this comment in mind, the most relevant

section is noted here: Discussion (4th paragraph): “We hypothesized that 11p15.5 LOI (H19/ICR1 gain of methylation) was a somatic mosaic epigenetic event that occurred in the early embryo in a mesodermal progenitor cell during the time when genomic imprints are being established and led to BWT predisposition. Genomic DNA methylation, which is the primary responsible mechanism for imprinting, reaches its final level near the time of embryonic gastrulation (i.e., formation of the three germ layers – ectoderm, endoderm, mesoderm). 48 This developmental timepoint is precisely when the mesodermal cells that give rise to the right and left intermediate mesoderm/kidney primordia are spatially isolated as they invaginate to establish the mesoderm and migrate to the right or left lateral sides of the embryo. 49 Multiple lines of evidence in this study support this mesodermal somatic mosaic hypothesis. First, 11p15.5 LOI is shared in many synchronous BWT, while other downstream common somatic genetic alterations (exact variants in CTNNB1, microRNA processing genes, SIX1/2, TP53, etc.) are not. Next, synchronous BWT with shared 11p15.5 LOI also exhibit a small number of shared noncoding variants relative to the total number of noncoding variants demonstrated in each paired tumor, consistent with shared clonal origin in the early embryo with subsequent divergence and independent evolution. Finally, the frequent detection of 11p15.5 H19/ICR1 hypermethylation at frank germline thresholds ($\beta > 0.7$) in adjacent non-diseased kidney tissue and associated tumors but not blood is consistent with mosaicism throughout the body.”

Reviewer response: Please define frank germline, methylation changes that do not have an underlying genetic cause (ie a deletion or duplication causing the LOI, are not germline. The authors state, “Finally, the frequent detection of 11p15.5 H19/ICR1 hypermethylation at frank germline thresholds ($\beta > 0.7$) in adjacent non-diseased kidney tissue and associated tumors but not blood is consistent with mosaicism throughout the body”. Presence of mosaicism indicates presence of a syndrome. The authors need to mention the limitation of their genetic experts to identify the syndromic samples and what is reported in cancer data collections like COG. Additionally, the discussion of LOI in blood, kidney, and tumor need to be revised to reflect that these are post-zygotic mosaic changes and that there are thresholds for detection being defined here and just because something falls below the threshold defined, it does not mean that the post-zygotic change is not occurring.

7. How do the authors define the patients with 11p15 alterations in blood and/or normal kidney in addition to tumor?

Response: We have updated the results section of the manuscript to feature patients more clearly with 11p15 alterations in multiple tissues: Results (2nd paragraph under 11p15.5 status is shared in synchronous BWT subheading): “However, of the 9 patients with definitive 11p15.5 alterations in multiple tissues, SJWLM066781 had 11p15.5 LOI in their tumor and adjacent non-diseased kidney and had a ureteral duplication. SJWLM069381 had 11p15.5 LOI in their tumor and adjacent non-diseased kidney tissue and had hemihypertrophy. SJWLM069390 had 11p15 LOH detected in their tumor and adjacent non-diseased kidney tissue. In addition, this patient was found to have mosaic 11p15.5 LOH in their blood as determined by whole genome sequencing (Supplementary Fig. 2). This patient was noted to have a BWSp clinical phenotype.”

Except for patient SJWLM069390 (featured in Supplementary Figure 2), who had definitive determination of 11p15.5 LOH mosaicism detected in peripheral blood, we could not definitively assign mosaic status to a blood sample from any other patient in this study. However, detection of 11p15.5 alterations in kidney and tumor, but not blood is highly suggestive of somatic mosaicism. This has been highlighted more in the Discussion section and throughout the manuscript. In addition (Discussion, 6th paragraph): , “Because the data discussed above support 11p15.5 LOI somatic mosaicism as a major contributor to the predisposition for BWT development, we hypothesized and confirmed that patients and survivors of BWT had higher levels of H19/ICR1 methylation detectable in peripheral blood than a cohort of healthy community control subjects and unilateral WT. Using the method of Fiala et. al.²³, we determined that 20 patients from the current study had “low-level” gain of methylation detectable in peripheral blood defined as two standard deviations above the mean H19/ICR1 beta value in the community control cohort. Seventeen of these 20 patients were female and 18/20 had 11p15.5 LOI detected in their tumor sample. Fiala et. al. previously showed that 7/7 BWT patients with “low-level” H19/ICR1 hypermethylation detectable in peripheral blood were also

female.23”

Reviewer response: The authors need to distinguish between post-zygotic changes and somatic mosaicism. They also need to discuss the threshold they are using to define the mosaicism and not use the thresholds which are arbitrary (even though they are cited as based on previous data) to define the mechanism occurring. There is also no specific known correlation between females with low level LOI. There is not enough data in either the Fiala or this cohort to make that determination. The authors are not clearly discussing the limits of their testing and are using the limits of their testing to define mechanism. Additionally, the change in LOI over time in is further confusing the discussion. Comparing blood at a static time point in patients with BWT and without is fine but they authors are also not using age-matched datasets.

8. Is paternal uniparental isodisomy and LOH defined as the same thing/process? Please comment on the timing of these events and how that correlates with the data presented.

Response: Thank you for this clarifying question. We have attempted to include language with more clarity in the introduction section of the manuscript to address this question. 11p15.5 LOH and 11p15.5 uniparental isodisomy are defined as synonymous terms in this study: Introduction (2nd paragraph): “Among patients with BWS, those with epigenetic gain of methylation at H19/ICR1 (loss of imprinting - LOI) or paternal uniparental disomy (loss of genetic material from the maternal 11p15.5 locus with duplication of the paternal allele in this region; a state known as copy neutral loss of heterozygosity - LOH), both of which result in biallelic expression of IGF2, have the highest risk for any WT development.”

The discussion section has been modified in response to this comment in an effort to focus more explicitly on the timing of 11p15.5 LOH events: Discussion (3rd paragraph): “Two synchronous BWT sample pairs in this study (SJWLM066776 and SJWLM069396) from patients with inactivating pathogenic blood germline WT1 variants demonstrated a small number of shared noncoding tumor somatic variants in addition to tumor 11p15.5 LOH (Table 2). This constellation of findings implies 11p15.5 LOH can occur in the early embryo and may be required ahead of malignant transformation on the background of WT1 mutation. The sequence of pathogenic germline WT1 mutation and subsequent somatic 11p15.5 LOH is often followed by development of activating tumor CTNNB1 variants, which were the most common somatic variants found in the current study. Our results are consistent with the temporal sequence first reported by Fukuzawa et. al. who demonstrated that WT1 variants were detected in nephrogenic rests and WT, but CTNNB1 variants were only found in the adjacent WT.42 WT1 and CTNNB1 variants often co-occur in WT and the spatial distribution of CTNNB1 variants has been demonstrated to exhibit intra-tumor genetic heterogeneity.21,43 WT driven by WT1 variants are known to exhibit stromal/rhabdomyoblastic differentiation and poor volumetric regression in response to neoadjuvant chemotherapy.44,45 Therefore, future knowledge of germline WT1 status at diagnosis in patients with BWT could guide expectations regarding volumetric tumor regression and timing of surgical resection. Taken together, these data and a recent study by Hol. et. al that demonstrated a much higher than predicted incidence of germline WT1 variants in unselected WT patients (either unilateral or bilateral, often without syndromic features), support expanded germline genetic testing at diagnosis for all patients with BWT...19”

Reviewer Response: The authors have addressed comment 8.

9. Figure 1 should be in the supplement.

Response: Thank you for this suggestion. Figure 1 has been relocated to the Supplement (now Supplementary Figure 1) and the remaining figures and supplementary figures have been re-numbered accordingly.

Reviewer Response: The authors have addressed comment 9.

10. Sample acquisition section: how do the levels of LOH or LOI compare to the % tumor in a specimen? How was this accounted for/determined?

Response: Because most bilateral Wilms tumor specimens are pretreated with chemotherapy (as indicated in Figure 2) and therefore can exhibit significant necrosis or fibrosis/stromal change, we did not have the luxury of only utilizing samples with extremely high tumor purity in this study. Furthermore, because of the genetic heterogeneity described among paired synchronous bilateral Wilms tumor in this study, even different tumors within the same patient can differentially respond to treatment and exhibit different purity when samples are collected. We set a percent tumor threshold of >50% for the study, which was determined by pediatric pathologists at the time of sample inclusion and nucleic acid isolation. After nucleic acids were received, tumor purity estimates were then computed with a deconvolution-based approach using DNA methylation data as previously described (Chakravarthy, A. et al. Pan-cancer deconvolution of tumour composition using DNA methylation. *Nat Commun* 9, 3220 [2018]). For tumor samples with corresponding whole genome sequencing data, we found a high positive correlation between tumor purity estimates derived from DNA methylation and whole genome sequencing data ($r=0.78$, $p=3.2e-13$).

In response to this comment, we have now reported the tumor purity estimates for each sample along with the corresponding 11p15.5 H19/ICR1 and ICR2 methylation Beta values in Supplementary Data 5. As we predicted, tumor purity positively correlated with ICR1 methylation Beta values [Pearson's correlation, 0.43 ($P = 8.3e-08$); Spearman's correlation, 0.66 ($P < 2.2e-16$)]; and negatively correlated with ICR2 methylation Beta values Pearson's correlation, -0.46 ($P = 1.2e-08$); Spearman's correlation, -0.52 ($P = 4.6e-11$).

We have updated the methods section in response to this comment: Methods (1st paragraph under Methylation analysis subheading): "Tumor purity was estimated from the methylation array data using a deconvolution-based approach as previously described.⁵⁸ These purity estimates were validated by comparison to estimates derived from whole genome sequencing data in the COG specimens."

We have updated the results section in response to this comment: Results (1st paragraph under 11p15.5 status is shared in synchronous BWT subheading): "Tumor purity calculated using a deconvolution-based approach from methylation array data correlated strongly with tumor purity calculated using whole genome sequencing data in the COG specimens (Pearson $r=0.78$, $p=3.2e-13$). Tumor purity estimates and corresponding 11p15.5 H19/ICR1 and KCNQ1OT1/ICR2 methylation β values are shown in Supplementary Data 5. Tumor purity positively correlated with ICR1 methylation β values (Pearson $r=0.43$, $p=8.3e-08$) and negatively correlated with KCNQ1OT1/ICR2 methylation β values (Pearson $r=-0.46$, $p=1.2e-08$)."

We have updated the limitations paragraph of the discussion section in response to this comment: Discussion (2nd to last paragraph): "Still, the number of paired synchronous BWT specimens was limited by inconsistent sample collection, insufficient tumor purity, availability, and heterogeneous treatment response which could have caused some tumors to be ineligible for inclusion due to necrosis, stroma, or tumor content thresholds. Our results show that tumor purity can affect measurement of methylation at 11p15.5 H19/ICR1 and ICR2; however, variations in tumor purity in pre-treated BWT will always be present due to the heterogeneity discussed above."

Reviewer Response: It is still unclear how the authors assess LOI or LOH if tumor purity is being based off of the same data as LOI and LOH.

11. Figure 2: Add the paired normal samples to the top part of the figure. Where are the normal samples in the discovery cohort? What were the tumor analyses compared to in this case? Are the signatures from discovery cohort comparable to expansion cohort? Figure 2 needs more detailing with respect to sample processing and comparison.

Response: In the cohort from St. Jude Children's Research Hospital, paired adjacent non-diseased kidney tissue was not available for analysis. Because of the high rate of bilateral nephron-sparing surgery performed for bilateral Wilms tumor patients using an enucleation technique at this center, no normal kidney is typically resected at the time of surgery. In contrast, in the COG study AREN0534, only 39% of the cohort underwent bilateral nephron-sparing surgery, and the remainder underwent

radical nephroureterectomy on at least one side. Therefore, adjacent non-diseased kidney tissue was available only from the COG cohort, usually just from one kidney. "Paired normal" samples are shown in Figure 1 (n=29) and are termed "adjacent non-diseased kidney." For determination of tumor somatic variants, a comparison between sequencing data from tumors and blood germline was made in both the St. Jude and COG cohorts as outlined in the Methods section: "For variant discovery, a paired analysis was performed comparing tumor-derived DNA to germline DNA obtained from peripheral blood leukocytes."

Reviewer Response: The authors have addressed comment 11.

12. Why is the discovery cohort named as such? If there is no validation cohort because the cohorts are designed differently and are showing different things, please rename and state this in the manuscript.

Response: Thank you for pointing this out. We agree that no formal a priori designation of discovery and validation cohorts was made. Therefore, the terms discovery and validation cohort have been eliminated from the manuscript. The cohorts are now simply referred to as St. Jude and COG cohorts. The text and Figure 1 have been updated to reflect these changes.

Reviewer Response: The authors have addressed comment 12.

13. Clinical data: how was it determined if the patients had congenital anomalies or syndromes? Please define this or state it is not known.

Response: The methods section has been revised according to this comment: Methods (under Clinical Data subheading): "For the St. Jude cohort, syndromes were ascertained by thorough review of the medical record and all molecular testing documentation. For the COG cohort, the presence of genetic syndromes was a collected/reported data point from the originating studies and was delivered to the investigators after molecular analysis was complete."

Reviewer Response: The authors have partially addressed comment 13. The authors need to mention "state is not known" for patients that show mosaicism and indicate the limitation of the data in COG and the St. Jude Cohort.

14. Total-strand RNA-seq – were normal sample run here? Where are the data?

Authors response : "Thank you for calling our attention to this oversight. RNA-seq data have now been utilized/included in the manuscript in the following manner: unsupervised hierarchical clustering plot of all tumors (Supplementary Figure 9), TSNE plot of paired bilateral samples (Supplementary Figure 10), and the Cis-X allele-specific expression analysis that identified recurrent reduction to allele specific expression in RNF185 located at chromosome 22q (Supplementary Table 2-3).

Overall, these results demonstrated that paired synchronous bilateral Wilms tumor samples clustered less robustly with one another by gene expression than methylation: Results section (1st paragraph under subheading 11p15.5 status and genome-wide methylation/molecular identity in BWT): "In contrast, paired synchronous BWT samples clustered less robustly by unsupervised hierarchical clustering and TNSE clustering of total-strand RNA-seq data (Supplementary Fig. 9-10)."

Integrated analysis of RNA-seq and whole genome sequencing data using the Cis-X method identified a perturbation of allele specific expression due to chromosome 22 loss in samples with 11p15.5 LOI or LOH, which prompted a more in-depth investigation into chromosome 22 (detailed below): Results section (2nd and third paragraphs under subheading 11p15.5 status and genome-wide methylation/molecular identity in BWT): "To further explore the suggestion that tumor 11p15.5 status correlates with a broader tumor molecular identity rather than only changes at the 11p15.5 locus itself, we determined genome-wide alterations in allele specific expression stratified by tumor 11p5.5 status (LOI, LOH, ROI) in the COG cohort using the Cis-X method, which integrates whole genome sequencing and total-strand RNA-seq data to determine the allele of origin for the RNA being expressed. Due to sample size limitations, we filtered results according to an unadjusted p-value of < 0.05 and a false-discovery rate p-value corrected for multiple testing of <0.25 for this analysis. As predicted, KCNQ1OT1 (FDR p=0.0046) and KCNQ1 (FDR p=0.0046) were found to have perturbation

of allele-specific expression in tumor samples with 11p15.5 LOH, but not ROI or LOI, while INS-IGF2 and IGF2 showed perturbation of allele-specific expression in both samples with 11p15.5 LOH and LOI. Five genes and/or noncoding RNAs outside the 11p15.5 region were found to have disruption in the normal patterning of allele-specific expression that correlated with tumor 11p15.5 status: RNF185, SNORD116-4, C1QL3, CLEC12A, and NPAS2. RNF185, a gene located at chromosome 22q12.2 that codes for a component of the E3 ubiquitin ligase pathway, was found to have reduction to monoallelic expression in 43.3% of tumor samples with 11p15.5 LOI (FDR $p=0.240$) and 35.7% of tumor samples with 11p15.5 LOH (FDR $p=0.041$), but 0% of samples with 11p15.5 ROI (Supplementary Table 2). We then queried for possible mechanisms of RNF185 reduction to monoallelic expression and noted recurrent chromosome 22q loss correlated with this phenomenon. Combined with the copy number data detailed above that showed recurrent chromosome 22q loss in our data set and shared chromosome 22q loss in 3 synchronous BWT pairs, we performed a focused analysis to determine the clinical and other molecular features of BWT samples with chromosome 22q loss. We found an enrichment for female biological sex, 11p15.5 LOI, and DGCR8 RNA microprocessor mutations in this cohort of 11 tumor samples with chromosome 22q loss from 8 patients (Supplementary Table 3)."

Reviewer response: The Supplementary figures 9-10, do not show RNA-seq data for normal kidney samples. This concern has not been addressed.

15. Methylation analysis – define what was used as "germline." Blood? Kidney? Define the criteria for normal/abnormal in the results and discussion of somatic mosaicism as it differs from what is defined in this section. Were normal kidney/tumor pairs tested? Be clearer if it is tumor/tumor pairs vs tumor/normal pairs. How were methylation results from the whole genome sequencing confirmed? In the results, methylation analysis and WGS results should both be provided. Please add these data. What was done with methylation data beyond the 11p15 region?

Response: 850K MethylationEPIC beadchip analysis was performed on all germline samples including DNA derived from peripheral blood and DNA derived from adjacent non-diseased kidney. In response to this comment, samples are now more clearly identified in the Figure 5 and 6 legends: "G1 – blood, G2 – kidney." In addition, we have now attempted to include the word "blood" when referring to germline results throughout the manuscript.

Because 11p15.5 loss of heterozygosity is a genetic event (copy neutral loss of heterozygosity aka paternal uniparental disomy) in which the maternal allele is deleted and the paternal allele is duplicated, this phenomenon could be validated by whole genome sequencing data using the CONSERING algorithm. For 11p15.5 loss of imprinting (LOI; H19/ICR1 gain of methylation), since this is a purely epigenetic event, the phenomenon could not be validated using whole genome sequencing data. Beyond the 11p15.5 region, methylation data were used for unsupervised hierarchical clustering and TSNE clustering of samples to demonstrate a relationship between 11p15.5 methylation status and a more global pattern of methylation in samples (Figures 4-5).

Reviewer Response: The authors have addressed comment 15.

16. Long terms survivorship analysis: please define how this analysis was done more clearly as the discussion focuses on variations that are within the normal range per the parameters set.

Response: The methods section of the manuscript has been updated in response to this comment: Methods (final paragraph under Long-term survivorship cohort analysis from peripheral blood DNA subheading): "Blood-derived Infinium MethylationEPIC Beadchip (850K) array germline DNA methylation data from the 61 patients in the current study were combined with blood-derived germline DNA methylation data from 282 healthy community controls and 171 long-term Wilms tumor survivors (>5 years from cancer diagnosis; $n=154$ unilateral, $n=17$ bilateral) from the St. Jude Lifetime Cohort Study (SJLIFE).³⁵ These data were normalized and processed with the subset-quantile within array normalization (SWAN) method using the R package minifi. The average β values from the 11p15.5 H19/ICR1 and KCNQ1OT1/ICR2 regions defined above were computed as detailed above. Because of known changes in leukocyte global DNA methylation that occur with age, methylation at 11p15.5 H19/ICR1 was plotted according to age.⁶⁴ Using these data, the relationship between age

and methylation at 11p15.5 H19/ICR1 was determined using a linear regression model. The relationship between age and methylation from unilateral WT and BWT samples were compared against the learned model from the healthy community control population.”

The limitations paragraph of the discussion section has been updated in response to this comment: Discussion (2nd to last paragraph): “Finally, all data in this manuscript reporting evidence of mosaic 11p15.5 H19/ICR1 gain of methylation detectable in peripheral blood are based on statistical comparison or observed changes in comparison to a normal cohort, while no patients in this manuscript exhibited frank hypermethylation above the established threshold of a β value of 0.7 at 11p15.5 H19/ICR1 detectable in DNA obtained from peripheral blood.”

Reviewer Response: The survivorship data does not add anything to the manuscript. Given that the results fall below the threshold of what is the point of including these data? If they are going to be included, then the authors need to refine the thresholds for the LOI findings throughout the manuscript. It is unclear what the point of showing trends that fall below the level of significance for the rest of the study. What does “frank methylation” mean?

17. How do the different or similar CTNNB1 mutations fit into the timing of the 11p15 alterations discussion?

Response: Different single nucleotide variants in paired synchronous bilateral Wilms tumors from the same patient that result in activating exon 3 Beta catenin mutations is suggestive of convergent evolution toward Wnt pathway activation of independent cellular clones that become distinct tumors. This is outlined in the results section: Results (1st paragraph under Germline variant analysis from blood and associated tumor findings subheading): “Among 10 total samples harboring blood germline WT1 variants, somatic tumor 11p15.5 LOH, and somatic tumor CTNNB1 variants, there were three sets of paired synchronous BWT (SJWILM066776, 066780, 051028) in which each of the paired tumors had somatic exon 3 CTNNB1 variants. In two cases, the tumor CTNNB1 variants were distinct (SJWLM066776 CTNNB1 p.T41A vs. p.S45del, SJWLM051028 CTNNB1 p.S45P vs. p.S45del) and in the remaining case (SJWLM066780) the CTNNB1 p.S45F variant was shared in both paired tumors (Fig. 2).”

The different CTNNB1 mutations constitute independent genetic events but are convergent evolution toward Wnt pathway activation; this suspected process and the timing of events is shown in the top panel of Figure 6.

Reviewer Response: The authors have addressed comment 17.

18. Clarify how many WT1 patients also had 11p15 alterations.

Response: Out of 14 tumor samples from the 9 patients with germline WT1 variants in the current study, all 14 (100%) were found to have 11p15.5 LOH. Thus, in the current study all tumor samples from patients with pathogenic germline variants in WT1 also had 11p15.5 LOH.

The results section has been updated in response to this comment: Results (1st paragraph under Germline variant analysis from blood and associated tumor findings subheading): “For 14 tumor samples from 9 patients with germline WT1 variants, 11p15.5 LOH (paternal uniparental disomy) determined by methylation analysis and/or the CONSERING algorithm was present in all tumors (Figure 2).”

Reviewer Response: The authors have addressed comment 18.

19. Clarify the parameters of patients with 11p15 alterations in multiple tissues and their clinical phenotypes, specifically if there were subtle clinical phenotypes.

Response: The results section has been updated to address this question: Results (2nd paragraph under 11p15.5 status is shared in synchronous BWT subheading): “Most patients with 11p15.5 LOH or 11p15.5 LOI detected in their tumor samples had no clinical features suggestive of a syndrome (Fig. 2). However, of the 9 patients with definitive 11p15.5 alterations in multiple tissues, SJWLM066781 had 11p15.5 LOI in their tumor and adjacent non-diseased kidney and had a ureteral duplication.

SJWLM069381 had 11p15.5 LOI in their tumor and adjacent non-diseased kidney tissue and had hemihypertrophy. SJWLM069390 had 11p15 LOH detected in their tumor and adjacent non-diseased kidney tissue. In addition, this patient was found to have mosaic 11p15.5 LOH in their blood as determined by whole genome sequencing (Supplementary Fig. 2). This patient was noted to have a BWSp clinical phenotype.”

Reviewer Response: The authors have addressed comment 19.

20. Figure 3 is too small to read and blurry.

Response: A high-resolution version of this figure (now Figure 2) has now been provided to the journal with the resubmitted manuscript. I believe the figure becomes blurry when it is incorporated into the reviewer *.pdf.

Reviewer Response: The authors have addressed comment 20.

21. Line 381-388 - For section, “Somatic variants are not shared in synchronous BWT”, the authors need to specify how many genes were used for pathway analysis. The Supplementary Table 1 shows 4 clusters. How were these clusters derived? Which genes were included in each set? Please discuss the method and results in greater details. How does the RNA data correlate with the methylation data for the same regions?

Response: The purpose of the pathway analysis was to determine the functional cellular pathways that were affected by somatic variants detected in bilateral Wilms tumor samples. The pathway analysis was performed using the DAVID Bioinformatics Database/Resource (<https://david.ncifcrf.gov/home.jsp>). The input for the DAVID pathway analysis consisted of the entire list of genes (with counts) determined to exhibit somatic mutations in the bilateral Wilms tumor samples as outlined in Supplementary Data 2. The functional annotation clustering is part of the DAVID algorithm and classifies highly related genes into functionally related groups. RNA and methylation data were not utilized for this analysis.

In response to this comment, former Supplementary Table 1 has now been provided as Supplementary Data 4 so the reader can click on the gene lists in the excel spreadsheet to determine the number and names of each highly-related gene that is a member of the annotated lists. For example, the WIKIPATHWAYS WP2338~miRNA biogenesis gene set includes the genes ENSG00000124571 (XPO5), ENSG00000113360 (DROSHA), ENSG00000100697 (DICER1), and ENSG00000128191 (DGCR8).

Reviewer Response: The authors have addressed comment 21. The authors should correlate this data with RNA-seq data to find out the drivers of BWT. For example, if gene is mutated but that expressed, the that gene is a passive driver.

22. For section, “Somatic variants are not shared in synchronous BWT”, based on Supplementary figure 3, chromosome 22 also showed variations. Please comment on this.

Response: Thank you for this critical observation, which was previously unappreciated but upon consideration of this comment has tied several pieces of data within the manuscript together.

The results section has been modified in response to this comment/observation: Results (3rd paragraph under Tumor somatic variants are not shared in synchronous BWT subheading):

“Comparison of tumor copy number variants showed markedly different genome wide copy number profiles in paired synchronous BWT except for SJWLM069391 (Supplementary Fig. 4-5). In addition to SJWLM069391, similar regions of chromosome 22q loss were noted in paired synchronous BWT SJWLM069399 and SJWLM066784 (Supplementary Fig. 5). Overall, chromosome 22q copy number loss was detected in 11/85 (12.9%) of BWT samples.”

This observation about recurrent chromosome 22q copy loss tied into an analysis of allele specific expression which we have now included in the manuscript:

Methods section (Total-strand RNA-seq subheading): “Integrated analysis of whole genome sequencing and total-strand RNA-seq data was performed using the previously described Cis-X method to determine genome wide allele-specific expression patterns in BWT.29”

Results section (2nd and 3rd paragraphs under 11p15.5 status and genome-wide

methylation/molecular identity in BWT subheading): "To further explore the suggestion that tumor 11p15.5 status correlates with a broader tumor molecular identity rather than only changes at the 11p15.5 locus itself, we determined genome-wide alterations in allele specific expression stratified by tumor 11p5.5 status (LOI, LOH, ROI) in the COG cohort using the Cis-X method, which integrates whole genome sequencing and total-strand RNA-seq data to determine the allele of origin for the RNA being expressed. Due to sample size limitations, we filtered results according to an unadjusted p-value of < 0.05 and a false-discovery rate p-value corrected for multiple testing of <0.25 for this analysis. As predicted, KCNQ1OT1 (FDR p=0.0046) and KCNQ1 (FDR p=0.0046) were found to have perturbation of allele-specific expression in tumor samples with 11p15.5 LOH, but not ROI or LOI, while INS-IGF2 and IGF2 showed perturbation of allele-specific expression in both samples with 11p15.5 LOH and LOI. Five genes and/or noncoding RNAs outside the 11p15.5 region were found to have disruption in the normal patterning of allele-specific expression that correlated with tumor 11p15.5 status: RNF185, SNORD116-4, C1QL3, CLEC12A, and NPAS2. RNF185, a gene located at chromosome 22q12.2 that codes for a component of the E3 ubiquitin ligase pathway, was found to have reduction to monoallelic expression in 43.3% of tumor samples with 11p15.5 LOI (FDR p=0.240) and 35.7% of tumor samples with 11p15.5 LOH (FDR p=0.041), but 0% of samples with 11p15.5 ROI (Supplementary Table 2).

We then queried for possible mechanisms of RNF185 reduction to monoallelic expression and noted recurrent chromosome 22q loss correlated with this phenomenon. Combined with the copy number data detailed above that showed recurrent chromosome 22q loss in our data set and shared chromosome 22q loss in 3 synchronous BWT pairs, we performed a focused analysis to determine the clinical and other molecular features of BWT samples with chromosome 22q loss. We found an enrichment for female biological sex, 11p15.5 LOI, and DGCR8 RNA microprocessor mutations in this cohort of 11 tumor samples with chromosome 22q loss from 8 patients (Supplementary Table 3)."

The discussion section has been updated to include discussion of data related to this comment: Discussion (9th paragraph): "Specifically, our study showed that BWT with 11p15.5 LOI and LOH commonly exhibited reduction to monoallelic expression of the ubiquitin-ligase associated gene RNF185 located at chromosome 22q12.2. The most common mechanism for RNF185 reduction to monoallelic expression was chromosome 22q copy loss. All tumors that exhibited copy loss at 22q had also had 11p15.5 LOI and 6 of these 8 patients were female. This group also included all 4 patients with DGCR8 microprocessor hotspot p.E518K mutations in the current study and included 5 patients with high-risk SIOP post-treatment histology. Allelic loss at chromosome 22q loss has been previously associated with high-risk WT.48,49 Furthermore, in depth analysis of two cases of paired WT and precursor nephrogenic rests has implicated loss of chromosome 22 in the progression from perilobar nephrogenic rests to WT.50 DGCR8 (located at chromosome 22q11.21) mutations were shown to have an extreme female predominance in previous WT studies: Wegert et. al. found that 23/26 (88.5%) and Walz et. al. found that 15/17 (88.2%) patients with somatic DGCR8 mutations were female.32,51 The possible functional significance of RNF185 in WT and the constellation of somatic mosaic 11p15.5 LOI, female biological sex, chromosome 22q loss, and DGCR8 microprocessor mutations in the development and progression of WT will be the subjects of future investigation."

Figure 6 has been modified to include chromosome 22q loss and microRNA processing gene mutations as key events in bilateral Wilms tumor development.

Reviewer Response: The authors have addressed comment 22.

23. Figure 4 is confusing and does not add to the explanation of the data. Please make a table instead and add blood status of 11p15 as well.

Response: Figure 4 has been made into Table 1. Blood status of 11p5.5 (whether H19/ICR1 methylation in blood was greater than 2 standard deviations above the healthy control cohort mean value) has now been added to Table 1.

Reviewer Response: The authors have addressed comment 23.

24. The authors mention in the method section that samples with average beta value H19/ICR1 <0.7 have normal ROI. Figure 5A,D show that all the samples from all three cohorts have normal ROI. Why did the authors make an age dependent methylation comparison of ICR1? Are the results within the

normal range or outside or just demonstrating assay or sample variability? Specifically define LOI, LOH and ROI for this figure and how it is similar or different to the ranges for the other data presented. If 11p15.5 status is shared in BWT: What is the 11p15.5 status in the normal tissue? The “established thresholds” are not well defined in the context of the somatic analysis, please clarify. Please clarify why the extent of LOH did not overlap in one case. Was that between tumors, between tumor and kidney or blood? Was this LOH or paternal uniparental isodisomy?

Response:

Thank you for this important observation and the chance to clarify our hypothesis, results, and conclusions: Results (5th paragraph under 11p15.5 status is shared in synchronous BWT subheading):

“To validate the finding of somatic mosaicism for 11p15.5 LOI detectable in peripheral blood on the cohort level, we combined identically processed and normalized H19/ICR1 methylation β values from leukocyte-derived DNA between our current BWT cohort (n=61) and a cohort of healthy community controls (n=282) and WT cancer long-term survivors (including survivors of both unilateral [n=154] and BWT [n=17]) from the St. Jude Life Cohort Study.³⁵ We hypothesized that H19/ICR1 values in peripheral blood DNA would be higher in BWT patients than healthy community controls and potentially unilateral WT patients. The peripheral blood 11p15.5 H19/ICR1 and KCNQ10T1/ICR2 β values were determined to be normally distributed in all groups using the Kolmogorov-Smirnov test. The mean H19/ICR1 peripheral blood methylation β value in the healthy community control cohort was 0.499 ± 0.0248 (mean \pm standard deviation). The mean H19/ICR1 peripheral blood methylation β value from the BWT cohort (0.534 ± 0.0298) was higher than both the unilateral WT (0.521 ± 0.0258 ; $p < 0.0001$) and healthy community control cohorts ($p < 0.0001$; Fig 3D.) Using the method of Fiala, et al.²³, we determined that 26 BWT patients (n=20 from current study; n=6 from St. Jude Life cohort) had a H19/ICR1 methylation β value greater than two standard deviations from the mean community control cohort β value (i.e., H19/ICR1 $\beta > 0.54864$; Fig. 2; Supplementary Table 1). Of these 20 patients, 17/20 (85%) were female, 18/20 (90%) had 11p15.5 LOI in their tumor, 2/20 (10%) had 11p15.5 LOH in their tumor, and 0 had 11p15.5 ROI in their tumor (Supplementary Table 1).”

Leukocyte DNA methylation is known to change with age (Terry, M.B., Delgado-Cruzata, L., Vin-Raviv, N., Wu, H.C. & Santella, R.M. DNA methylation in white blood cells: association with risk factors in epidemiologic studies. *Epigenetics* 6, 828-37 (2011). Because of the wide range of ages included in the survivor analysis, we wanted to provide a control for DNA methylation changes according to age. In doing so, we noted that bilateral survivors and patients had a strikingly different pattern in leukocyte DNA methylation at 11p15 ICR1 than unilateral patients and healthy community controls as depicted in Figure 4. The methods section of the manuscript has been updated to reflect this comment: Methods (under Long-term survivorship cohort analysis from peripheral blood DNA subheading): “Because of known changes in leukocyte global DNA methylation that occur with age, methylation at 11p15.5 H19/ICR1 was plotted according to age.⁴¹ The relationship between age and methylation from unilateral WT and BWT samples were compared against the learned model from the healthy community control population.”

Results (6th paragraph under 11p15.5 status is shared in synchronous BWT subheading): “However, because leukocyte DNA methylation is known to change with age³⁶ and the healthy control population with peripheral blood DNA available for this study was derived from cancer survivors of a higher median age than our current study population, we wanted to assess H19/ICR1 methylation as a function of age. We noted an inverse relationship between methylation at H19/ICR1 and increasing age in healthy community controls (Fig. 3E). In contrast, we detected positive correlation between increasing age and H19/ICR1 methylation in BWT patients and long-term survivors. This positive correlation was not seen in long-term survivors of unilateral WT (Fig. 3E).”

The limitations paragraph of the discussion section has been updated in response to this comment: Discussion (2nd to last paragraph): “Finally, all data in this manuscript reporting evidence of mosaic 11p15.5 H19/ICR1 gain of methylation detectable in peripheral blood are based on statistical comparison or observed changes in comparison to a normal cohort, while no patients in this manuscript exhibited frank hypermethylation above the established threshold of a β value of 0.7 at 11p15.5 H19/ICR1 detectable in DNA obtained from peripheral blood.”

The Results section has also been updated in response to this comment: “Among 19 tumors with 11p15.5 LOH determined by whole genome sequencing, the extent of 11p cnLOH overlapped both the

11p15.5 and WT1/11p13 loci in 18/19 cases (94.7%; Supplementary Figure 7). Among paired synchronous BWT from patients with pathogenic germline WT1 mutations and 11p15.5 LOH detected in each of their tumors, the extent of 11p LOH was not identical when the two tumor samples were compared. The differential extent of 11p LOH between paired synchronous BWT suggests that 11p LOH occurs as an independent genetic event in each tumor in most cases rather than having a shared clonal origin (Supplementary Figure 7)."

Reviewer Response: This has still not been addressed fully, see reviewer responses to author responses to comments 6, 7, and 16.

25. The authors propose that post-zygotic 11.p15.5 alterations lead to development of BWT. However, Figure 6 shows that 11.p15.5 alterations are responsible for both unilateral and BWT. Lines 487-505 state molecular signatures that identify BWT. The authors need to state these molecular identifiers clearly. In addition, authors state, "We noted that BWT predominantly clustered distinctly from unilateral WT and closer to non-diseased adjacent kidney tissue, consistent with previously published results (Figure 6)." However, the published results state that there were no consistent differences in DNA methylation between unilateral and bilateral tumors. Please provide a consistent explanation between these results and the published results, given shared authorship with this manuscript. What is the threshold for defining a genetic/epigenetic syndrome in these patients?

Thank you for this observation. 11p15.5 alterations are certainly present in both unilateral and bilateral WT. However, the emphasis of the current manuscript is that 11p15.5 is often somatic mosaic in patients who develop bilateral Wilms tumor and therefore describes a mechanism by which both kidneys are predisposed to tumor development.

The publication the reviewer refers to (Brzezinski, J. et al. Clinically and biologically relevant subgroups of Wilms tumour defined by genomic and epigenomic analyses. *Br J Cancer* 124, 437-446 [2021].) demonstrates two Wilms tumor subgroups (cluster A and B) according to global methylation patterns determined by the Infinium HumanMethylation450 BeadChip (Illumina) or the Infinium MethylationEPIC BeadChip (Illumina). They demonstrated that bilateral disease was significantly more prevalent in Subgroup A. Their combined results from the discovery and validation cohort analyses showed that subgroup A (n=22 total tumors) was 68% bilateral and subgroup B (n=62 total tumors) was 26% bilateral (p=0.0007; shown in Table 1 of the manuscript). Subgroup A (which was predominantly bilateral tumors) clustered closer to normal kidney than Subgroup B. As the reviewer notes, the study found that there were no differences between unilateral and bilateral samples according to methylation within each of these two clusters (e.g. there were no differences in methylation between unilateral and bilateral tumors within subgroup A). However, since the primary cluster differentiation among samples was enriched for bilateral Wilms tumors in one of the groups, essentially bilateral and unilateral Wilms tumors are clustering different from one another.

This is very consistent with results from our current study depicted in Figure 4 (unsupervised hierarchical clustering of samples according to the top 10,000 most variable probes in the methylation dataset)- which show two clusters among Wilms tumor samples (clusters 3 and 4 in the Figure). Cluster 3 is enriched for bilateral tumors and clusters closer to adjacent non-diseased kidney tissue (cluster 2), while cluster 4 is enriched for unilateral tumors. However, you can see within each cluster that there are both unilateral and bilateral tumors.

Reviewer response: The authors have not addressed the following comment- "The authors need to state these molecular identifiers clearly". The authors need to perform differential gene expression studies to show unique markers for BWT.

26. Figure 6 displays 4 tissue types in study clustered separately. Authors need to show the molecular identities of each tissue. In addition, show these distinct clusters in Supplementary Figure 7. For Figure 6: "sex" not "gender" should be reported. Why are metastases included in this Figure and the analysis?

Response: The term "gender" has now been replaced with "sex" in Figure 4 (former Figure 6) and in all figures and supplementary Figures in the manuscript. The tissue type has been clarified in the Figure 4 legend (Blood, Non-diseased kidney, Wilms tumor). The metastatic samples have now been eliminated from the analysis and this figure, and the analysis and figure for the TSNE clustering analysis.

Supplementary Figure 8 (former Supplementary Figure 7) contains only paired synchronous bilateral tumor tissues oriented in a Spearman correlation matrix from most to least similar according to methylation. Because it contains only paired tumors, no unilateral tumors, and no "normal tissues" it could not be clustered in similar manner to Figure 4.

Reviewer response: The authors have partially addressed comment 26. The authors need to show the molecular identities of each tissue and fully address the comment.

27. Figure 7: If tumor purity was initially defined as greater than 50% and then >80% it is unclear what samples throughout the paper have what levels of tumors and that influences the results. There needs to be clear assessment of that in each data set and each sample. Perhaps the sequencing data can be used as a proxy for this. Given that somatic levels of 11p15 changes are being discussed this needs to be thoroughly clarified throughout the manuscript.

Response: Please see above response to comment 10 regarding tumor purity and accompanying changes to the manuscript.

Reviewer Response: The authors need to address the reviewer response to comment 10.

28. The authors introduced pathogenic or likely pathogenic variants in the abstract. Figure 8 shows role of pathogenic variants in prezygotic stage. The authors need to add the predisposing variants to section of "3. somatic mosaic 11p15.5 loss of imprinting" based on findings in Table 1. Also please add this to the discussion.

Response: Panel 2 of now Figure 6 addresses the pathogenic germline variants in the pre-zygotic stage. The possibility of 11p15.5 loss of imprinting can also occur in patients who have pathogenic germline variants. As the reviewer pointed out, Table 1 also illustrates this possibility, although it is less frequent than 11p15.5 LOH or 11p15.5 ROI in patients with germline variants. The discussion section has been updated to reflect this comment: Discussion (3rd paragraph): "It should also be noted that tumors from patients with pre-zygotic variants in Wilms tumor or cancer predisposition genes can also exhibit 11p15.5 LOI, but this is more rare than 11p15.5 LOH or ROI in this group."

Reviewer Response: The authors have addressed comment 28.

29. In the discussion is the mechanism germline mutation plus LOH? If so, state that in the first paragraph. Does LOH occur first or after the germline mutations? How does LOH affect non-chromosome 11 germline changes?

Response: Our proposed mechanism as outlined in Figure 7 (the best example being in the context of germline WT1 pathogenic variants) is that germline mutations occur before 11p15.5 LOH. This is outlined in the 2nd and 3rd paragraphs in the Discussion section. The first paragraph of the discussion section has been updated in response to this comment: Discussion section (1st paragraph): "1) Pre-zygotic germline genetic variants readily detectable in DNA derived from peripheral blood (WT1, NYNRIN, TRIM28, BRCA complex genes) often followed by 11p15.5 LOH..."

Reviewer Response: The authors have addressed comment 29.

30. How do the blood 11p15 findings factor into the developmental model?

Response: Prior work from Coorens et. al. suggested that mosaic 11p15.5 alterations occurred in the early embryo and were enriched in mesodermal tissues, specifically the bilateral kidneys. These carefully conducted studies were performed on a multitude of tissues, but from a limited number of patients. The authors concluded that failure to detect the 11p15.5 alterations in the blood was supportive of mosaicism. We wholeheartedly agree with the findings from these studies. However, because the blood cells are also derived from the mesodermal germ layer during embryonic development, we sought additional evidence on the cohort level to support the concept of mosaicism. We reasoned that since blood cells are originally derived from the mesodermal layer that we should

also be able to detect such alterations using DNA isolated from peripheral blood leukocytes. On a population/study cohort level, we did identify such alterations that suggest somatic mosaicism, although it is exceedingly difficult to assign a label of "somatic mosaicism" to an individual case. Our results document that such alterations are likely present in peripheral blood of bilateral Wilms tumor patients and that 20 patients from the current study exhibited "low-level" H19/ICR1 hypermethylation defined by Fiala et. al as greater than two standard deviations above the mean methylation value at this locus from a normal control cohort. We attempted to perform a single-cell methylation analysis using banked peripheral blood to document this phenomenon more completely, but the existing technology did not permit such an approach and this will have to remain a future direction of our work.

Reviewer response: The authors have not addressed this comment. See previous comments 6,7, and 16 and the reviewer response to authors comments for additional information and address this comment.

31. Why does 11p15 alteration level increase with age?

Response: We hypothesize that progressive, clonal selection of blood cells/leukocytes with 11p15.5 alterations in patients with somatic mosaicism for 11p15.5 loss of imprinting leads to a higher measured H19/ICR1 methylation beta value in older subjects. This was not found in community controls. In the kidney this phenomenon has been termed clonal nephrogenesis but would be difficult to measure longitudinally. In the bloodstream, prior literature has noted that such a process may be responsible for late-onset beta thalassemia as noted in the discussion section: Discussion (7th paragraph): "Progressive selection of mosaic hematopoietic cells containing uniparental paternal isodisomy for 11p15 has been suggested as a mechanism for late onset β -thalassemia major in patients who are heterozygous carriers of pathogenic HBB variants (which is also located at 11p15); however, evidence for this phenomenon is limited overall.⁴⁷"

Reviewer response: The reference 47 cited for this comment does discuss correlation of 11p15 with age. The authors need to clarify this.

32. How do different pathologic findings in these tumors fit into the two BWT models proposed? Addition of this pathology data would be informative.

Response: The current study showed that 15/17 (88.2%) specimens with SIOP high-risk post-treatment histology (defined as blastemal predominance or diffuse anaplasia after neoadjuvant chemotherapy treatment) had 11p15.5 LOI. This observation leads to our future hypothesis that tumors with 11p15.5 LOI may have inferior radiographic response rates or oncologic outcomes in patients with BWT. This is noted as a future direction in the Discussion section of the manuscript.

Reviewer response: The authors have addressed comment 32.

33. How does "clonal nephrogenesis" result in BWT? Developmentally when does this clonality occur?

Response: Progressive selection and expansion of nephrogenic clones with normal histology and H19/ICR1 gain of methylation provides predisposition for BWT development. However, additional hits (such as loss of chromosome 22q, microRNA processing gene mutations) then occur before tumors develop. Our evidence provides support from this original clonality tracing back to the early embryo, but then subsequent expansion of these nephrogenic clones is thought to occur during kidney development.

Reviewer response: The authors need to incorporate these findings in the manuscript with relevant references.

34. How do the changes in long term survivors of the 11p15 region validate somatic mosaic findings in the BWT patients? It is unclear how the data presented support this statement.

Response: The Discussion section has been reworked in response to this comment: Discussion (6-8th paragraphs): "Identification of mosaic 11p15.5 LOI in the peripheral blood is difficult to definitively determine in an individual patient sample. Therefore, we sought to evaluate for supporting evidence of 11p15.5 H19/ICR1 hypermethylation (LOI) on the cohort level. Indeed, our analysis showed a statistically significant increase in H19/ICR1 methylation in peripheral blood from patients with 11p15.5 LOI in their tumor specimens when compared to those with 11p15.5 ROI in their tumors, further supporting the concept of epigenetic somatic mosaicism with detection of low-level gain of methylation. Because the data discussed above support 11p15.5 LOI somatic mosaicism as a major contributor to the predisposition for BWT development, we hypothesized and confirmed that patients and survivors of BWT had higher levels of H19/ICR1 methylation detectable in peripheral blood than a cohort of healthy community control subjects and unilateral WT. Using the method of Fiala et. al.²³, we determined that 20 patients from the current study had "low-level" gain of methylation detectable in peripheral blood defined as two standard deviations above the mean H19/ICR1 beta value in the community control cohort. Seventeen of these 20 patients were female and 18/20 had 11p15.5 LOI detected in their tumor sample. Fiala et. al. previously showed that 7/7 BWT patients with "low-level" H19/ICR1 hypermethylation detectable in peripheral blood were also female.²³

Because we incorporated DNA samples from a long-term survivorship cohort and because global leukocyte DNA methylation changes with age, we started this analysis by plotting peripheral blood H19/ICR1 beta values versus age. We found a positive correlation between increased age and H19/ICR1 methylation that was specific to BWT patients and long-term survivors and not found in healthy community controls or unilateral WT patients or long-term survivors. We speculate that increased methylation with age at 11p15.5 H19/ICR1 in the BWT population could be due to gradual selection/expansion of cellular clones with H19/ICR1 hypermethylation in the peripheral blood over time in a manner similar to what was described as clonal nephrogenesis in the kidney.²² Progressive selection of mosaic hematopoietic cells containing uniparental paternal isodisomy for 11p15 has been suggested as a mechanism for late onset β -thalassemia major in patients who are heterozygous carriers of pathogenic HBB variants (which is also located at 11p15); however, evidence for this phenomenon is limited overall.⁵¹ These results are very unlikely to be related to circulating tumor DNA because the positive correlation between age and H19/ICR1 methylation was also seen in BWT long-term survivors, whose DNA samples were obtained at least 5 years after cancer diagnosis. These results provide further support for the 11p15.5 LOI mosaicism hypothesis because they provide indirect evidence of expanding 11p15.5 LOI mosaic clones in peripheral blood with time.

Furthermore, our data show that 11p15.5 status can be used as a surrogate biomarker for more global methylation patterns/molecular subgroups of BWT because when samples were clustered according to broader patterns of methylation, they tended to cluster according to 11p15.5 status. Whether 11p15.5 status (ROI, LOI, LOH) correlates with volumetric or histologic response to neoadjuvant chemotherapy, event-free, or overall survival will be the subject of future clinical translational investigation. However, as a preliminary window into this question, the current study showed that 15/17 (88.2%) specimens with SIOP high-risk post-treatment histology and 15/17 (88.2%) with the adverse prognostic biomarker 1q gain had 11p15.5 LOI. The current study provides strong rationale for prospectively following outcomes in BWT patients according to tumor 11p15.5 status and this will be included as an observational biologic aim in the Children's Oncology Group BWT protocol currently under development."

Reviewer response: The authors state that, "Because the data discussed above support 11p15.5 LOI somatic mosaicism as a major contributor to the predisposition for BWT development, we hypothesized and confirmed that patients and survivors of BWT had higher levels of H19/ICR1 methylation detectable in peripheral blood than a cohort of healthy community control subjects and unilateral WT." This is not relevant as these methylation changes were in the normal range. Please address this comment

35. The data on amplified clonal mosaicism increasing over time is limited, where the data on mosaic 11p15 changes in patients with Beckwith-Wiedemann Spectrum is more plentiful. Please comment on this in the context of the discussion.

Response: We agree with this comment. We have added this observation to the the discussion section: Discussion (7th paragraph): "Progressive selection of mosaic hematopoetic cells containing uniparental paternal isodisomy for 11p15 has been suggested as a mechanism for late onset β -thalassemia major in patients who are heterozygous carriers of pathogenic HBB variants (which is also located at 11p15); however, evidence for this phenomenon is limited overall.⁴⁷"

Reviewer response: The reference cited is not relevant. Please address this comment.

36. How is 11p15 status a biomarker for global methylation patterns? Please revise this and provide the evidence to support that revision. Was it high or low stage risk? Both are stated?

Response: When tumors are clustered according to methylation M values derived from the top 10,000 most variable CpG probes in the methylation array data set (using both unsupervised hierarchical or TSNE approaches), the bilateral Wilms tumors illustrate clustering according to 11p15 status as depicted in Figure 4 and Figure 5B. We do not understand if this pertains to risk or stage of tumors; however, among tumors with SIOP high-risk histology (which is used clinically to define treatment and prognosis), the majority had 11p15 LOI. This association will need to be explored prospectively in future studies.

The comment about 11p15 status determining outcome in low-risk tumors may be confusing in this context. This observation came from a study looking specifically at unilateral Wilms tumor with several low-risk parameters (age <2 years, tumor weight < 500g, stage I, favorable histology). It has been eliminated from the Discussion section to improve clarity of the manuscript and avoid confusion.

Reviewer's response: This needs to be further clarified in the text, the revision is still unclear.

Reviewer #3:

Remarks to the Author:

I'm happy with the way the authors have addressed my comments and questions. The new manuscript is for me much improved and more balanced. I congratulate the authors on a lovely and important piece of work.

REVIEWER COMMENTS

Reviewer #1 (Remarks to the Author):

Comment: The authors have revised their manuscript extensively, including provision of new data and correction of figures and numbers of samples. Their comprehensive revision has addressed all my comments to my satisfaction. They also appear to have answered both of the other reviewers' comments in a similar expansive fashion.

Response: Thank you for your review of our manuscript and encouraging comments.

Reviewer #2 (Remarks to the Author):

Comment: The authors have revised the manuscript significantly. However, there are still several issues with the manuscript that need to be addressed. Specifically, there needs to be clarity of language and word usage for “somatic mosaicism”, “post-zygotic”, and “germline”. These terms can have different meanings and usage by oncologists and geneticists, and they need to be defined and used appropriately.

Response: Thank you for your thorough and constructive review of the manuscript. In response to this comment, we have added precise definitions of “somatic mosaicism”, “postzygotic”, and “germline” to the introduction section of the manuscript that are consistent with how they are used throughout the text: Introduction, 2nd paragraph: *“Somatic mosaicism refers to two (epi)genetically distinct populations of cells that are found in the same individual, due to a (epi)genetic variant that occurs after fertilization (i.e. postzygotic). This is in contrast to a germline (epi)genetic event, that occurs prior to fertilization and is therefore found in every cell of the body in an individual.^{16”}*

Comment: Additionally, the thresholds for LOI and LOH measurements in the different tissues need to be uniform, if the thresholds are defined one way and as previously described for tumor and kidney but then used differently looking at trends in blood, with all of the data being used to create a unified narrative, there is an issue with the narrative and the way the data are being analyzed to form the narrative. The blood findings and the changes overtime need to be reframed or removed from the results because the current presentation confounds the results. Additionally, in light of the definitions requested above, the authors need to consider and comment on the molecular findings in the blood in line with the current clinical and molecular definitions of BWS if these results are left in the manuscript.

Response: Thank you for these comments. The methylation thresholds for 11p15.5 LOI and LOH definitions are outlined under “Methylation analysis” in the Methods section and we have verified that they are uniform throughout the manuscript. The analysis of 11p15.5 *H19/ICR1* methylation in peripheral blood samples is now more explicitly labeled as an exploratory analysis for low-level methylation changes at 11p15.5 *H19/ICR1* that are within the “normal

range” and do not meet the definition of 11p15.5 LOI outlined in the manuscript. However, this analysis, and the prior publication by Fiala et. al., may provide preliminary evidence for a minor, mosaic subpopulation of leukocytes with 11p15.5 H19/ICR1 hypermethylation. Validation of this hypothesis, potentially using single-cell methylation analyses, will be a future direction of our work. Specific changes to the manuscript in response to this overall comment are outlined below in response to the itemized reviewer comments.

Comment: They also need to comment of why the blood methylation over time is included and why BWT and non-BWT patients who are not matched for age make sense for comparison. All of these issues were previously raised by the reviewer and not sufficiently addressed in the author response. Please address each of these comments as well as the remaining comments listed below as not completely addressed.

Response: In response to this comment, we have removed the analysis of blood methylation according to age/time from the manuscript. An age-matched cohort of peripheral blood DNA analyzed by 850K methylationEPIC beadchip from healthy infants, toddlers, and children is not available for analysis.

Comment: The authors state, ““Although all tumor samples from patients with pathogenic WT1 germline variants in the current study exhibited 11p15.5 LOH, prior work from our group and others in unilateral Wilms tumor samples demonstrated somatic WT1 mutations without 11p15.5 LOH.” and have referred to -

a) Murphy, A.J. et al. Forty-five patient-derived xenografts capture the clinical and biological heterogeneity of Wilms tumor. *Nat Commun* 10, 5806 (2019).

b) Gadd, S. et al. A Children's Oncology Group and TARGET initiative exploring the genetic landscape of Wilms tumor. *Nat Genet* 49, 1487-1494 (2017).

The data in these papers clearly indicate that WT1 plus LOH are not mutually exclusive. Please amend comments.

Response: The following sentences in the discussion section have been revised in response to this comment: Discussion, 2nd paragraph: *“Although all tumor samples from patients with pathogenic WT1 germline variants in the current study exhibited 11p15.5 LOH, prior work from our group and others in unilateral Wilms tumor samples demonstrated somatic WT1 pathogenic variants without accompanied 11p15.5 LOH; therefore WT1 pathogenic variants and 11p15.5 LOH often, but not always, accompany one another.”^{38,43}*

Comment: The authors need to state that the clinical evaluation and reported data may not be complete. There are several publications indicating that clinical diagnosis of BWS may be missed until after a WT is detected and as such patients may not be labeled “syndromic” in these cancer databases. Additionally, while the ranges of methylation utilized in this work and in the work cited are quite broad and may miss lower level mosaicism for methylation changes and

LOH consistent with BWSp. This is noted in the blood levels for these patients but discounted in the results and discussion. Together, this mosaicism question needs to be addressed further.

Response: The following sentence has been added to the limitations section of the manuscript in response to this comment: Discussion section, 2nd to last paragraph: *“The clinical evaluation and reported data about syndromic features may be incomplete, especially for patients who may have exhibited a subtle clinical phenotype.”*

We have also made reference to the point the reviewer is making in the Introduction section, 2nd paragraph: *“In fact, some patients with mosaic distribution of 11p15.5 abnormalities do not have overt syndromic features and are first diagnosed by detection of subtle clinical abnormalities and germline evaluation at the time of embryonal tumor presentation.”¹⁷⁻²⁰*

Comment: The authors had stated, “Samples with average b value H19/ICR1 < 0.7 and KCNQ1OT1/ICR2 > 0.3 were determined to have normal retention of imprinting (ROI).” Based on this any change in methylation within this range is normal. By stating, “We found a statistically significant increase in H19/ICR1 methylation detectable in peripheral blood in patients who had tumors with 11p15.5 LOI compared to those with retention of imprinting”, the authors are contradicting their own findings. Furthermore, several groups have demonstrated low-level mosaic LOI or LOH/UPD as molecularly diagnostic of BWSp, this text needs to be revised to reflect that these patients have molecular findings suggestive of BWSp that may fall below the threshold for detection for this molecular testing modality. The drift of the changes over time is not relevant and it is unclear why the authors are discussing it in such detail.

Response: In response to this comment, we have reframed the peripheral blood methylation analysis as an exploratory effort aimed to detect “low-level” methylation changes within the “normal range” that fall below the thresholds for 11p15.5 LOI or LOH used in this study. In the discussion, we refer to the detection of these statistically significant, but “low-level” methylation increases within the normal range as potentially indicative of a minor, mosaic population of lymphocytes with hypermethylation at *H19/ICR1*. The Fiala article is referenced to demonstrate an example of a study that demonstrated low-level methylation gain at *H19/ICR1* that falls below the threshold used to clinically diagnose BWSp and we believe it is relevant since these findings (which are mirrored in the current manuscript) were detected in patients with bilateral Wilms tumor. In response to this comment, we have eliminated the analysis of methylation according to age/time from the revised version of the manuscript.

Changes in response to this comment are found under the “Exploratory analysis of 11p15.5 H19/ICR1 methylation in peripheral blood” subheading of the results section: *“Exploratory analysis of 11p15.5 H19/ICR1 methylation in peripheral blood...”*

The above analyses demonstrated that 11p15.5 LOI was found in paired synchronous BWT that also shared tumor noncoding somatic variants. In addition, 11p15.5 LOI was found in adjacent non-diseased kidney tissue, but not found in DNA obtained from peripheral blood leukocytes. This pattern in which some, but not all, cells throughout the body are affected by a genetic or

epigenetic change that can be inferred to occur in the early embryo based on shared noncoding somatic variants is consistent with post-zygotic somatic mosaicism. However, we reasoned that, if present, post-zygotic somatic mosaicism could manifest as increased methylation at 11p15.5 H19/ICR1 in peripheral blood cells because hematopoietic progenitor cells have a mesodermal embryonic origin.³⁶ To explore whether increased 11p15.5 H19/ICR1 methylation could be detected on the cohort level in peripheral blood samples from patients with tumors bearing 11p15.5 LOI, we compared the H19/ICR1 β values from leukocyte-derived DNA from peripheral blood among patients according to the 11p15.5 status of their tumors (ROI, LOH, LOI). We found a statistically significant increase in H19/ICR1 methylation detectable in peripheral blood in patients who had tumors with 11p15.5 LOI compared to those with retention of imprinting (Fig. 3A). Of note, this statistically significant gain of methylation was low-level and in the “normal” range below the β value of 0.7 associated with established definitions for germline 11p15.5 LOI. In contrast, 7 of the adjacent non-diseased kidney samples exhibited 11p15.5 LOI with a H19/ICR1 β value > 0.7 (Fig. 3B).

To further explore the finding of increased 11p15.5 H19/ICR1 methylation found in peripheral blood of BWT patients and to determine if this was different from healthy community controls and unilateral WT patients, we combined identically processed and normalized H19/ICR1 methylation β values from leukocyte-derived DNA between our current BWT cohort (n=61) and a cohort of healthy community controls (n=282) and WT cancer long-term survivors (including survivors of both unilateral [n=154] and BWT [n=17]) from the St. Jude Life Cohort Study.³⁷ We hypothesized that H19/ICR1 values in peripheral blood DNA would be higher in BWT patients than healthy community controls and potentially unilateral WT patients. The peripheral blood 11p15.5 H19/ICR1 and KCNQ1OT1/ICR2 β values were determined to be normally distributed in all groups using the Kolmogorov-Smirnov test. The mean H19/ICR1 peripheral blood methylation β value in the healthy community control cohort was 0.499 ± 0.0248 (mean \pm standard deviation). The mean H19/ICR1 peripheral blood methylation β value from the BWT cohort (0.534 ± 0.0298) was higher than both the unilateral WT (0.521 ± 0.0258 ; $p < 0.0001$) and healthy community control cohorts ($p < 0.0001$; Fig 3D.) We determined that 26 BWT patients (n=20 from current study; n=6 from St. Jude Life cohort) had a H19/ICR1 methylation β value greater than two standard deviations from the mean community control cohort β value (i.e., H19/ICR1 $\beta > 0.54864$; Fig. 2; Supplementary Table 1). Of these 20 patients, 17/20 (85%) were female, 18/20 (90%) had 11p15.5 LOI in their tumor, 2/20 (10%) had 11p15.5 LOH in their tumor, and 0 had 11p15.5 ROI in their tumor (Supplementary Table 1).”

Response (continued): Additional changes in response to this comment are found in the 6th paragraph of the Discussion section: “Coorens et al. concluded that somatic mosaic H19/ICR1 hypermethylation is detectable in adjacent non-diseased kidney, but not in peripheral blood. Overall, our results are consistent with these findings because no peripheral blood sample in our cohort of BWT patients was found to have 11p15.5 LOI defined as an H19/ICR1 β value > 0.7. Patients with 11p15.5 LOI detected in tumor and adjacent non-diseased kidney, but not in peripheral blood can be presumed to be mosaic for 11p15.5 LOI. However, we believe it is likely there are additional patients with “low-level” mosaicism for 11p15.5. LOI in the kidney and blood

that does not achieve the thresholds used in this study. We reasoned that, because lymphocytes are derived from the embryonic mesodermal cell population, there could be evidence of “low-level” 11p15.5 H19/ICR1 increased methylation detectable in peripheral blood. Our exploratory analysis of 11p15.5 H19/ICR1 methylation in peripheral blood demonstrated that BWT patients with 11p15.5 LOI in their tumors had significantly higher peripheral blood H19/ICR1 methylation values than patients with tumors having normal 11p15.5 retention of imprinting. In addition, we hypothesized and subsequently determined that patients and survivors of BWT had higher levels of H19/ICR1 methylation detectable in peripheral blood than a cohort of healthy community control subjects and unilateral WT. While these statistically significant increases in H19/ICR1 methylation in BWT patients still fall within the “normal” range, they provide evidence that a mosaic, minor population of lymphocytes with 11p15.5 H19/ICR1 LOI may exist. These findings are unlikely to be related to circulating tumor DNA because the result was confirmed in BWT long-term survivors, whose blood DNA samples were obtained at least 5 years after cancer diagnosis. Fiala et. al. also previously demonstrated “low-level” gain of methylation detectable in peripheral blood samples from 7 patients with BWT, which they defined as two standard deviations above the mean H19/ICR1 β value in a healthy control cohort.²⁴ Because 11p15.5 H19/ICR1 LOI is a purely epigenetic phenomenon, refinements in single-cell methylation approaches in the future will be required to more definitively evaluate this hypothesis. For 11p15.5 LOH, which is a genetic change caused by copy neutral loss of heterozygosity, we were able to confirm mosaicism detectable in a peripheral blood sample using whole genome sequencing data (SJWLM069390).”

Comment: Please define frank germline, methylation changes that do not have an underlying genetic cause (ie a deletion or duplication causing the LOI, are not germline. The authors state, “Finally, the frequent detection of 11p15.5 H19/ICR1 hypermethylation at frank germline thresholds ($b > 0.7$) in adjacent non-diseased kidney tissue and associated tumors but not blood is consistent with mosaicism throughout the body”. Presence of mosaicism indicates presence of a syndrome. The authors need to mention the limitation of their genetic experts to identify the syndromic samples and what is reported in cancer data collections like COG. Additionally, the discussion of LOI in blood, kidney, and tumor need to be revised to reflect that these are post-zygotic mosaic changes and that there are thresholds for detection being defined here and just because something falls below the threshold defined, it does not mean that the post-zygotic change is not occurring.

Response: The term “frank germline” has been removed from the manuscript considering the more precise definitions of germline, somatic mosaic, and postzygotic included in this revised version of the manuscript prompted by the above reviewer comments.

The following sentence has been added to the limitations section of the manuscript in response to this comment about ascertaining syndromes in the patient population: Discussion section, 2nd to last paragraph: *“The clinical evaluation and reported data about syndromic features may be incomplete, especially for patients who may have exhibited a subtle clinical phenotype.”*

The concept that post-zygotic mosaic methylation changes at 11p15.5 *H19/ICR1* may be present but fall below the established thresholds for 11p15.5 LOI is precisely why we performed the analysis of blood samples despite the methylation values in these samples falling into the “normal” range. We have attempted to use more precise wording in the results and discussion section when referring to this analysis in the revised version of the manuscript (see response to previous reviewer comment above).

Comment: The authors need to distinguish between post-zygotic changes and somatic mosaicism. They also need to discuss the threshold they are using to define the mosaicism and not use the thresholds which are arbitrary (even though they are cited as based on previous data) to define the mechanism occurring. There is also no specific known correlation between females with low level LOI. There is not enough data in either the Fiala or this cohort to make that determination. The authors are not clearly discussing the limits of their testing and are using the limits of their testing to define mechanism. Additionally, the change in LOI over time in is further confusing the discussion. Comparing blood at a static time point in patients with BWT and without is fine but they authors are also not using age-matched datasets.

Response: In this manuscript, post-zygotic (epi)genetic changes lead to the finding of somatic mosaicism. We have updated the introduction section to more precisely define the terms “postzygotic changes” and “somatic mosaicism” that are used in the manuscript: Introduction section, 2nd paragraph: *“Somatic mosaicism refers to two (epi)genetically distinct populations of cells that are found in the same individual, due to a (epi)genetic variant that occurs after fertilization (i.e. postzygotic). This is in contrast to a germline (epi)genetic event, that occurs prior to fertilization and is therefore found in every cell of the body in an individual.”*¹⁶

Patients with 11p15.5 LOI detected in tumor and adjacent non-diseased kidney, but not in peripheral blood can be presumed to be mosaic for 11p15.5 LOI. However, we believe it is likely there are additional patients with “low-level” mosaicism for 11p15.5 LOI in the kidney and blood that does not achieve the thresholds used in this study. Current methods do not permit definitive determination of “low level” mosaicism for 11p15.5 LOI in an individual sample. However, in comparison to a control cohort, because *ICR1* values are normally distributed in our data sets, individuals with methylation *ICR1* levels greater than two standard deviations above the mean *ICR1* value in the control cohort have a $\leq 5\%$ probability of this being due to chance alone. So, the threshold of two standard deviations from the mean normal *ICR1* cohort value used in the Fiala study (and subsequently utilized in the current study for exploratory analysis of increased *H19/ICR1* low-level methylation within the “normal” range) is not arbitrary, but rather based statistics conventionally used for hypothesis testing.

The disclosure of a relationship between biologic sex and results in the manuscript was required/requested by the journal as part of the editor comments during the first revision and so we felt it was proper to report the finding that 17/20 (85%) with blood *H19/ICR1* methylation values > 2 standard deviations above the mean control population *H19/ICR1* methylation value were female and that 7/7 (100%) of such BWT patients in the Fiala study were female.

We have eliminated the data and discussion of changes in 11p15.5 *H19/ICR1* methylation according to age/time from the revised version of the manuscript.

We respectfully submit that the limitations paragraph of this manuscript is quite extensive, particularly after peer review, and was written with a genuine attempt to disclose the limitations of the samples, testing, and conclusions of the study.

Comment: It is still unclear how the authors assess LOI or LOH if tumor purity is being based off of the same data as LOI and LOH.

Response: Tumor purity was evaluated using three separate methods in the study that are detailed in the Methods section and in the Results section under *"11p15.5 status is shared in synchronous BWT."* Therefore, tumor purity was not only assessed using the same data as the determination of 11p15.5 LOH and LOI:

1. Histologic assessment by a pathologist to verify that a frozen biospecimen contained greater than 50% viable tumor. Samples with lower than 50% viable tumor were not included in the study (all samples).
2. Tumor purity was computationally estimated from the methylation array data using a deconvolution-based approach (all samples).
3. Tumor purity was computationally estimated from whole genome sequencing data reads in the COG specimens. Purity determined by methods 2 and 3 was found to strongly correlate (Pearson $r=0.78$, $p=3.2e-13$).

We acknowledge the reviewer comment that tumor purity is a confounder of 11p15.5 LOI and LOH assessment by methylation array and this is stated in the *"11p15.5 status is shared in synchronous BWT"* of the Results section and the limitations paragraph of the Discussion section. We have also updated the limitations paragraph of the discussion section to state that *"Because tumor purity correlated positively with ICR1 methylation and negatively with ICR2 methylation, samples of perfect purity would be expected to exhibit increased likelihood of 11p15.5 LOI or LOH; therefore, our results could underestimate the number of tumor samples with these findings."*

Comment: The authors have partially addressed comment 13. The authors need to mention "state is not known" for patients that show mosaicism and indicate the limitation of the data in COG and the St. Jude Cohort.

Response: As noted above, we have added a sentence to the limitations paragraph of the discussion section detailing the that *"The clinical evaluation and reported data about syndromic*

features may be incomplete, especially for patients who may have exhibited a subtle clinical phenotype.”

Comment: The Supplementary figures 9-10, do not show RNA-seq data for normal kidney samples. This concern has not been addressed.

Response: As we have attempted to indicate in Figure 1, RNA from non-diseased kidney tissue and peripheral blood is not available in this study. Therefore, we are unfortunately unable to incorporate normal kidney into the RNA-seq analysis.

Comment: The survivorship data does not add anything to the manuscript. Given that the results fall below the threshold of what is the point of including these data? If they are going to be included, then the authors need to refine the thresholds for the LOI findings throughout the manuscript. It is unclear what the point of showing trends that fall below the level of significance for the rest of the study. What does “frank methylation” mean?

Response: As the reviewer mentioned above, “low-level” changes in methylation could represent somatic mosaicism, but where the somatic mosaic changes fall below established thresholds used to define clinical syndromes. These “low-level” thresholds would be, by definition, within the “normal” range. The incorporation of the survivorship data allowed us to compare bilateral WT patients, and unilateral WT patients to a normal control population to further explore “low-level” methylation gains within the normal range. Nevertheless, this has been more clearly defined as an exploratory analysis in the revised version of the manuscript, and the analysis of methylation versus age/time has been eliminated considering the reviewer remarks. The term “frank methylation” has been eliminated from the manuscript as part of edits aimed to increase precision of language regarding methylation changes in response to the above reviewer comments.

Comment: The authors have addressed comment 21. The authors should correlate this data with RNA-seq data to find out the drivers of BWT. For example, if gene is mutated but that expressed, the that gene is a passive driver.

Response: We respectfully disagree with this reviewer comment. We do not believe that defining passenger versus driver somatic mutations can be definitively determined without functional studies. For example, *CTNNB1* mutations in exon 3 are recurrent gain of function mutations in this data set that are not predicted to increase the transcript level of *CTNNB1*. The primary purpose of identifying somatic variants in the BWT samples was to compare somatic variants among paired bilateral tumors. Determining the functional significance of passenger versus driver somatic mutations is beyond the scope of the current study aims.

Comment: The authors have not addressed the following comment- “The authors need to state these molecular identifiers clearly”. The authors need to perform differential gene expression studies to show unique markers for BWT.

Response: The molecular identifiers have now been more clearly incorporated into the Figure 4 legend. Differential gene expression studies have been performed using RNA-seq data from Figure 4 samples to compare Bilateral to Unilateral WT. The following differentially expressed genes were determined to be upregulated in unilateral compared to bilateral WT; no genes were determined to be significantly upregulated in bilateral compared to unilateral WT. This list has been included as Supplementary Table 2.

	Type	logFC (Bilateral-Unilateral)	P.Value	adj.P.Val
PPEF2	protein_coding	-2.176533241	7.07E-09	3.81E-05
METTL12	protein_coding	-1.442306213	1.40E-12	3.03E-08
SSTR5-AS1	processed_transcript	-1.273871276	6.27E-07	0.00112632
SNORD3C	lincRNA	-1.217415994	9.61E-08	0.00031741
SCARNA17	antisense	-1.146609211	1.09E-09	1.17E-05
SNORD3D	lincRNA	-1.141392927	2.57E-09	1.84E-05
RNU12	lincRNA	-1.128456629	1.18E-07	0.00031741
7SK	lincRNA	-1.106597589	5.82E-08	0.00025101
EIF4A1	protein_coding	-1.012750053	1.87E-07	0.00043002
SNORD3B-2	lincRNA	-0.988668468	1.09E-07	0.00031741
SNHG3	sense_intronic	-0.916016679	5.39E-07	0.00105589
RMRP	lincRNA	-0.793019041	1.99E-07	0.00043002

Comment: The authors have partially addressed comment 26. The authors need to show the molecular identities of each tissue and fully address the comment.

Response: The molecular identifiers have now been more clearly incorporated into the Figure 4 legend.

Comment: The reference 47 cited for this comment does discuss correlation of 11p15 with age. The authors need to clarify this.

Response: This reference and the analysis of methylation with age/time have been eliminated from the revised version of the manuscript based on the above reviewer comments.

Comment: How does “clonal nephrogenesis” result in BWT? Developmentally when does this

clonality occur?

Previous Response: Progressive selection and expansion of nephrogenic clones with normal histology and H19/ICR1 gain of methylation provides predisposition for BWT development. However, additional hits (such as loss of chromosome 22q, microRNA processing gene mutations) then occur before tumors develop. Our evidence provides support from this original clonality tracing back to the early embryo, but then subsequent expansion of these nephrogenic clones is thought to occur during kidney development.

Comment (Continued): The authors need to incorporate these findings in the manuscript with relevant references.

Response: We have updated the language in the discussion section in response to this comment, with reference of the paper by Coorens, et al. Discussion, 5th paragraph: *“Also in their study, the H19/ICR1 hypermethylation was detected in adjacent non-diseased kidney tissue and associated with clonal expansion of histologically normal renal cells deemed “clonal nephrogenesis.” This clonal expansion of nephrogenic cells is thought to occur during kidney development, and provides the precursor cell population for multifocal and BWT development.”*

The process of clonal nephrogenesis being followed by 2nd hits including chromosome 22q loss and/or microRNA processing gene mutations is outlined in Figure 6, which summarizes the findings of the study.

Comment: The authors state that, “Because the data discussed above support 11p15.5 LOI somatic mosaicism as a major contributor to the predisposition for BWT development, we hypothesized and confirmed that patients and survivors of BWT had higher levels of H19/ICR1 methylation detectable in peripheral blood than a cohort of healthy community control subjects and unilateral WT.” This is not relevant as these methylation changes were in the normal range. Please address this comment

Response: As the reviewer mentioned above, “low-level” changes in methylation could represent somatic mosaicism, but where the somatic mosaic changes fall below established thresholds used to define clinical syndromes. These “low-level” thresholds would be, by definition, within the “normal” range. Nevertheless, we have revised the manuscript to clearly indicate that the blood sample analysis is an exploratory analysis of potential “low-level” methylation changes that fall within the normal range as noted above.

Comment: The data on amplified clonal mosaicism increasing over time is limited, where the data on mosaic 11p15 changes in patients with Beckwith-Wiedemann Spectrum is more plentiful. Please comment on this in the context of the discussion.

Previous Response: We agree with this comment. We have added this observation to the the discussion section: Discussion (7th paragraph): “Progressive selection of mosaic hematopoietic cells containing uniparental paternal isodisomy for 11p15 has been suggested as a mechanism

for late onset β -thalassemia major in patients who are heterozygous carriers of pathogenic HBB variants (which is also located at 11p15); however, evidence for this phenomenon is limited overall.⁴⁷”

Comment (Continued): The reference cited is not relevant. Please address this comment.

Response: The language about progressive selection of mosaic hematopoietic cells, this reference, and the analysis of methylation versus age/time have all been eliminated from the manuscript in response to the above reviewer comments.

Comment: How is 11p15 status a biomarker for global methylation patterns? Please revise this and provide the evidence to support that revision. Was it high or low stage risk? Both are stated?

Previous Response: When tumors are clustered according to methylation M values derived from the top 10,000 most variable CpG probes in the methylation array data set (using both unsupervised hierarchical or TSNE approaches), the bilateral Wilms tumors illustrate clustering according to 11p15 status as depicted in Figure 4 and Figure 5B. We do not understand if this pertains to risk or stage of tumors; however, among tumors with SIOP high-risk histology (which is used clinically to define treatment and prognosis), the majority had 11p15 LOI. This association will need to be explored prospectively in future studies.

The comment about 11p15 status determining outcome in low-risk tumors may be confusing in this context. This observation came from a study looking specifically at unilateral Wilms tumor with several low-risk parameters (age <2 years, tumor weight < 500g, stage I, favorable histology). It has been eliminated from the Discussion section to improve clarity of the manuscript and avoid confusion.

Comment (Continued): This needs to be further clarified in the text, the revision is still unclear.

Response: We have updated the language in the discussion section to clarify the conclusion that 11p15.5 status is a surrogate biomarker of more global methylation status: Discussion section, 7th paragraph: *“Furthermore, our data show that 11p15.5 status can be used as a surrogate biomarker for more global methylation patterns/molecular subgroups of BWT because when samples were clustered according to broader, genome-wide patterns of methylation, they tended to cluster according to 11p15.5 status.”*

Reviewer #3 (Remarks to the Author):

I'm happy with the way the authors have addressed my comments and questions. The new manuscript is for me much improved and more balanced. I congratulate the authors on a lovely and important piece of work.

Response: Thank you for your review of our manuscript and encouraging comments.

Reviewers' Comments:

Reviewer #2:

Remarks to the Author:

Comment: The authors have revised the manuscript significantly. However, there are still several issues with the manuscript that need to be addressed. Specifically, there needs to be clarity of language and word usage for "somatic mosaicism", "post-zygotic", and "germline". These terms can have different meanings and usage by oncologists and geneticists, and they need to be defined and used appropriately.

Response: Thank you for your thorough and constructive review of the manuscript. In response to this comment, we have added precise definitions of "somatic mosaicism", "postzygotic", and "germline" to the introduction section of the manuscript that are consistent with how they are used throughout the text: Introduction, 2nd paragraph: "Somatic mosaicism refers to two (epi)genetically distinct populations of cells that are found in the same individual, due to a (epi)genetic variant that occurs after fertilization (i.e. postzygotic). This is in contrast to a germline (epi)genetic event, that occurs prior to fertilization and is therefore found in every cell of the body in an individual.¹⁶"

Reviewer's response: The authors have addressed this comment.

Comment: Additionally, the thresholds for LOI and LOH measurements in the different tissues need to be uniform, if the thresholds are defined one way and as previously described for tumor and kidney but then used differently looking at trends in blood, with all of the data being used to create a unified narrative, there is an issue with the narrative and the way the data are being analyzed to form the narrative. The blood findings and the changes overtime need to be reframed or removed from the results because the current presentation confounds the results. Additionally, in light of the definitions requested above, the authors need to consider and comment on the molecular findings in the blood in line with the current clinical and molecular definitions of BWS if these results are left in the manuscript.

Response: Thank you for these comments. The methylation thresholds for 11p15.5 LOI and LOH definitions are outlined under "Methylation analysis" in the Methods section and we have verified that they are uniform throughout the manuscript. The analysis of 11p15.5 H19/ICR1 methylation in peripheral blood samples is now more explicitly labeled as an exploratory analysis for low-level methylation changes at 11p15.5 H19/ICR1 that are within the "normal range" and do not meet the definition of 11p15.5 LOI outlined in the manuscript. However, this analysis, and the prior publication by Fiala et. al., may provide preliminary evidence for a minor, mosaic subpopulation of leukocytes with 11p15.5 H19/ICR1 hypermethylation. Validation of this hypothesis, potentially using single-cell methylation analyses, will be a future direction of our work. Specific changes to the manuscript in response to this overall comment are outlined below in response to the itemized reviewer comments.

Reviewer's response: The authors have addressed this comment.

Comment: They also need to comment of why the blood methylation over time is included and why BWT and non-BWT patients who are not matched for age make sense for comparison. All of these issues were previously raised by the reviewer and not sufficiently addressed in the author response. Please address each of these comments as well as the remaining comments listed below as not completely addressed.

Response: In response to this comment, we have removed the analysis of blood methylation according to age/time from the manuscript. An age-matched cohort of peripheral blood DNA analyzed by 850K methylationEPIC beadchip from healthy infants, toddlers, and children is not available for analysis.

Reviewer's response: The authors have addressed this comment.

Comment: The authors state, “Although all tumor samples from patients with pathogenic WT1 germline variants in the current study exhibited 11p15.5 LOH, prior work from our group and others in unilateral Wilms tumor samples demonstrated somatic WT1 mutations without 11p15.5 LOH.” and have referred to -

a) Murphy, A.J. et al. Forty-five patient-derived xenografts capture the clinical and biological heterogeneity of Wilms tumor. *Nat Commun* 10, 5806 (2019).

b) Gadd, S. et al. A Children's Oncology Group and TARGET initiative exploring the genetic landscape of Wilms tumor. *Nat Genet* 49, 1487-1494 (2017).

The data in these papers clearly indicate that WT1 plus LOH are not mutually exclusive. Please amend comments.

Response: The following sentences in the discussion section have been revised in response to this comment: Discussion, 2nd paragraph: “Although all tumor samples from patients with pathogenic WT1 germline variants in the current study exhibited 11p15.5 LOH, prior work from our group and others in unilateral Wilms tumor samples demonstrated somatic WT1 pathogenic variants without accompanied 11p15.5 LOH; therefore WT1 pathogenic variants and 11p15.5 LOH often, but not always, accompany one another.38,43”

Reviewer's response: The authors have addressed this comment.

Comment: The authors need to state that the clinical evaluation and reported data may not be complete. There are several publications indicating that clinical diagnosis of BWS may be missed until after a WT is detected and as such patients may not be labeled “syndromic” in these cancer databases. Additionally, while the ranges of methylation utilized in this work and in the work cited are quite broad and may miss lower level mosaicism for methylation changes and LOH consistent with BWSp. This is noted in the blood levels for these patients but discounted in the results and discussion. Together, this mosaicism question needs to be addressed further.

Response: The following sentence has been added to the limitations section of the manuscript in response to this comment: Discussion section, 2nd to last paragraph: “The clinical evaluation and reported data about syndromic features may be incomplete, especially for patients who may have exhibited a subtle clinical phenotype.”

We have also made reference to the point the reviewer is making in the Introduction section, 2nd paragraph: “In fact, some patients with mosaic distribution of 11p15.5 abnormalities do not have overt syndromic features and are first diagnosed by detection of subtle clinical abnormalities and germline evaluation at the time of embryonal tumor presentation.17-20”

Reviewer's response: The authors have addressed this comment by stating the limitation of their work.

Comment: The authors had stated, “Samples with average b value H19/ICR1 < 0.7 and KCNQ1OT1/ICR2 > 0.3 were determined to have normal retention of imprinting (ROI).” Based on this any change in methylation within this range is normal. By stating, “We found a statistically significant increase in H19/ICR1 methylation detectable in peripheral blood in patients who had tumors with 11p15.5 LOI compared to those with retention of imprinting”, the authors are contradicting their own findings. Furthermore, several groups have demonstrated low-level mosaic LOI or LOH/UPD as molecularly diagnostic of BWSp, this text needs to be revised to reflect that these patients have molecular findings suggestive of BWSp that may fall below the threshold for detection for this molecular testing modality. The drift of the changes over time is not relevant and it is unclear why the authors are discussing it in such detail.

Response: In response to this comment, we have reframed the peripheral blood methylation analysis as an exploratory effort aimed to detect “low-level” methylation changes within the “normal range” that fall below the thresholds for 11p15.5 LOI or LOH used in this study. In the discussion, we refer to the detection of these statistically significant, but “low-level” methylation increases within the normal range as potentially indicative of a minor, mosaic population of lymphocytes with hypermethylation at H19/ICR1. The Fiala article is referenced to demonstrate an example of a study that demonstrated low-level methylation gain at H19/ICR1 that falls below the threshold used to clinically diagnose BWSp and we believe it is relevant since these findings (which are mirrored in the current manuscript) were detected in patients with bilateral Wilms tumor. In response to this comment, we have eliminated the analysis of methylation according to age/time from the revised version of the manuscript.

Changes in response to this comment are found under the “Exploratory analysis of 11p15.5 H19/ICR1 methylation in peripheral blood” subheading of the results section: “Exploratory analysis of 11p15.5 H19/ICR1 methylation in peripheral blood...”

The above analyses demonstrated that 11p15.5 LOI was found in paired synchronous BWT that also shared tumor noncoding somatic variants. In addition, 11p15.5 LOI was found in adjacent non-diseased kidney tissue, but not found in DNA obtained from peripheral blood leukocytes. This pattern in which some, but not all, cells throughout the body are affected by a genetic or epigenetic change that can be inferred to occur in the early embryo based on shared noncoding somatic variants is consistent with post-zygotic somatic mosaicism. However, we reasoned that, if present, post-zygotic somatic mosaicism could manifest as increased methylation at 11p15.5 H19/ICR1 in peripheral blood cells because hematopoietic progenitor cells have a mesodermal embryonic origin.³⁶ To explore whether increased 11p15.5 H19/ICR1 methylation could be detected on the cohort level in peripheral blood samples from patients with tumors bearing 11p15.5 LOI, we compared the H19/ICR1 β values from leukocyte-derived DNA from peripheral blood among patients according to the 11p15.5 status of their tumors (ROI, LOH, LOI). We found a statistically significant increase in H19/ICR1 methylation detectable in peripheral blood in patients who had tumors with 11p15.5 LOI compared to those with retention of imprinting (Fig. 3A). Of note, this statistically significant gain of methylation was low-level and in the “normal” range below the β value of 0.7 associated with established definitions for germline 11p15.5 LOI. In contrast, 7 of the adjacent non-diseased kidney samples exhibited 11p15.5 LOI with a H19/ICR1 β value > 0.7 (Fig. 3B).

To further explore the finding of increased 11p15.5 H19/ICR1 methylation found in peripheral blood of BWT patients and to determine if this was different from healthy community controls and unilateral WT patients, we combined identically processed and normalized H19/ICR1 methylation β values from leukocyte-derived DNA between our current BWT cohort (n=61) and a cohort of healthy community controls (n=282) and WT cancer long-term survivors (including survivors of both unilateral [n=154] and BWT [n=17]) from the St. Jude Life Cohort Study.³⁷ We hypothesized that H19/ICR1 values in peripheral blood DNA would be higher in BWT patients than healthy community controls and potentially unilateral WT patients. The peripheral blood 11p15.5 H19/ICR1 and KCNQ1OT1/ICR2 β values were determined to be normally distributed in all groups using the Kolmogorov-Smirnov test. The mean H19/ICR1 peripheral blood methylation β value in the healthy community control cohort was 0.499 ± 0.0248 (mean \pm standard deviation). The mean H19/ICR1 peripheral blood methylation β value from the BWT cohort (0.534 ± 0.0298) was higher than both the unilateral WT (0.521 ± 0.0258 ; $p < 0.0001$) and healthy community control cohorts ($p < 0.0001$; Fig 3D.) We determined that 26 BWT patients (n=20 from current study; n=6 from St. Jude Life cohort) had a H19/ICR1 methylation β value greater than two standard deviations from the mean community control cohort β value (i.e., H19/ICR1 $\beta > 0.54864$; Fig. 2; Supplementary Table 1). Of these 20 patients, 17/20 (85%) were female, 18/20 (90%) had 11p15.5 LOI in their tumor, 2/20 (10%) had 11p15.5 LOH in their tumor, and 0 had 11p15.5 ROI in their tumor (Supplementary Table 1).”

Response (continued): Additional changes in response to this comment are found in the 6th paragraph of the Discussion section: “Coorens et al. concluded that somatic mosaic H19/ICR1 hypermethylation is detectable in adjacent non-diseased kidney, but not in peripheral blood. Overall, our results are

consistent with these findings because no peripheral blood sample in our cohort of BWT patients was found to have 11p15.5 LOI defined as an H19/ICR1 β value > 0.7 . Patients with 11p15.5 LOI detected in tumor and adjacent non-diseased kidney, but not in peripheral blood can be presumed to be mosaic for 11p15.5 LOI. However, we believe it is likely there are additional patients with “low-level” mosaicism for 11p15.5. LOI in the kidney and blood that does not achieve the thresholds used in this study. We reasoned that, because lymphocytes are derived from the embryonic mesodermal cell population, there could be evidence of “low-level” 11p15.5 H19/ICR1 increased methylation detectable in peripheral blood. Our exploratory analysis of 11p15.5 H19/ICR1 methylation in peripheral blood demonstrated that BWT patients with 11p15.5 LOI in their tumors had significantly higher peripheral blood H19/ICR1 methylation values than patients with tumors having normal 11p15.5 retention of imprinting. In addition, we hypothesized and subsequently determined that patients and survivors of BWT had higher levels of H19/ICR1 methylation detectable in peripheral blood than a cohort of healthy community control subjects and unilateral WT. While these statistically significant increases in H19/ICR1 methylation in BWT patients still fall within the “normal” range, they provide evidence that a mosaic, minor population of lymphocytes with 11p15.5 H19/ICR1 LOI may exist. These findings are unlikely to be related to circulating tumor DNA because the result was confirmed in BWT long-term survivors, whose blood DNA samples were obtained at least 5 years after cancer diagnosis. Fiala et. al. also previously demonstrated “low-level” gain of methylation detectable in peripheral blood samples from 7 patients with BWT, which they defined as two standard deviations above the mean H19/ICR1 β value in a healthy control cohort.²⁴ Because 11p15.5 H19/ICR1 LOI is a purely epigenetic phenomenon, refinements in single-cell methylation approaches in the future will be required to more definitively evaluate this hypothesis. For 11p15.5 LOH, which is a genetic change caused by copy neutral loss of heterozygosity, we were able to confirm mosaicism detectable in a peripheral blood sample using whole genome sequencing data (SJWLM069390).”

Reviewer’s response: If the authors are planning to keep this exploratory analysis in the manuscript, they need to further define the “new exploratory thresholds” they are using for the low level analysis and clarify why they did not use such thresholds for the larger study. Fiala et. al. are using normal range as $50.00\% \pm 1.12\%$ whereas the normal range for this study is “average β value H19/ICR1 < 0.7 and KCNQ1OT1/ICR2 > 0.3 were determined to have normal retention of imprinting (ROI)”. The authors need to provide something similar. Additionally a minor point, the scales of Y-axis in Figure 3 are not uniform.

Comment: Please define frank germline, methylation changes that do not have an underlying genetic cause (ie a deletion or duplication causing the LOI, are not germline. The authors state, “Finally, the frequent detection of 11p15.5 H19/ICR1 hypermethylation at frank germline thresholds ($b > 0.7$) in adjacent non-diseased kidney tissue and associated tumors but not blood is consistent with mosaicism throughout the body”. Presence of mosaicism indicates presence of a syndrome. The authors need to mention the limitation of their genetic experts to identify the syndromic samples and what is reported in cancer data collections like COG. Additionally, the discussion of LOI in blood, kidney, and tumor need to be revised to reflect that these are post-zygotic mosaic changes and that there are thresholds for detection being defined here and just because something falls below the threshold defined, it does not mean that the post-zygotic change is not occurring.

Response: The term “frank germline” has been removed from the manuscript considering the more precise definitions of germline, somatic mosaic, and postzygotic included in this revised version of the manuscript prompted by the above reviewer comments.

The following sentence has been added to the limitations section of the manuscript in response to this comment about ascertaining syndromes in the patient population: Discussion section, 2nd to last paragraph: “The clinical evaluation and reported data about syndromic features may be incomplete, especially for patients who may have exhibited a subtle clinical phenotype.”

The concept that post-zygotic mosaic methylation changes at 11p15.5 H19/ICR1 may be present but fall below the established thresholds for 11p15.5 LOI is precisely why we performed the analysis of blood samples despite the methylation values in these samples falling into the "normal" range. We have attempted to use more precise wording in the results and discussion section when referring to this analysis in the revised version of the manuscript (see response to previous reviewer comment above).

Reviewer's response: The authors have addressed this comment by stating the limitation of their work.

Comment: The authors need to distinguish between post-zygotic changes and somatic mosaicism. They also need to discuss the threshold they are using to define the mosaicism and not use the thresholds which are arbitrary (even though they are cited as based on previous data) to define the mechanism occurring. There is also no specific known correlation between females with low level LOI. There is not enough data in either the Fiala or this cohort to make that determination. The authors are not clearly discussing the limits of their testing and are using the limits of their testing to define mechanism. Additionally, the change in LOI over time in is further confusing the discussion. Comparing blood at a static time point in patients with BWT and without is fine but they authors are also not using age-matched datasets.

Response: In this manuscript, post-zygotic (epi)genetic changes lead to the finding of somatic mosaicism. We have updated the introduction section to more precisely define the terms "postzygotic changes" and "somatic mosaicism" that are used in the manuscript: Introduction section, 2nd paragraph: "Somatic mosaicism refers to two (epi)genetically distinct populations of cells that are found in the same individual, due to a (epi)genetic variant that occurs after fertilization (i.e. postzygotic). This is in contrast to a germline (epi)genetic event, that occurs prior to fertilization and is therefore found in every cell of the body in an individual.¹⁶"

Patients with 11p15.5 LOI detected in tumor and adjacent non-diseased kidney, but not in peripheral blood can be presumed to be mosaic for 11p15.5 LOI. However, we believe it is likely there are additional patients with "low-level" mosaicism for 11p15.5. LOI in the kidney and blood that does not achieve the thresholds used in this study. Current methods do not permit definitive determination of "low level" mosaicism for 11p15.5 LOI in an individual sample. However, in comparison to a control cohort, because ICR1 values are normally distributed in our data sets, individuals with methylation ICR1 levels greater than two standard deviations above the mean ICR1 value in the control cohort have a $\leq 5\%$ probability of this being due to chance alone. So, the threshold of two standard deviations from the mean normal ICR1 cohort value used in the Fiala study (and subsequently utilized in the current study for exploratory analysis of increased H19/ICR1 low-level methylation within the "normal" range) is not arbitrary, but rather based statistics conventionally used for hypothesis testing.

The disclosure of a relationship between biologic sex and results in the manuscript was required/requested by the journal as part of the editor comments during the first revision and so we felt it was proper to report the finding that 17/20 (85%) with blood H19/ICR1 methylation values > 2 standard deviations above the mean control population H19/ICR1 methylation value were female and that 7/7 (100%) of such BWT patients in the Fiala study were female.

We have eliminated the data and discussion of changes in 11p15.5 H19/ICR1 methylation according to age/time from the revised version of the manuscript.

We respectfully submit that the limitations paragraph of this manuscript is quite extensive, particularly after peer review, and was written with a genuine attempt to disclose the limitations of the samples, testing, and conclusions of the study.

Reviewer's response: The authors have addressed this comment.

Comment: It is still unclear how the authors assess LOI or LOH if tumor purity is being based off of the same data as LOI and LOH.

Response: Tumor purity was evaluated using three separate methods in the study that are detailed in the Methods section and in the Results section under "11p15.5 status is shared in synchronous BWT." Therefore, tumor purity was not only assessed using the same data as the determination of 11p15.5 LOH and LOI:

1. Histologic assessment by a pathologist to verify that a frozen biospecimen contained greater than 50% viable tumor. Samples with lower than 50% viable tumor were not included in the study (all samples).
2. Tumor purity was computationally estimated from the methylation array data using a deconvolution-based approach (all samples).
3. Tumor purity was computationally estimated from whole genome sequencing data reads in the COG specimens. Purity determined by methods 2 and 3 was found to strongly correlate (Pearson $r=0.78$, $p=3.2e-13$).

We acknowledge the reviewer comment that tumor purity is a confounder of 11p15.5 LOI and LOH assessment by methylation array and this is stated in the "11p15.5 status is shared in synchronous BWT" of the Results section and the limitations paragraph of the Discussion section. We have also updated the limitations paragraph of the discussion section to state that "Because tumor purity correlated positively with ICR1 methylation and negatively with ICR2 methylation, samples of perfect purity would be expected to exhibit increased likelihood of 11p15.5 LOI or LOH; therefore, our results could underestimate the number of tumor samples with these findings."

Reviewer's response: The authors have addressed this comment by stating the limitation of their work.

Comment: The authors have partially addressed comment 13. The authors need to mention "state is not known" for patients that show mosaicism and indicate the limitation of the data in COG and the St. Jude Cohort.

Response: As noted above, we have added a sentence to the limitations paragraph of the discussion section detailing the that "The clinical evaluation and reported data about syndromic features may be incomplete, especially for patients who may have exhibited a subtle clinical phenotype."

Reviewer's response: The authors have addressed this comment.

Comment: The Supplementary figures 9-10, do not show RNA-seq data for normal kidney samples. This concern has not been addressed.

Response: As we have attempted to indicate in Figure 1, RNA from non-diseased kidney tissue and peripheral blood is not available in this study. Therefore, we are unfortunately unable to incorporate normal kidney into the RNA-seq analysis.

Reviewer's response: The authors have addressed this comment.

Comment: The survivorship data does not add anything to the manuscript. Given that the results fall below the threshold of what is the point of including these data? If they are going to be included, then

the authors need to refine the thresholds for the LOI findings throughout the manuscript. It is unclear what the point of showing trends that fall below the level of significance for the rest of the study. What does "frank methylation" mean?

Response: As the reviewer mentioned above, "low-level" changes in methylation could represent somatic mosaicism, but where the somatic mosaic changes fall below established thresholds used to define clinical syndromes. These "low-level" thresholds would be, by definition, within the "normal" range. The incorporation of the survivorship data allowed us to compare bilateral WT patients, and unilateral WT patients to a normal control population to further explore "low-level" methylation gains within the normal range. Nevertheless, this has been more clearly defined as an exploratory analysis in the revised version of the manuscript, and the analysis of methylation versus age/time has been eliminated considering the reviewer remarks. The term "frank methylation" has been eliminated from the manuscript as part of edits aimed to increase precision of language regarding methylation changes in response to the above reviewer comments.

Reviewer's response: The authors have addressed this comment.

Comment: The authors have addressed comment 21. The authors should correlate this data with RNA-seq data to find out the drivers of BWT. For example, if gene is mutated but that expressed, the that gene is a passive driver.

Response: We respectfully disagree with this reviewer comment. We do not believe that defining passenger versus driver somatic mutations can be definitively determined without functional studies. For example, CTNNB1 mutations in exon 3 are recurrent gain of function mutations in this data set that are not predicted to increase the transcript level of CTNNB1. The primary purpose of identifying somatic variants in the BWT samples was to compare somatic variants among paired bilateral tumors. Determining the functional significance of passenger versus driver somatic mutations is beyond the scope of the current study aims.

Reviewer's response: The authors have addressed this comment.

Comment: The authors have not addressed the following comment- "The authors need to state these molecular identifiers clearly". The authors need to perform differential gene expression studies to show unique markers for BWT.

Response: The molecular identifiers have now been more clearly incorporated into the Figure 4 legend. Differential gene expression studies have been performed using RNA-seq data from Figure 4 samples to compare Bilateral to Unilateral WT. The following differentially expressed genes were determined to be upregulated in unilateral compared to bilateral WT; no genes were determined to be significantly upregulated in bilateral compared to unilateral WT. This list has been included as Supplementary Table 2.

Type	logFC (Bilateral-Unilateral)	P.Value	adj.P.Val
PPEF2 protein_coding	-2.176533241	7.07E-09	3.81E-05
METTL12 protein_coding	-1.442306213	1.40E-12	3.03E-08
SSTR5-AS1 processed_transcript	-1.273871276	6.27E-07	0.00112632
SNORD3C lincRNA	-1.217415994	9.61E-08	0.00031741
SCARNA17 antisense	-1.146609211	1.09E-09	1.17E-05
SNORD3D lincRNA	-1.141392927	2.57E-09	1.84E-05
RNU12 lincRNA	-1.128456629	1.18E-07	0.00031741
7SK lincRNA	-1.106597589	5.82E-08	0.00025101

EIF4A1 protein_coding -1.012750053 1.87E-07 0.00043002
SNORD3B-2 lincRNA -0.988668468 1.09E-07 0.00031741
SNHG3 sense_intronic -0.916016679 5.39E-07 0.00105589
RMRP lincRNA -0.793019041 1.99E-07 0.00043002

Reviewer's response: The authors have addressed this comment.

Comment: The authors have partially addressed comment 26. The authors need to show the molecular identities of each tissue and fully address the comment.

Response: The molecular identifiers have now been more clearly incorporated into the Figure 4 legend.

Reviewer's response: The authors have addressed this comment.

Comment: The reference 47 cited for this comment does discuss correlation of 11p15 with age. The authors need to clarify this.

Response: This reference and the analysis of methylation with age/time have been eliminated from the revised version of the manuscript based on the above reviewer comments.

Reviewer's response: The authors have addressed this comment.

Comment: How does "clonal nephrogenesis" result in BWT? Developmentally when does this clonality occur?

Previous Response: Progressive selection and expansion of nephrogenic clones with normal histology and H19/ICR1 gain of methylation provides predisposition for BWT development. However, additional hits (such as loss of chromosome 22q, microRNA processing gene mutations) then occur before tumors develop. Our evidence provides support from this original clonality tracing back to the early embryo, but then subsequent expansion of these nephrogenic clones is thought to occur during kidney development.

Comment (Continued): The authors need to incorporate these findings in the manuscript with relevant references.

Response: We have updated the language in the discussion section in response to this comment, with reference of the paper by Coorens, et al. Discussion, 5th paragraph: "Also in their study, the H19/ICR1 hypermethylation was detected in adjacent non-diseased kidney tissue and associated with clonal expansion of histologically normal renal cells deemed "clonal nephrogenesis." This clonal expansion of nephrogenic cells is thought to occur during kidney development, and provides the precursor cell population for multifocal and BWT development."

The process of clonal nephrogenesis being followed by 2nd hits including chromosome 22q loss and/or microRNA processing gene mutations is outlined in Figure 6, which summarizes the findings of the study.

Reviewer's response: The authors have addressed this comment.

Comment: The authors state that, "Because the data discussed above support 11p15.5 LOI somatic mosaicism as a major contributor to the predisposition for BWT development, we hypothesized and confirmed that patients and survivors of BWT had higher levels of H19/ICR1 methylation detectable in peripheral blood than a cohort of healthy community control subjects and unilateral WT." This is not relevant as these methylation changes were in the normal range. Please address this comment

Response: As the reviewer mentioned above, "low-level" changes in methylation could represent somatic mosaicism, but where the somatic mosaic changes fall below established thresholds used to define clinical syndromes. These "low-level" thresholds would be, by definition, within the "normal" range. Nevertheless, we have revised the manuscript to clearly indicate that the blood sample analysis is an exploratory analysis of potential "low-level" methylation changes that fall within the normal range as noted above.

Reviewer's response: If there is reference to low level methylation changes within normal methylation range then the thresholds of these methylation changes needs to be included as stated above.

Comment: The data on amplified clonal mosaicism increasing over time is limited, where the data on mosaic 11p15 changes in patients with Beckwith-Wiedemann Spectrum is more plentiful. Please comment on this in the context of the discussion.

Previous Response: We agree with this comment. We have added this observation to the the discussion section: Discussion (7th paragraph): "Progressive selection of mosaic hematopoietic cells containing uniparental paternal isodisomy for 11p15 has been suggested as a mechanism for late onset β -thalassemia major in patients who are heterozygous carriers of pathogenic HBB variants (which is also located at 11p15); however, evidence for this phenomenon is limited overall.⁴⁷"

Comment (Continued): The reference cited is not relevant. Please address this comment.

Response: The language about progressive selection of mosaic hematopoietic cells, this reference, and the analysis of methylation versus age/time have all been eliminated from the manuscript in response to the above reviewer comments.

Reviewer's response: The authors have addressed this comment.

Comment: How is 11p15 status a biomarker for global methylation patterns? Please revise this and provide the evidence to support that revision. Was it high or low stage risk? Both are stated?

Previous Response: When tumors are clustered according to methylation M values derived from the top 10,000 most variable CpG probes in the methylation array data set (using both unsupervised hierarchical or TSNE approaches), the bilateral Wilms tumors illustrate clustering according to 11p15 status as depicted in Figure 4 and Figure 5B. We do not understand if this pertains to risk or stage of tumors; however, among tumors with SIOP high-risk histology (which is used clinically to define treatment and prognosis), the majority had 11p15 LOI. This association will need to be explored prospectively in future studies.

The comment about 11p15 status determining outcome in low-risk tumors may be confusing in this context. This observation came from a study looking specifically at unilateral Wilms tumor with several low-risk parameters (age <2 years, tumor weight < 500g, stage I, favorable histology). It has been eliminated from the Discussion section to improve clarity of the manuscript and avoid confusion.

Comment (Continued): This needs to be further clarified in the text, the revision is still unclear.

Response: We have updated the language in the discussion section to clarify the conclusion that 11p15.5 status is a surrogate biomarker of more global methylation status: Discussion section, 7th paragraph: "Furthermore, our data show that 11p15.5 status can be used as a surrogate biomarker for more global methylation patterns/molecular subgroups of BWT because when samples were clustered according to broader, genome-wide patterns of methylation, they tended to cluster according to 11p15.5 status."

Reviewer's response: The authors have addressed this comment.

Overall, the authors have addressed most of the comments. The main comment to still be addressed is defining low level methylation changes within normal range. What is the threshold of these changes based on the data presented and does this have any impact?

REVIEWER COMMENTS

Reviewer #2 (Remarks to the Author):

Comment: If the authors are planning to keep this exploratory analysis in the manuscript, they need to further define the “new exploratory thresholds” they are using for the low level analysis and clarify why they did not use such thresholds for the larger study. Fiala et. al. are using normal range as $50.00\% \pm 1.12\%$ whereas the normal range for this study is “average β value $H19/ICR1 < 0.7$ and $KCNQ1OT1/ICR2 > 0.3$ were determined to have normal retention of imprinting (ROI)”. The authors need to provide something similar. Additionally a minor point, the scales of Y-axis in Figure 3 are not uniform.

Response: Thank you for your detailed review and constructive comments. We have now revised Figure 3 to utilize a uniform Y-axis scale from a Beta value of 0.4 to 0.9 in panels A-D.

The following modifications have been made to clearly define the “new exploratory thresholds”:

Methods, 2nd paragraph under “Methylation analysis” subheading: *“For the exploratory analysis of H19/ICR1 methylation in peripheral blood, “low level” gain of methylation at 11p15.5 H19/ICR1 was defined as a β value greater than two standard deviations above the mean H19/ICR1 β value of a healthy control cohort.”*

Results, 2nd paragraph under “Exploratory analysis of 11p15.5 H19/ICR1 methylation in peripheral blood” subheading: *“For this exploratory analysis, we defined “low-level” gain of methylation as an H19/ICR1 β value $>$ two standard deviations of the mean H19/ICR1 β value detected in the healthy community control cohort.”*

As noted above, all definitions for abnormal methylation, “low-level” methylation, and normal methylation are now appropriately included in the revised version of the manuscript. We have also now pointed out that Fiala et. al. used methylation-sensitive PCR as their assay for methylation and this may explain the difference in definition of “normal” between the two different modalities used in their study versus ours: *“Fiala et. al. also previously demonstrated “low-level” gain of methylation detectable in peripheral blood samples from 7 female patients with BWT, which they defined as two standard deviations above the mean H19/ICR1 methylation level in a healthy control cohort using methylation-sensitive PCR.²⁴”*

Previous Comment: The authors state that, “Because the data discussed above support 11p15.5 LOI somatic mosaicism as a major contributor to the predisposition for BWT development, we hypothesized and confirmed that patients and survivors of BWT had higher levels of H19/ICR1 methylation detectable in peripheral blood than a cohort of healthy community control subjects and unilateral WT.” This is not relevant as these methylation changes were in the normal range. Please address this comment

Previous Author Response: As the reviewer mentioned above, “low-level” changes in methylation could represent somatic mosaicism, but where the somatic mosaic changes fall below established thresholds used to define clinical syndromes. These “low-level” thresholds would be, by definition, within the “normal” range. Nevertheless, we have revised the manuscript to clearly indicate that the blood sample analysis is an exploratory analysis of potential “low-level” methylation changes that fall within the normal range as noted above.

Current Comment/Reviewer’s response: If there is reference to low level methylation changes within normal methylation range then the thresholds of these methylation changes needs to be included as stated above.

Response: Please see the above response regarding explicit definitions of “low-level” methylation changes included in the revised version of the manuscript.

Comment: Overall, the authors have addressed most of the comments. The main comment to still be addressed is defining low level methylation changes within normal range. What is the threshold of these changes based on the data presented and does this have any impact?

Response: Please see above responses regarding the explicit definitions included in this revised version of the manuscript regarding “low-level” methylation changes. In terms of the impact of these data and thresholds, the following language has been added to the discussion section, 8th paragraph: *“These findings demonstrate that a normal H19/ICR1 methylation value from a blood sample is insufficient to exclude mosaic 11p15.5 LOI in a patient with BWT, and non-diseased kidney and tumor samples may be required for definitive evaluation. Because 11p15.5 H19/ICR1 LOI is a purely epigenetic phenomenon, refinements in single-cell methylation or alternative approaches will be required to definitively evaluate an individual blood sample for evidence of “low-level” 11p15.5 LOI in the future.”*

Reviewers' Comments:

Reviewer #2:

Remarks to the Author:

The reviewer's comments have been addressed.

Response to Reviewers Document

REVIEWERS' COMMENTS

Reviewer #2 (Remarks to the Author):

The reviewer's comments have been addressed.

Response: Thank you again for your rigorous review of our manuscript.